# M4GN: Mesh-based Multi-segment Hierarchical Graph Network for Dynamic Simulations

**Bo Lei**                                                                                                *lei4@llnl.gov*
*Lawrence Livermore National Laboratory*

**Victor M. Castillo**                                                                           *castillo3@llnl.gov*
*Lawrence Livermore National Laboratory*

**Yeping Hu**                                                                                            *hu25@llnl.gov*
*Lawrence Livermore National Laboratory*

**Reviewed on OpenReview:** *https://openreview.net/forum?id=R3vDbqWa1v*

## Abstract

Mesh-based graph neural networks (GNNs) have become effective surrogates for PDE simulations, yet their deep message passing incurs high cost and over-smoothing on large, long-range meshes; hierarchical GNNs shorten propagation paths but still face two key obstacles: (i) building coarse graphs that respect mesh topology, geometry, and physical discontinuities, and (ii) maintaining fine-scale accuracy without sacrificing the speed gained from coarsening. We tackle these challenges with M4GN—a three-tier, segment-centric hierarchical network. M4GN begins with a hybrid segmentation strategy that pairs a fast graph partitioner with a superpixel-style refinement guided by modal-decomposition features, producing contiguous segments of dynamically consistent nodes. These segments are encoded by a permutation-invariant aggregator, avoiding the order sensitivity and quadratic cost of aggregation approaches used in prior works. The resulting information bridges a micro-level GNN—which captures local dynamics—and a macro-level transformer that reasons efficiently across segments, achieving a principled balance between accuracy and efficiency. Evaluated on multiple representative benchmark datasets, M4GN improves prediction accuracy by up to 56% while achieving up to 22% faster inference than state-of-the-art baselines.

## 1 Introduction

Numerically solving partial differential equations (PDEs) to model dynamical systems is fundamental in science and engineering but is often computationally intensive, especially in time-sensitive applications requiring rapid inference. This has prompted increased attention on adopting learning-based surrogate models (Sun et al., 2020) to expedite numerical simulations, addressing the computational challenges associated with traditional solvers. Among these methods, mesh-based Graph Neural Network (GNN) methods (Pfaff et al., 2020; Gao et al., 2022; Hu et al., 2023) have proved highly effective for simulating dynamical systems discretized on unstructured meshes. Information is propagated by stacking successive message-passing layers across mesh edges. When the mesh graph becomes very large—or when the underlying physics couples spatially distant regions (e.g., vortex–vortex interactions in fluids or boundary loads transmitted along an elastic beam)—a substantial number of message-passing iterations is required for information to traverse the graph (Fortunato et al., 2022). This growth in node count and propagation depth inflates computational cost, while the deeper propagation over-smooths node embeddings and erodes accuracy (Chen et al., 2020; Yang et al., 2020; Keriven, 2022). To mitigate these effects, recent work has introduced hierarchical GNNs that learn coarse graph

representations and pass messages across multiple resolutions, thereby shortening information-propagation paths and reducing the depth required for expressive receptive fields (Gao & Ji, 2019; Li et al., 2020; Lino et al., 2022). Such models have achieved state-of-the-art accuracy on challenging benchmarks and some of them have shown substantial speed-ups over single-scale mesh-based GNNs. Nevertheless, two limitations persist, which we examine in detail in the following subsections.

Table 1: Comparison of strategies used to construct coarse-level graphs in hierarchical GNN models for physics-based simulations. Here are the definitions for each evaluation criterion: *Heuristic*: the method is driven by a fixed, hand-crafted rule rather than by parameters learned from data; *Contiguity*: the method produces coarse elements that remain internally connected and never bridge gaps, holes, or material interfaces; *Geometric Fidelity*: the method maintains the original element shapes and sizes closely enough to avoid severe geometric distortion; *Physics-Aware*: the method incorporates physical information—such as material domains, boundary conditions, or regions of nearly uniform fields.

| Method | Heuristic | Contiguity | Geometric Fidelity | Physics Aware |
|---|---|---|---|---|
| (a) Learnable pooling (Gao & Ji, 2019) | ✗ | ✗ | ✗ | ✗ |
| (b) Spatial proximity pooling (Lino et al., 2022) | ✓ | ✗ | ✓ | ✗ |
| (c) Bi-Stride pooling (Cao et al., 2023) | ✓ | ✓ | ✗ | ✗ |
| (d) Same size $k$-means (Janny et al., 2023) | ✓ | ✗ | ✓ | ✗ |
| (e) Hybrid mesh graph segmentation (Ours) | ✓ | ✓ | ✓ | ✓ |

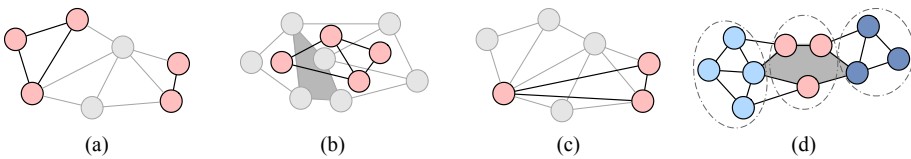

| (a) | (b) | (c) | (d) |

(a) Learnable pooling can indiscriminately merge distance vertices, disrupting connectivity, and—because it lacks a fixed heuristic—precludes offline sub-graph generation, shifting additional computation into the model runtime; (b) Spatial proximity pooling and (d) Same size $k$-means clustering ignore holes or interfaces (shaded areas), connecting/grouping nodes across gaps; (c) Bi-Stride keeps connectivity but its hop-count stripes warp geometry into elongated, jagged stripes with poor aspect ratios.

## 1.1 Challenge in sub-graph construction

One of the key elements of these hierarchical methods is to generate coarse-level graphs, yet existing strategies each have different drawbacks (Table 1). For example, learnable (Gao & Ji, 2019) or random pooling (Li et al., 2020) can introduce artificial partitions in the sub-level graphs, which impedes information exchange across partitions. Methods like spatial proximity pooling (Lino et al., 2022; Fortunato et al., 2022) can lead to wrong connections across the boundaries at the coarser level. While Bi-Stride (Cao et al., 2023) does guarantee 2-hop connectivity and avoids cross-boundary edges, its hop-count frontiers can severely warp the geometric metric. Alternatively, (Janny et al., 2023) preserves the original mesh and utilizes same size k-means to cluster mesh nodes by treating the mesh graph as point clouds. However, it ignores edge topology entirely: two nodes that are Euclidean-close yet separated by a crack, thin wall, or hole can be grouped together. Because their physical states are incompatible, pooling them into a single 'super-node' yields segment embeddings that misrepresent local physics; these distorted features propagate through the model and ultimately amplify prediction error (Cao et al., 2023; Chen et al., 2020). *Hence, the first challenge is to design a graph-coarsening strategy that simultaneously preserves mesh topology and geometric fidelity while respecting physical discontinuities.*

## 1.2 Challenge in balancing accuracy and efficiency

Having an appropriate segmentation or pooling strategy addresses only part of the problem; the organization of sub-level graphs into a multi-resolution hierarchy and the way information exchanges between levels are equally decisive for speed and fidelity. Most pooling-based hierarchies (Gao & Ji, 2019; Cao et al., 2023) adopt a U-Net-style encoder–decoder (Ronneberger et al., 2015): a lightweight gating function scores vertices,

the top-$k$ fraction is retained, and the rest are discarded. Each pooling step shrinks the graph rapidly, so deeper levels operate on far fewer nodes and enjoy substantial computational savings. The downside is that aggressive pooling or poorly designed coarsening levels act like an irreversible low-pass (and sometimes aliasing) filter on the graph signal (Chen et al., 2020; Li et al., 2020), so the high-frequency physics "smears or vanishes" when the prediction is mapped back to the fine mesh, resulting in accuracy reduction. By contrast, segment-based approaches such as EAGLE (Janny et al., 2023) keep the original mesh intact within each segment, ensuring that no local geometric or physical detail is discarded. However, their reliance on gated recurrent units (GRU) (Cho et al., 2014) means that every node in every segment incurs three gating operations per time step, so computational cost grows with both segment size and feature dimension. Beyond the runtime and memory overhead on large meshes, long GRU chains also suffer from order sensitivity and information dilution (Vinyals et al., 2015; Bengio et al., 1994), which can erode predictive accuracy even when enough compute is available. *The second challenge, therefore, is to design a hierarchical architecture that retains fine-grained geometric and physical cues like segment-based methods, yet extracts and propagates segment representations through an aggregator that remains permutation-invariant, information-preserving, and computationally light—thus achieving a principled balance between accuracy and efficiency.*

### 1.3 Contributions

To overcome the challenges outlined above and to comprehensively evaluate different surrogates, we make three primary contributions in this paper:

- **Hybrid mesh–graph segmentation.** We propose a two-stage mesh-graph segmentation strategy: (i) a lightweight graph partitioner first produces a coarse segmentation, and (ii) each partition is then adaptively refined by a superpixel-inspired algorithm guided by modal-decomposition features encoding local physics. The resulting segments preserve contiguity and geometric fidelity while encompassing nodes with coherent physical behavior.
- **Multi-segment hierarchical graph network.** We introduce M4GN, a three-tier, segment-centric hierarchy with two crucial innovations compared to previous works: (i) integration of a hybrid mesh–graph segmentation scheme that yields segments of higher geometric and physical fidelity, facilitating communication between adjacent hierarchy levels; and (ii) a permutation-invariant aggregator for extracting segment features, which is insensitive to mesh node orders and computationally efficient. As a result, M4GN preserves fine-scale physics (micro-level), compress them into faithful segment tokens (intermediate-level), and enables efficient inter-segment reasoning (macro-level), yielding an effective balance between predictive accuracy and computational efficiency.
- **Additional dataset and its scaled-up version.** We contribute DeformingBeam and its scaled counterpart DeformingBeam (large), the first public 3-D Lagrangian contact-deformation benchmark that includes a scale-up version. The meshes' elongated geometry produces graph diameters several times larger than those in previous solid-mechanics benchmarks, exposing explicit long-range interactions. It enables rigorous testing of hierarchical surrogates and serves as a benchmark for probing model scalability and cross-scale generalization.

## 2 Hybrid Mesh-graph Segmentation

This section is organized as follows. Section 2.1 motivates the hybrid mesh–graph segmentation approach. Section 2.2 introduces the mathematical notation, and Section 2.3 illustrates the modal-decomposition technique. Section 2.4 presents the complete segmentation pipeline, which serves as the intermediate-level module of the M4GN framework shown in Figure 1.

### 2.1 Motivation

To avoid the uninterpretable and potentially erroneous dynamics that coarsened graphs or added edges might introduce, we propose preserving the original mesh structure and facilitating long-range information exchange through communication between segmented mesh graphs. Traditional graph segmentation methods (Alpert & Yao, 1995; Delingette, 1999) often prioritize geometric properties and computational efficiency

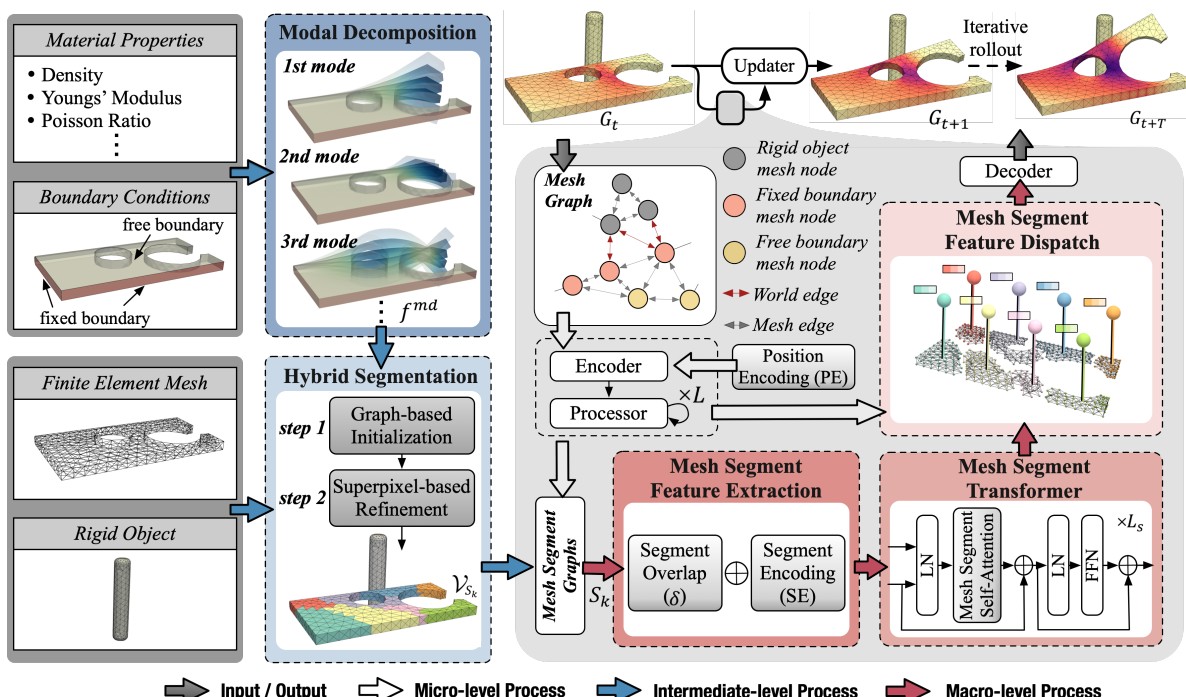

Figure 1: Architecture of the proposed M4GN framework: **M**esh-based **M**ulti-Segment Hierarchical (**M**icro-Intermediate-**M**acro) **G**raph **N**etwork. The framework operates on three modules: micro-level module to capture fine-scale dynamic, intermediate-level module to generate mesh-based segmentation, and macro-level module to model segment-level interactions. The colored arrows trace the data flow within each module.

over underlying physical attributes. Conversely, superpixel approaches (Veksler et al., 2010; Achanta et al., 2012) group pixels based on user-defined similarity measures but rely on careful cluster-center initialization to maintain segmentation quality. To merge the strengths of both, we apply a *graph-based method* ($f_{gb}$) for initial mesh segmentation and refine it using a *superpixel-based method* ($f_{sb}$), guided by leveraging features associated with dominant modes identified in the modal decomposition module (Section 2.3). In this way, we ensure these segments remain physically coherent and well-structured for effective communication. Grouping elements with similar physical properties enhances model convergence by minimizing discontinuities within each segment (Diao et al., 2023), while grouping nodes with similar behaviors streamlines learning and ensures uniform handling of similar interactions (Dolean et al., 2015). Eventually, this hybrid approach offers efficient and reliable geometric partitioning alongside adaptive, feature-based refinement, producing high-quality mesh segments adaptable to diverse dynamical systems.

## 2.2 Mathematical Notation

We define the segmentation policy $\pi(G) = f_s(G, I)$, where the segmentation function $f_s$ takes the input graph $G$ and prior physical information $I$ (e.g., boundary conditions, material properties), and outputs a set of graph segments $\{S_1^0, S_2^0 \ldots, S_K^0\}$. The superscript 0 denotes non-overlapping segmentation. For each segment $S_k = (\mathcal{V}_{S_k}, \mathcal{E}_{S_k})$, the set of nodes $\mathcal{V}_{S_k} \subseteq \mathcal{V}$ and $\mathcal{E}_{S_k} \subseteq \mathcal{E}$ are subsets of the original graph $G$. The union of all segments reconstructs the original graph, such that $\mathcal{V} = \cup \mathcal{V}_{S_k}^0$ and $\mathcal{E} = \cup \mathcal{E}_{S_k}^0$. In some cases, it may be beneficial to allow for overlapping segments, where nodes in $\mathcal{V}$ can belong to more than one segment. This overlap helps create smoother transitions between segments and reduces discontinuities at segment boundaries. We define the overlap amount by $\delta \in \mathbb{N}$, with $\delta = 0$ representing no overlap. For $\delta > 0$, the node set $\mathcal{V}_{S_k}^\delta$ is defined recursively as $\mathcal{V}_{S_k}^\delta = \mathcal{V}_{S_k}^{\delta-1} \cup \{Adj(i) \mid i \in \mathcal{V}_{S_k}^{\delta-1}\}$. To simplify the presentation, we disregard the superscript $\delta$ in the remainder of this paper and use $\delta = 1$ for all experiments with overlapping. The effect of adding overlapping segments is discussed in our ablation study, as shown in Table 9.

### 2.3 Modal Decomposition

Modal decomposition is a fundamental technique for extracting dominant spatiotemporal patterns, or *modes*, from complex physical systems (Fu & He, 2001; Schmid et al., 2011; Taira et al., 2017). Each mode encapsulates coherent behavior—such as a characteristic deformation shape or flow structure—allowing a reduced but meaningful representation of the underlying dynamics. In complex physical simulations, these dominant modes can effectively guide downstream tasks such as mesh segmentation, where the domain is subdivided based on physical coherence (Yang et al., 2016; Huang et al., 2009). In this work, we employ two different modal decomposition approaches to address *solid* and *fluid* problems separately, given their distinct physical behaviors (Bathe, 2001). The pseudocode of the modal decomposition module can be found in Algorithm 1.

**Structural Modal Analysis:** For solids, the decomposition naturally arises from the mass–stiffness relationship in elastodynamics, capturing genuine dynamic displacements (Andersen, 2006). Let $\mathbf{K}$ be the global stiffness matrix and $\mathbf{M}$ the global mass matrix arising from finite element assembly. The free vibration modes of a structure are obtained by solving the generalized eigenvalue problem:

$$\mathbf{K}\boldsymbol{\phi} = \lambda \mathbf{M}\boldsymbol{\phi}, \tag{1}$$

where $\lambda$ represents the square of the natural frequency, and $\boldsymbol{\phi} = (\phi_1, \phi_2, \ldots, \phi_{\dim})$ is the corresponding *structural modes*, whose dimension (dim) matches the number of displacement components (e.g., 2D or 3D). Physically, each mode shape indicates a fundamental deformation pattern under vibrational motion (Fu & He, 2001; Wilson, 2002), which is tied to the solid's geometry, boundary conditions, and material parameters. In practice, it is typical to select the first $m$ modes ($\lambda_1 \le \lambda_2 \le \cdots \le \lambda_m$) to construct an $m$-dimensional feature at each mesh node $i$: $f_i^{md} = (\boldsymbol{\phi}_1(i), \boldsymbol{\phi}_2(i), \ldots, \boldsymbol{\phi}_m(i))$.

**Laplacian Eigenfunctions:** In fluid contexts, particularly when lacking multiple snapshots or a steady base flow (Wang et al., 2024), Laplacian eigenfunctions (Grebenkov & Nguyen, 2013) are used to capture geometry- and boundary-driven harmonic modes by solving:

$$-\nabla^2\phi = \lambda\phi, \text{ subject to boundary constraints}, \tag{2}$$

yielding *harmonic modes* $\phi_1, \ldots, \phi_m$. These modes serve as a practical proxy for flow-related structures, providing a minimal but informative decomposition that respects the domain shape and boundary conditions (De Witt et al., 2012; Taira et al., 2017). Similar to the solid case, each node $i$ in the fluid mesh is associated with a feature vector: $f_i^{md} = (\phi_1(i), \phi_2(i), \ldots, \phi_m(i))$.

### 2.4 Detailed Methodology

In the hybrid segmentation module, we first use METIS (Karypis & Kumar, 1998) for initial mesh segmentation due to its great balance of partition quality and speed. Formally, given a graph $G$, the partition function $f_{gb}$ will split it into $K$ non-overlapped mesh-segment graphs: $\{S_1, \ldots, S_K \mid S_i \cap S_j = \varnothing, \forall i \ne j\} = f_{gb}(G)$. Then, we apply SLIC (Achanta et al., 2012), the state-of-the-art superpixel-based clustering methods, to these mesh segments to iteratively update the segmentation centroids $\{C_1, \ldots, C_K\}$ and corresponding node assignments using information obtained from modal decomposition (Section 2.3). It is worth noting that standard modal decomposition does not account for external obstacles (Fu & He, 2001). Therefore, in models with moving rigid objects, this information will need to be incorporated separately. For node $i$ in graph $G$, we represent it by its spatial coordinates $\mathbf{x}_i$, features related to rigid objects or obstacles $f_i^{obs}$, and features obtained from modal decomposition $f_i^{md}$. For a given mesh segment $S_k$ containing $|\mathcal{V}_{S_k}|$ nodes, we define its centroid $C_k$ as its mean value along the features:

$$C_k = [\mathbf{x}_{C_k}, f_{C_k}^{obs}, f_{C_k}^{md}]^T = \frac{1}{|\mathcal{V}_{S_k}|} \sum_{i \in \mathcal{V}_{S_k}} [\mathbf{x}_i, f_i^{obs}, f_i^{md}]^T. \tag{3}$$

Within each iteration, we improve the mesh segmentation by minimizing a distance measure that considers both physical similarity and spatial proximity. The distance measure $d(i, C_k)$ between a node $i \in \mathcal{V}$ and a segment's centroid $C_k$ is defined as:

$$d(i, C_k) = \|f_i^{obs} - f_{C_k}^{obs}\| + \|f_i^{md} - f_{C_k}^{md}\| + \tau\|\mathbf{x}_i - \mathbf{x}_{C_k}\|, \tag{4}$$

where $\tau$ is used to control the compactness of a mesh segment. The pseudo code of the hybrid segmentation module can be found in Algorithm 2. In Appendix B.4, we present a comprehensive comparison of various segmentation methods and their variants based on different distance measures. Additionally, we evaluate the impact of varying the number of mesh segments on model performance in Appendix D.5 and Appendix E.2. We also introduce several metrics to measure the quality of different mesh segmentations, specifically to understand the intra-segment and inter-segment characteristics, which can be found in Appendix C.2

## 3 M4GN: Multi-segment Hierarchical Graph Network

This section details our proposed hierarchical framework (Figure 1). After a formal problem statement in Section 3.1, we describe its three-level modules: (i) a micro-level module (Section 3.2) that performs message passing along mesh edges to capture fine-scale dynamics; (ii) an intermediate-level segment module (Section 2) that constructs mesh segments offline using our heuristic hybrid segmentation algorithm; and (iii) a macro-level module (Section 3.3) that aggregates segment features and exchanges information across segments to model long-range interactions.

### 3.1 Problem Definition

Let $G = (\mathcal{V}, \mathcal{E})$ be a mesh graph with $\mathcal{V}$ being the set of nodes and $\mathcal{E}$ being the set of edges. The graph has $N = |\mathcal{V}|$ nodes and $E = |\mathcal{E}|$ edges, with adjacency matrix $A \in \mathbb{R}^{N \times N}$ representing graph connectivity. The dynamic simulation task is to learn a forward model of the dynamic quantities of the mesh graph at the next time step $\hat{G}_{t+1}$ given the current mesh graph $G_t$ and (optionally) a history of previous mesh graphs $\{G_{t-1}, \ldots, G_{t-h}\}$. Finally, the rollout trajectory can be generated through the simulator iteratively based on the previous prediction: $G_t, \hat{G}_{t+1}, \ldots, \hat{G}_{t+T}$, where $T$ is the total simulation steps. In this paper, the proposed model (M4GN) can simulate both Eulerian and Lagrangian systems (Bontempi & Faravelli, 1998). For Eulerian systems examined in this paper, where continuous fields such as velocity evolve on a stationary mesh, the graph $\mathcal{E}$ includes only mesh-related edges $\mathcal{E}^M$. Conversely, for Lagrangian systems considered in this paper, where the mesh represents a moving and deforming surface or volume, additional world edges $\mathcal{E}^W$ are incorporated into the graph. These edges enable the model to learn external dynamics such as collision and contact. The node features of node $i$ are denoted by $\mathbf{x}_i$, while the features for an edge between node $i$ and $j$ are indicated by $\mathbf{e}_{ij}$.

### 3.2 Micro-level Module

Within the micro-level module, each node engages in the exchange of information with its neighboring nodes. This process holds particular significance in dynamical systems, where the behavior of adjacent nodes is closely intertwined (Booij & Holthuijsen, 1987; Emanuel, 1994; Fahy, 2007; Kennett, 2009). Furthermore, this module serves a crucial role in addressing discontinuities that may arise at the boundaries of adjacent mesh segments (Lai et al., 2009). By prioritizing micro-level information exchange, we effectively mitigate discontinuities introduced by subsequent macro-level operations. We follow the Encoder-Process-Decoder (EPD)(Pfaff et al., 2020) architecture for our micro-level information exchange as it has shown great performance in dealing with mesh-based graphs. Specifically, we use the encoder and processor blocks for micro-level message passing, while the decoder head is detached and applied only after the macro-level module (Section 3.3). For a given graph $G_t$ at time $t$, the model begins with extracting node and edge features through two separate Multi-Layer Perceptrons (MLPs):

$$\mathbf{h}_{i,t}^0 = f_n(\mathbf{x}_{i,t}), \quad \mathbf{h}_{ij,t}^{M,0} = f_e^M(\mathbf{e}_{ij,t}^M), \quad \mathbf{h}_{ij,t}^{W,0} = f_e^W(\mathbf{e}_{ij,t}^W), \tag{5}$$

where $\mathbf{x}_{i,t} \in \mathcal{V}$, $\mathbf{e}_{ij,t}^M \in \mathcal{E}^M$, and $\mathbf{e}_{ij,t}^W \in \mathcal{E}^W$ denote node feature, mesh edge feature, and world edge feature vector at time $t$, respectively. For Lagrangian systems, world edges are created by spatial proximity, where for a fixed radius $r_W$, a world edge is added between nodes $i$ and $j$ when $|\mathbf{x}_i - \mathbf{x}_j| < r_W$, excluding node pairs already connected in the mesh. The outputs of two MLPs (i.e. $f_n$ and $f_e$) for node and edge are denoted as $\mathbf{h}_{i,t}^0$ and $\mathbf{h}_{ij,t}^0$, respectively. Then, a $L$-step message passing (MP) is performed such that each node can receive and aggregate information from neighboring nodes within $L$ steps of edge traversing. For each MP

from 1 to $L$, the node and edge representations are updated as:

$$\mathbf{h}_{i,t}^l = f_n^l(\mathbf{h}_{i,t}^{l-1}, \sum_{j \in Adj(i)} \mathbf{h}_{ij,t}^{M,l-1}, \sum_{j \in Adj(i)} \mathbf{h}_{ij,t}^{W,l-1}), \tag{6}$$

$$\mathbf{h}_{ij,t}^{M,l} = f_e^l(\mathbf{h}_{ij,t}^{M,l-1}, \mathbf{h}_{i,t}^{l-1}, \mathbf{h}_{j,t}^{l-1}), \quad \mathbf{h}_{ij,t}^{W,l} = f_e^l(\mathbf{h}_{ij,t}^{W,l-1}, \mathbf{h}_{i,t}^{l-1}, \mathbf{h}_{j,t}^{l-1}), \tag{7}$$

where $Adj(i)$ denotes all adjacent nodes of node $i$. Up until this point, the node and edge information of the graph $G_t$ are updated. Additionally, we implement a technique from (Godwin et al., 2021), which involves corrupting the input graph with noise and adding a noise-correcting node-level loss. We evaluate the impact of varying the number of message passing steps during micro-level information exchange step, where details can be found in Appendix D.1.

### 3.3 Macro-level Module

#### 3.3.1 Mesh Segment Feature Extraction

**Segment Encoding (SE)** – In order to extract a global feature for each mesh segment, we perform average pooling on all node vectors in $S_k$ and apply a MLP ($f_s$) to get the fixed-sized segment embedding: $\mathbf{h}_{S_k,t} = f_s(\frac{1}{|\mathcal{V}_{S_k}|} \sum_{i \in \mathcal{V}_{S_k}} \mathbf{h}_{i,t}^L)$. Different from the state-of-the-art clustering-based hierarchical model (Janny et al., 2023), we replace the GRU-based aggregator with a permutation-invariant pooling operation when extracting segment features, which provides a more efficient and robust alternative for segment-level representation. It sidesteps the order bias and gradient noise of sequence models, yielding representations that remain stable across epochs and insensitive to mesh reordering (Vinyals et al., 2015). Moreover, average pooling circumvents the information-dilution problem inherent in long RNN chains (Bengio et al., 1994), preserving salient local features from vanishing-gradient effects and thereby enhancing predictive accuracy. Meanwhile, its computational burden is only $O(Nd)$, compared with the $O(Nd^2)$ matrix operations and lengthy back-propagation demanded by GRU pooling (Chung et al., 2014). These properties make average pooling a proper choice when accuracy and efficiency need to be balanced.

**Position Encoding (PE)** – As dynamic effect propagates continuously over mesh domains, knowing relative location among segments could provide extra information for next-step macro-level information exchange and increase expressivity of the network. Mathematically, for each pair of mesh segment graph, $\{S_i, S_j\}$, their relative positional information can be obtained through segment-level adjacency matrix $A^K \in \mathbb{R}^{K \times K}$: $A_{S_i S_j}^K = \sum_{m \in \mathcal{V}_{S_i}} \sum_{n \in \mathcal{V}_{S_j}} A_{mn}$. We follow the strategy in (Rampášek et al., 2022) that uses random-walk structural encoding (RWSE) (Dwivedi et al., 2021) for PE calculation. Then the PE for the $k$-th segment, denoted as $\mathbf{p}_{S_k,t}$, is processed through an MLP layer ($f_{sp}$) and then added to update the SE as follows: $\mathbf{h}_{S_k,t} \leftarrow \mathbf{h}_{S_k,t} + f_{sp}(\mathbf{p}_{S_k,t})$. We can further enhance the network's expressivity by adding absolute PE to the graph nodes. We use an MLP ($f_{np}$) to process each node's PE ($\mathbf{p}_{i,t}$), calculated with a similar approach as segment level, and add it to the input node feature. Thus, Eq (5) becomes $\mathbf{h}_{i,t}^0 = f_n(\mathbf{x}_{i,t} + f_{np}(\mathbf{p}_{i,t}))$. By incorporating node PE directly into the input features, these features participate in the micro-level information exchange described in Section 3.2, potentially improving the continuity of the extracted mesh segment features. Table 10 presents ablation results illustrating the impact of including or excluding PE on prediction performance.

#### 3.3.2 Mesh Segment Transformer, Feature Dispatch, and Training

We construct a fully connected mesh segment graph, where the $i$-th mesh segment feature is represented by $\mathbf{h}_{S_i}$. Note that since the transformer operates on mesh segments rather than individual mesh nodes, and the total number of mesh segments ($K$) is significantly smaller than the total number of mesh nodes ($N$), the computational cost of our transformer is substantially reduced compared to a traditional graph transformer that operates on graph nodes (i.e. $O(K^2) \ll O(N^2)$). The $l$-th block of the mesh segment transformer layer is defined as follows:

$$\bar{\mathbf{h}}_{S_i}^l = \|_{k=1}^H \sum_{j=1}^K \mathbf{a}_{S_i S_j}^{k,l}(\mathbf{V}_h^{k,l} \text{LN}(\mathbf{h}_{S_j}^l)), \quad \mathbf{h}_{S_i}^{l+1} = \mathbf{h}_{S_i}^l + \mathbf{O}_h^l \bar{\mathbf{h}}_{S_i}^l + \text{FFN}_h^l(\text{LN}(\mathbf{h}_{S_i}^l + \mathbf{O}_h^l \bar{\mathbf{h}}_{S_i}^l)), \tag{8}$$

where $\mathbf{a}^{k,l}_{S_i S_j}$ is self-attention weight between $S_i$ and $S_j$, $\mathbf{V}^{k,l}_h \in \mathbb{R}^{d_h \times d}$ is a trainable parameter matrix, and $\mathbf{O}^l_h \in \mathbb{R}^{d \times d}$ is the learned output project matrix. $k = 1$ to $H$ denotes the number of attention heads, and $\|$ denotes concatenation. $d_h$ is the dimension of mesh segment feature for each head, and $d$ is the input and output dimension. We adopt a Pre-Layer Norm architecture (Xiong et al., 2020), which is denoted as $\mathrm{LN}(\cdot)$, and the point-wise Feed Forward Network is represented as $\mathrm{FFN}(\cdot)$. The mesh segment transformer module facilitates information exchange among all mesh segments, updating the feature of each segment $\mathbf{h}_{S_i}$ after passing through $L_S$ mesh segment transformer blocks.

The mesh segment feature dispatch module (as shown in Figure 1) integrates information obtained from both macro-level and micro-level exchanges. Specifically, the final feature for node $i$ at time step $t$ is updated as $\mathbf{h}_{i,t} \leftarrow [\mathbf{h}_{i,t}, \mathbf{h}_{S_i,t}]$ where $i \in \mathcal{V}_{S_i}$. This ensures that each node incorporates information from both neighboring mesh nodes and spatially distant, yet correlated regions. Finally, we train our dynamics model by supervising on the per-node output features $\hat{\mathbf{x}}_{i,t+1}$, produced by feeding $\mathbf{h}_{i,t}$ into an MLP-based decoder, using an $L_2$ loss between $\hat{\mathbf{x}}_{i,t+1}$ and the corresponding ground truth values $\mathbf{x}_{i,t+1}$.

# 4 Experiment

## 4.1 Experiment Setup

**Datasets** – We benchmark on six datasets that together capture the two principal application regimes for mesh–based surrogates—Eulerian incompressible flow and Lagrangian hyper-elastic solids—while also probing both short-range and long-range interactions, small and large graphs, and steady versus highly-transient behavior. *CylinderFlow* and *DeformingPlate* are the widely-used public datasets of (Pfaff et al., 2020). CylinderFlow varies cylinder diameter, position and inlet velocity, while DeformingPlate changes obstacle trajectory, plate geometry, and boundary conditions. To challenge long-range coupling we introduce the *DeformingBeam*, a 3-D hyper-elastic beam whose length, cross-section, and end loads vary, yielding the largest graph diameter in the suite. Moreover, *DeformingBeam-Large* doubles the physical span of that beam and the total number of meshes to probe scalability. Finally, we utilize two supplementary benchmarks: *FlagSimple* (Pfaff et al., 2020) and *EAGLE* (Janny et al., 2023) to further demonstrate the robustness of M4GN across distinct physical regimes and dataset types. For these two datasets, we report results in the Appendix and do not perform full ablation analysis to keep the experimental scope tractable and to maintain clarity in interpreting results. This mix exercises both short-range and long-range interactions, small and large graphs, and steady versus highly-transient behavior. Because geometry, loading and obstacle parameters all change from run to run, the sets jointly test a surrogate's ability to generalize across shape, boundary condition and scale variations, giving a balanced yet concise benchmark suite.

At each time step, the network is provided with (i) nodal coordinates and edge vectors that describe the local mesh geometry, (ii) the current physical state, and (iii) categorical node masks identifying walls, inlets/outlets, or fixed/handle/obstacle regions. For the modal-decomposition stage, we further supply the material parameters and boundary conditions already defined in the finite-element setup (density, Young's modulus, Poisson's ratio). The network outputs the next-step state (e.g., positions for solids) which is recursively fed back as input to generate full rollouts. Comprehensive dataset specifications are provided in Appendix A.

**M4GN and Baselines** – As a default configuration for our M4GN model, we use 7 message passing steps in the mesh graph network. The mesh segment transformer adopts 4 self-attention layers with 8 heads. We compare our method to five baseline models: 1) *GCN* (Kipf & Welling, 2016; Belbute-Peres et al., 2020), a basic GNN structure widely used for simulating fluid dynamics; 2) *g-U-Nets* (Gao & Ji, 2019; Alsentzer et al., 2020), a representative method that incorporates graph pooling modules to enhance long-range interactions; 3) *MeshGraphNets* (MGNs) (Pfaff et al., 2020), a single-level GNN architecture that achieves great performance and generalizability across various dynamical systems; 4) *BSMS-GNN* (Cao et al., 2023), a recent work featuring a multi-level hierarchical GNN architecture that aims to enhance computational efficiency in simulating physical systems; and 5) *EAGLE*(Janny et al., 2023), a recent work presenting a clustering-based pooling method along with transformer to enhance performance on large-scale turbulent fluid dynamics. Detailed descriptions of the these models and training procedures can be found in Appendix B.

**Metrics** – In addition to traditional accuracy metrics, we also report mesh-quality metrics (Table 2) for the Lagrangian cases, where the computational mesh deforms with the material and may accumulate distortion. These evaluations are not required for the Eulerian dataset we used, because its meshes remain fixed in space and therefore cannot experience element-quality degradation. Mesh-quality metrics serve objectives that a nodal RMSE alone cannot address: (i) Many workflows feed the predicted mesh into inverse design or topology-optimization loops (Han et al., 2012); if elements are inverted or highly skewed, gradient-based updates fail even when the nodal RMSE is low. (ii) Well-shaped, smoothly graded elements improve the numerical conditioning of the surrogate, yielding higher predictive accuracy and better generalization (Kamenski et al., 2014). Four mesh quality metrics are used with concise descriptions in Table 2, where each metric targets a distinct failure mode, so a model may excel in one yet falter in another. For instance, a mesh can align perfectly with the true geometry while still containing many sliver elements. Evaluating all four metrics together thus offers a genuinely comprehensive view of the surrogate's output quality. More detailed descriptions and mathematical definitions of these metrics can be found in Appendix C.1.

Table 2: Compact summary of mesh–quality metrics for evaluating prediction results

| Metric | Geometric intuition | What it catches | Impact on simulation outcomes |
|---|---|---|---|
| Hausdorff ($GF_h$) | Max node–surface gap | Isolated large outliers | Drives worst-case error; signals local shape failures. |
| Chamfer ($GF_c$) | Mean node–surface gap | Uniform global drift | Lowers overall fidelity; raises global error norms. |
| Mesh Continuity (MC) | Neighbour cell–volume ratio | Abrupt size "cliffs" | Introduces artificial discontinuities; noisy spatial gradients. |
| Aspect-Ratio error (AR) | Deviation from ideal element shape | Slivers / stretched cells | Injects anisotropic bias; hampers generalization. |

## 4.2 Results and Discussion

### 4.2.1 Overall Performance Evaluation Across Multiple Datasets

The quantitative results in Table 3 show that M4GN outperforms all baselines across multiple evaluation metrics. Specifically, for the CylinderFlow dataset, M4GN achieves a 36% reduction in test RMSE-all compared to the second-best performing model, EAGLE. For the DeformingPlate dataset, M4GN reduces the test RMSE-all by 32%. This improvement is even more pronounced for DeformingBeam dataset, where M4GN demonstrates a 56% reduction in test RMSE-all. Such performance in 50-step and longer-step predictions underscores its enhanced capability for long-term predictions. In addition to its high prediction accuracy, M4GN demonstrates strong mesh quality, with up to a 48% reduction in GF and a 14% reduction in MC compared to the second-best model across both Lagrangian system datasets.

### 4.2.2 Segmentation Quality and Its Impact on Performance Metrics

To rigorously assess the quality of our hybrid segmentation strategy and its influence on dynamics prediction, we employ three metrics—*Conductance*, *Edge-Cut Ratio*, and *Silhouette Score*—that quantify intra-segment cohesion and inter-segment separation. These metrics assess segment isolation and intra-segment homogeneity properties that prevent feature dilution and lower prediction error. Formal definitions appear in Appendix C.2. Figure 2(a–c) correlates segmentation scores with mesh quality and prediction error for EAGLE's node-based partitions and three variants of our hybrid scheme (see Appendix B.4). The plots confirm that segmentation choice matters: our hybrid method, which better aligns partitions with underlying dynamic behaviors, consistently reduces error and improves mesh quality. Figure 2(d–f) color nodes by each segment's mean prediction error at successive time steps. Our segments stay nearly uniform in color across all datasets, indicating coherent intra-segment dynamics. For example, in (d), segmentation follows periodic wave patterns in fluid dynamics, while in (f), it reflects symmetrical system dynamics with symmetric segment coloring. These visualizations demonstrate that our segmentation effectively captures the temporal and spatial dynamics of the system, outperforming state-of-the-art method. Additional in-depth analysis can be found in Appendix D.2 and D.3.

Table 3: Comparison of results with state-of-the-art methods across three datasets, where each model is trained independently for each dataset. Prediction accuracy is evaluated using Root Mean Square Error (RMSE), with the output being the next-step positions for DeformingPlate/Beam and velocity/pressure fields for CylinderFlow. Errors are reported for 1-step rollout, 50-step rollouts, and the entire trajectory. Each mesh quality metric is evaluated at every time step then averaged over time—by comparing the predicted mesh to the ground-truth configuration. Results are averaged over three experiments with different random seeds and presented as mean and standard deviation.

| Dataset | Model | Mesh Quality Metrics ↓ | | | | Prediction Error Metrics ↓ | | |
|---|---|---|---|---|---|---|---|---|
| | | $GF_h$ $(\times 10^{-3})$ | $GF_c$ $(\times 10^{-6})$ | MC $(\times 10^{-3})$ | AR $(\times 10^{-3})$ | RMSE-1 $(\times 10^{-5})$ | RMSE-50 $(\times 10^{-4})$ | RMSE-all $(\times 10^{-4})$ |
| Cylinder Flow | GCN | - | - | - | - | $675 \pm 28$ | $382 \pm 69$ | $1702 \pm 310$ |
| | $g$-U-Net | - | - | - | - | $401 \pm 3.7$ | $179 \pm 32$ | $758 \pm 88$ |
| | MGN | - | - | - | - | $246 \pm 13$ | $62.8 \pm 2.8$ | $412 \pm 48$ |
| | BSMS-GNN | - | - | - | - | $\mathbf{181 \pm 22}$ | $252 \pm 8.2$ | $1218 \pm 83$ |
| | EAGLE | - | - | - | - | $456 \pm 24$ | $69.6 \pm 3.0$ | $525 \pm 24$ |
| | M4GN (Ours) | - | - | - | - | $288 \pm 19$ | $\mathbf{60.2 \pm 2.4}$ | $\mathbf{337 \pm 21}$ |
| Deforming Plate | GCN | $24.0 \pm 0.6$ | $323 \pm 4$ | $11.0 \pm 0.3$ | $9.33 \pm 0.57$ | $34.8 \pm 0.6$ | $26.1 \pm 0.1$ | $169 \pm 1$ |
| | $g$-U-Net | $36.1 \pm 8.5$ | $452 \pm 125$ | $20.1 \pm 0.5$ | $12.4 \pm 4.3$ | $41.2 \pm 0.2$ | $30.4 \pm 0.8$ | $179 \pm 7$ |
| | MGN | $12.7 \pm 0.9$ | $248 \pm 12$ | $9.25 \pm 0.39$ | $5.34 \pm 0.26$ | $\mathbf{22.8 \pm 0.2}$ | $20.0 \pm 0.4$ | $147 \pm 3$ |
| | BSMS-GNN | $23.8 \pm 2.6$ | $170 \pm 13$ | $18.3 \pm 4.4$ | $15.4 \pm 5.9$ | $30.3 \pm 5.6$ | $23.7 \pm 3.5$ | $118 \pm 4$ |
| | EAGLE | $6.75 \pm 0.8$ | $41.1 \pm 2.6$ | $5.56 \pm 0.12$ | $3.31 \pm 0.04$ | $36.4 \pm 5.2$ | $5.63 \pm 1.7$ | $38.7 \pm 1.8$ |
| | M4GN (Ours) | $\mathbf{4.29 \pm 0.07}$ | $\mathbf{7.05 \pm 1.05}$ | $\mathbf{4.82 \pm 0.06}$ | $\mathbf{2.67 \pm 0.06}$ | $26.7 \pm 0.5$ | $\mathbf{3.03 \pm 0.16}$ | $\mathbf{26.5 \pm 2.4}$ |
| Deforming Beam | GCN | $4.91 \pm 0.36$ | $3.53 \pm 0.51$ | $54.8 \pm 8.2$ | $69.5 \pm 3.8$ | $7.25 \pm 0.12$ | $5.08 \pm 0.11$ | $30.7 \pm 4.1$ |
| | $g$-U-Net | $4.91 \pm 0.50$ | $3.55 \pm 0.73$ | $34.7 \pm 1.8$ | $31.5 \pm 1.2$ | $7.28 \pm 0.39$ | $5.09 \pm 0.23$ | $31.7 \pm 4.0$ |
| | MGN | $0.82 \pm 0.04$ | $0.12 \pm 0.01$ | $16.9 \pm 0.1$ | $7.43 \pm 0.10$ | $4.43 \pm 0.08$ | $2.41 \pm 0.16$ | $4.72 \pm 0.27$ |
| | BSMS-GNN | $0.99 \pm 0.03$ | $0.21 \pm 0.04$ | $32.5 \pm 0.5$ | $16.1 \pm 0.3$ | $6.86 \pm 0.09$ | $1.95 \pm 0.22$ | $4.98 \pm 0.71$ |
| | EAGLE | $0.64 \pm 0.04$ | $0.17 \pm 0.01$ | $5.98 \pm 0.43$ | $5.17 \pm 0.37$ | $1.51 \pm 0.04$ | $0.67 \pm 0.12$ | $4.22 \pm 0.30$ |
| | M4GN (Ours) | $\mathbf{0.31 \pm 0.01}$ | $\mathbf{0.05 \pm 0.00}$ | $5.26 \pm 0.04$ | $3.08 \pm 0.06$ | $\mathbf{1.17 \pm 0.01}$ | $\mathbf{0.34 \pm 0.02}$ | $\mathbf{1.87 \pm 0.12}$ |

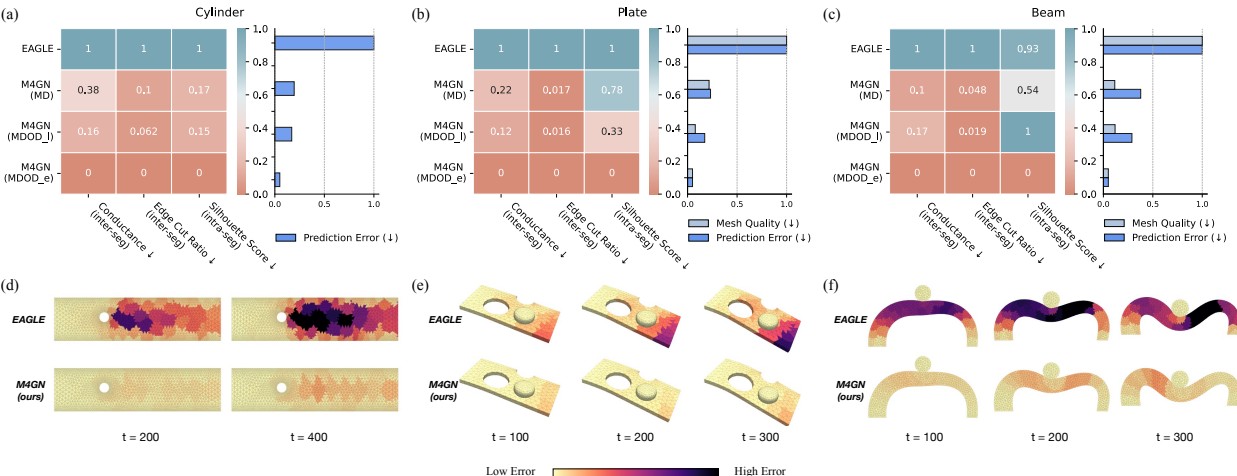

Figure 2: (a-c) Evaluation of different segmentation methods under three datasets. The heatmap (left) presents normalized Conductance, Edge Cut Ratio, and reversed Silhouette Score for EAGLE and three M4GN variants. Metrics are scaled between 0 and 1, where lower values indicate better segmentation quality. The sidebar plot (right) depicts normalized Prediction Error and Mesh Quality, with a minimum value of 0.05 applied to avoid invisible bars. (d-f) Visualization of simulation rollouts over time for three datasets, comparing our segmentation method with EAGLE. Nodes are colored based on the average prediction error within their segments.

### 4.2.3 Accuracy–Efficiency Trade-off Analysis

Figure 3(a) shows distinct trade-offs: MGN excels on the small-diameter CylinderFlow but loses accuracy and speed on DeformingPlate and DeformingBeam, where deep message passing leads to over-smoothing and higher latency; BSMS is memory-efficient thanks to bi-stride pooling, yet that pooling lowers mesh fidelity and accuracy and slows inference when long-range details must be reconstructed; and EAGLE offers moderate

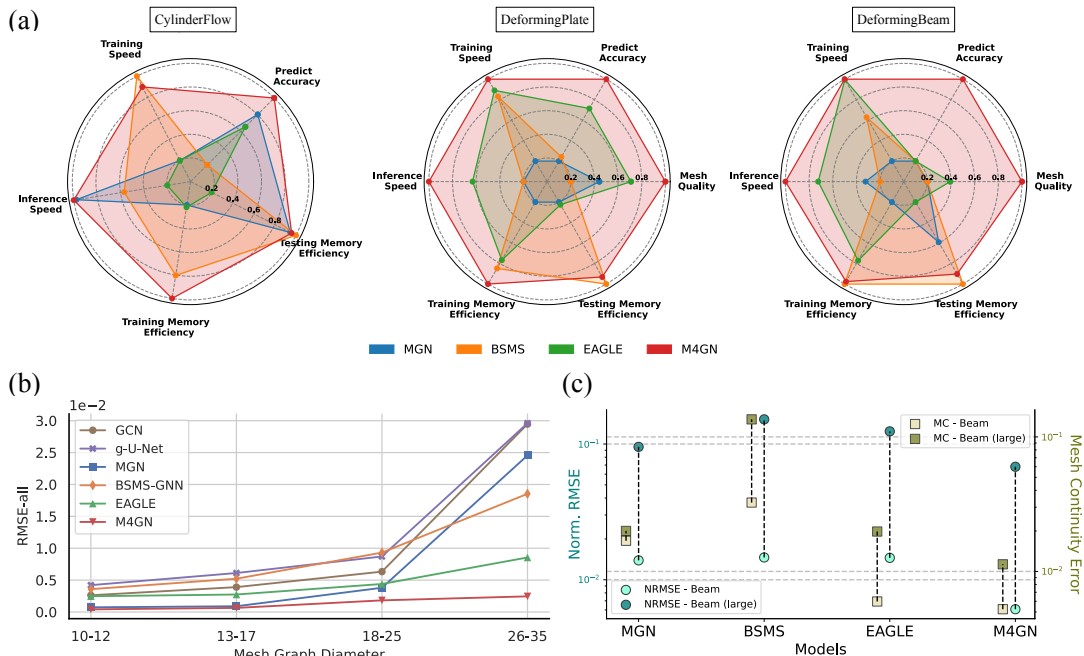

Figure 3: (a) Radar charts summarizing model performance on three datasets. All metrics are min-max normalized to the range $[0.2, 1.0]$; "lower-is-better" metrics are first inverted, and values below 0.2 are clipped to avoid visual collapse; larger filled areas reflect better overall performance. (b) RMSE-all accuracy versus graph diameter on the DeformingPlate dataset. (c) Performance comparison of each model when generalizing from DeformingBeam to DeformingBeam (large). Circles denote normalized RMSE (left axis), and squares denote mesh continuity error (right axis). Dashed lines highlight performance shifts. Lower values indicate better accuracy and mesh quality.

performance overall—its physics-agnostic clustering limits long-range capture, remaining efficient on the some datasets but slowing sharply on CylinderFlow as dense meshes inflate the number of required clusters. By contrast, M4GN maintains high accuracy and efficiency across all cases because its hybrid segmentation groups physically coherent nodes and its three-level hierarchy offloads long-range reasoning to a lightweight segment-level transformer, reducing both message-passing depth and token count. This design lets M4GN preserve local fidelity while controlling computational cost, yielding the balanced performance seen in the plots. Additional qualitative results are provided in Table 12.

### 4.2.4 Performance Analysis Across Graph Diameter and Scale

One of the main goals of hierarchical GNNs is to alleviate over-smoothing and capture long-range interactions. Exploiting the wide diameter range in our DeformingBeam benchmark, we group test cases by graph diameter and plot RMSE-all versus diameter in Figure 3(b). Errors for the baseline models escalate rapidly with increasing diameter, whereas M4GN's error increases only slightly, showing sustained accuracy on wide meshes with long-range interactions. This robustness comes from M4GN's hybrid segmentation, which clusters nodes that share modal behavior, and its segment-level transformer, which achieves global information exchange in one hop among segments; together, these mechanisms mitigate over-smoothing while preserving local detail, sustaining performance as graph diameter grows. Moreover, we perform generalization tests to show whether a surrogate trained on modest meshes stays reliable on larger domains. According to Figure 3(c), all four surrogates deteriorate when directly generalize to larger mesh domain. The modest degradation of MGN confirms that a flat model can generalize spatially if the growth factor is moderate, but only at the expense of longer inference times and an elevated risk of over-smoothing as graph diameter continues to rise. Hierarchical methods trim that cost, yet their performance hinges on how they coarsen the mesh and whether cross-level communication remains efficient. Among them, M4GN's smaller generalization gap indicates that its hybrid segmentation and permutation-invariant aggregation alleviate—but do not

eliminate—this sensitivity, suggesting future work on adaptive segment counts or depth-aware micro-level passes when extrapolating to substantially larger meshes.

## 4.3 Additional Studies

We conducted additional studies to comprehensively evaluate model performance, hyperparameter selection, and the impact of key architectural designs, with detailed results and discussions provided in the appendices. Results on additional datasets are reported in Appendix C.3, and the segmentation-effectiveness study appears in Appendix C.4. A systematic sensitivity study, detailing how each hyperparameter affects performance and offering practical tuning guidelines, can be found in Appendix D. Additionally, a comprehensive analysis of generalization performance is provided in Appendix E, and further insights into computational efficiency are included in Appendix F.

# 5 Related Works

## 5.1 Hierarchical GNN Models for Dynamical System Simulation

The application of Graph Neural Networks (GNN) for dynamic system prediction is an emerging research area in scientific machine learning due to their versatility and effectiveness (Chang et al., 2016; Li et al., 2018b; Belbute-Peres et al., 2020; Hu et al., 2023). Unlike image-based learning methods such as Convolutional Neural Networks (CNNs) (Um et al., 2018; Ummenhofer et al., 2019), GNNs can directly handle unstructured simulation meshes, making them well-suited for simulating systems with complex domain boundaries while ensuring spatial invariance and locality Wu et al. (2020). A notable milestone in this field is MeshGraphNets (Pfaff et al., 2020), which enables the general scheme for learning mesh-based dynamical simulations. To mitigate the over-smoothing issue (Li et al., 2018a) that typically occurs in GNNs when applied to large or complex datasets with long-range interactions, several hierarchical models have been introduced recently. For instance, GMR-GMUS (Han et al., 2022) utilizes a pooling method to select pivotal nodes through uniform sampling. Similarly, the EAGLE (Janny et al., 2023) employs a clustering-based pooling method along with a transformer mechanism, showing promising performance in fluid dynamics. MS-MGN (Fortunato et al., 2022) proposes a dual-layer framework that passes messages at both fine and coarse resolutions for mesh-based simulation learning. BSMS-GNN (Cao et al., 2023) analyzes limitations of existing pooling strategies and introduces a bi-stride pooling method using breadth-first search (BFS) to select nodes. (Yu et al., 2023) proposes a similar hierarchical structure as (Cao et al., 2023) but with two different transformers to enable long-range interactions.

## 5.2 Datasets for Unstructured Mesh-based Simulations

As a cornerstone in the field, MeshGraphNets (Pfaff et al., 2020) introduces a collection of datasets, showcasing the versatility of graph-based surrogates in various problems involving unstructured mesh simulations. Moreover, EAGLE (Janny et al., 2023) presents a fluid dynamics dataset capturing unsteady and turbulent airflows; BSMS-GNN (Cao et al., 2023) provides the InflatingFont dataset, which focuses on the quasi-static inflation of enclosed elastic surfaces. Despite recent progress, existing public datasets still leave two critical gaps for assessing hierarchical surrogates: (1) Many benchmarks contain thousands of vertices, yet their graph diameter—the maximum shortest-path length between any two mesh nodes—remains small, which under-exercises long-range message passing. (2) Most datasets expose only one mesh density or size band, preventing systematic studies of how a surrogate trained on small problems generalizes to larger ones. To bridge these gaps, we contribute *DeformingBeam* together with its enlarged variant *DeformingBeam-Large*. This pair constitutes the first public 3-D Lagrangian contact-deformation benchmark that also provides a scale-up case. The beam's slender geometry yields graph diameters several times longer than existing solid mechanics datasets, exposing long-range interactions. Such a dataset enables rigorous testing of hierarchical surrogates and serves as a benchmark for probing model scalability and cross-scale generalization.

# 6    Conclusion

In this work we addressed two challenges in hierarchical mesh-GNN surrogates—physics-faithful graph coarsening and the accuracy–efficiency imbalance—by introducing M4GN, a three-tier, segment-centric framework built on two key innovations: a physics-aware, hybrid segmentation strategy that yields contiguous, dynamically coherent segments, and a permutation-invariant and computational efficient mesh segment aggregator. We systematically compared M4GN with leading baselines across multiple benchmarks using both traditional error metrics and proposed mesh-quality metrics, and we quantified segmentation quality through three different intra- and inter-segment scores to illuminate how our hybrid segmentation shapes downstream accuracy. Additional studies assessed the accuracy-versus-speed trade-off and robustness across graph diameter and scale. Across all tests, M4GN achieved the strongest overall balance of accuracy, efficiency, and mesh fidelity, exhibited the smallest error growth on wide-diameter graphs, and maintained the smallest generalization gap among all hierarchical models. Despite these advances, M4GN still exhibits limitations that need further investigation. For example, the method currently imposes no hard constraints on contact boundaries and it offers no formal guarantees of physical consistency across segment interfaces. In addition, selecting segmentation hyper-parameters still requires modest empirical tuning despite the provided guidelines. In summary, the findings in this paper underscore the value of coupling principled segmentation with balanced hierarchical reasoning for scalable, high-fidelity mesh simulation. Future research will focus on incorporating constraint enforcement, automating hyper-parameter selection, and extending the framework to coupled multi-physics scenarios.

### Broader Impact Statement

This paper presents work whose goal is to advance the field of Machine Learning for Physics, Surrogate Modeling, and Dynamical System Simulation. There are many potential societal consequences of our work, none of which we feel must be specifically highlighted here.

### Acknowledgments

This work was performed under the auspices of the U.S. Department of Energy by the Lawrence Livermore National Laboratory under Contract DE-AC52-07NA27344. LLNL release number: LLNL-JRNL-2006212.

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

## Appendix: Table of Contents

# A  Datasets

## A.1  Datasets Details

**CylinderFlow** – This public dataset includes simulations of transient incompressible flow around a cylinder, with varying diameters and locations, on a fixed 2D Eulerian mesh. In all fluid domains, the node type distinguishes fluid nodes, wall nodes, and inflow/outflow boundary nodes. The inlet boundary conditions are given by a prescribed parabolic profile, $u_{in} = u_0[1 - 4(y/H)]$ where $u_0$ and H are the centerline velocity and the distance between the sidewalls, respectively. The dataset contains 1000 training simulations, 100 validation simulations, and 100 test simulations.

**EAGLE**[*] – This public dataset contains simulations of complex airflow generated by a 2D unmanned aerial vehicle maneuvering in 2D scenes with varying floor profiles. While the scene geometry varies, the UAV trajectory is constant: the UAV starts in the center of the scene and navigates, hovering near the floor surface. The node types are fluid nodes, wall nodes, and aerial vehicle nodes. The dataset contains 948 training simulations, 118 validation simulations, and 118 test simulations.

**FlagSimple**[*] – This public dataset includes simulations of a flag blowing in the wind and flag direction, with variation in wind speed. The mesh is static and remains the same for all simulations. Node types are flag nodes and handle nodes that are fixed. This dataset contains 1000 training simulations, 100 validation simulations, and 100 test simulations.

**DeformingPlate** – This public dataset includes simulations of hyperelastic plates deformed by a moving obstacle, with variations in plate design and obstacle design. The node types are plate nodes, handle nodes that are fixed, and obstacle nodes. This dataset contains 1000 training simulations, 100 validation simulations, and 100 test simulations.

**DeformingBeam** – This dataset is generated using *solids4foam* which is a toolbox for performing solid mechanics and fluid-solid interaction simulations in OpenFOAM and foam-extend. A nearly incompressible neo-Hookean model is used where the material properties are density $\rho_0 = 1000$ kg/m$^3$, Young's modulus $E$ = 1 MPa, and Poisson's ratio $\nu = 0.4$. The beam comes in different geometries with various initial conditions and boundary conditions. The node types are plate nodes, handle nodes that are fixed, and obstacle nodes. This dataset contains 355 training simulations, 40 validation simulations, and 60 test simulations.

**DeformingBeam (large)** – A large domain DeformingBeam dataset is created for generalization studies. The physical domain size is doubled. The size of the mesh cell is kept consistent with the regular DeformingBeam dataset. This generalization dataset contains 112 simulations.

Table 4: Detailed information for each dataset.

| System | Dataset | Avg. # Nodes | # Steps | Mesh Type | Graph Diameter | Node Feature | Edge Feature | Output |
|---|---|---|---|---|---|---|---|---|
| Fluid | CylinderFlow | 1885 | 600 | Triangle, Eulerian, 2D | 11 | $\mathbf{v}_i, \mathbf{n}_i$ | $\mathbf{m}_{ij}, \lvert\mathbf{m}_{ij}\rvert$ | $\dot{\mathbf{v}}_i, p_i$ |
| | EAGLE* | 3390 | 990 | Triangle, Eulerian, 2D | 29.5 ± 1.7 | $\mathbf{v}_i, \mathbf{p}_i, \mathbf{n}_i$ | $\mathbf{m}_{ij}, \lvert\mathbf{m}_{ij}\rvert$ | $\dot{\mathbf{v}}_i, \dot{\mathbf{p}}_i$ |
| Flow-Structure | FlagSimple* | 1579 | 400 | Triangle, Lagrangian, 3D | 41 | $\mathbf{x}_i, \dot{\mathbf{x}}_i, \mathbf{n}_i$ | $\mathbf{m}_{ij}, \lvert\mathbf{m}_{ij}\rvert$ | $\ddot{\mathbf{x}}_i$ |
| Solid | DeformingPlate | 1271 | 400 | Tetrahedron, Lagrangian, 3D | 16.9 ± 5.8 | $\mathbf{x}_i, \dot{\mathbf{x}}_{\text{OBS}}, \mathbf{n}_i$ | $\mathbf{x}_{ij}, \lvert\mathbf{x}_{ij}\rvert, \mathbf{m}_{ij}, \lvert\mathbf{m}_{ij}\rvert$ | $\dot{\mathbf{x}}_i$ |
| | DeformingBeam | 1542 | 400 | Prism, Lagrangian, 3D | 41.3 ± 11.8 | $\mathbf{x}_i, \dot{\mathbf{x}}_{\text{OBS}}, \mathbf{n}_i$ | $\mathbf{x}_{ij}, \lvert\mathbf{x}_{ij}\rvert, \mathbf{m}_{ij}, \lvert\mathbf{m}_{ij}\rvert$ | $\dot{\mathbf{x}}_i$ |
| | DeformingBeam (large) | 4540 | 400 | Prism, Lagrangian, 3D | 82.1 ± 23.0 | $\mathbf{x}_i, \dot{\mathbf{x}}_{\text{OBS}}, \mathbf{n}_i$ | $\mathbf{x}_{ij}, \lvert\mathbf{x}_{ij}\rvert, \mathbf{m}_{ij}, \lvert\mathbf{m}_{ij}\rvert$ | $\dot{\mathbf{x}}_i$ |

# B  Model Details

## B.1  M4GN Configurations

The GNN part of M4GN adopts the encoder and graph processor in the MGN model (Pfaff et al., 2020). The basic building block is Multi-Layer Perceptron (MLP). The MLP has 3 layers, a hidden dimension of 128, ReLU activation, and a single layer of Layer Normalization at the end. The node encoder and

---

[*]EAGLE and FlagSimple are included only as supplementary benchmarks to show robustness across different dataset types and support additional analysis; we exclude ablation studies on these datasets to maintain a focused experimental scope and ensure clarity in result interpretation.

edge encoder(s) are 3-layer MLPs. By default, the M4GN has 7 message passing steps in the GNN. The mesh segment transformer consists of 4 self-attention layers, each with 8 heads. The output decoder is a 3-layer MLP without Layer Normalization. For DeformingPlate and DeformingBeam, M4GN only considers world edges between contacting mesh objects. The world edge radius is set to 0.01 for DeformingPlate and 0.002 for DeformingBeam. As both the DeformingBeam and DeformingPlate datasets feature a rigid object with quasi-static motion, we use only the first mode from our modal decomposition, which sufficiently captures the largest-scale deformation pattern. For the CylinderFlow dataset, we employ 6 modes, determined by an energy threshold criterion, ensuring a more comprehensive representation of the flow's multi-scale dynamics. For Table 3, every dataset uses SLIC-MDODe segmentation, but the hyper-parameters differ: For the CylinderFlow dataset, 36 segments are used with compactness value $\tau = 1.0$, segment overlap, and positional encoding (PE) are both enabled. For DeformingPlate and DeformingBeam, 19 segments are used with the former using $\tau = 1.0$ with no overlap or PE, while the latter uses $\tau = 0.5$ with overlap but without PE. Ablation studies are also conducted based on these configurations.

## B.2    Baselines

**GCN** – The GCN model consists of 15 GCN layers with a hidden dimension of 128. The GCN model does not have edge input. Node input includes mesh position $\mathbf{x}_i$ for CylinderFlow. The implementation is from PyTorch Geometric.

**g-U-Net** – The g-U-Net model is a modified version from PyTorch Geometric. Instead of GCN layers, it is built using the GNN layers similar to MGN. The level of scale is 7 for CylinderFlow, 6 for DeformingPlate, and 4 for DeformingBeam.

**MGN** – Our implementation of MGN follows the one described in Pfaff et al. (2020). The processor of MGN contains 15 MP steps. World edges are constructed as specified in the paper, with a world edge radius of 0.03 for DeformingPlate and 0.003 for DeformingBeam.

**BSMS-GNN** – We followed the BSMS-GNN implementation Cao et al. (2023) from `https://github.com/Eydcao/BSMS-GNN`. We introduced a modification to the original code by incorporating output normalization, which we observed to enhance the model's performance. For CylinderFlow and DeformingPlate, we used the same number of multi-scale levels as specified in the BSMS-GNN paper, at 7 and 6 levels, respectively. The number of multi-scale levels for DeformingBeam is set at 4 as an optimal configuration.

**EAGLE** – The implementation of EAGLE follows the paper Janny et al. (2023) and the code repository `https://github.com/eagle-dataset/EagleMeshTransformer`. We set the number of nodes per cluster at 20, which offers a balanced performance and efficiency according to the paper. This results in 94 clusters for CylinderFlow, 64 for DeformingPlate, and 38 for DeformingBeam. In addition, we add contacting world edges in the EAGLE implementation for DeformingPlate and DeformingBeam to improve performance. The world edges are added the same as in M4GN.

## B.3    Training Details

During training, random Gaussian noise is added to the spatial node inputs, as described in Pfaff et al. (2020). For CylinderFlow, all models use a noise scale of 0.02. For DeformingPlate, all models use a noise scale of 0.003. For DeformingBeam, EAGLE and M4GN use a noise scale of 1e-4 and other models use a noise scale of 1e-3.

For GCN, g-U-Net, MGN, EAGLE and M4GN, we adopt the same training scheme: For CylinderFlow and DeformingPlate, we trained the model for 2M steps. The learning rate starts at 1e-4 and exponentially decays to 1e-6 from 1M to 2M steps. For DeformingBeam, we trained the model for 1M steps. The learning rate starts at 1e-4 and exponentially decays to 1e-6 from 500K to 1M steps.

For BSMS-GNN, we adopt the training scheme from the original implementation. Models for CylinderFlow and DeformingPlate were trained for 50 epochs, corresponding to 3.75M and 3M training steps, respectively. DeformingBeam model was trained for 100 epochs, corresponding to 1.775M training steps.

Across all models and datasets, we use a batch size of 8. Experiments were conducted using PyTorch distributed training over two Nvidia Tesla P100 GPUs.

### B.4 Hybrid Mesh Segmentation Details

In Figure 4, several cases are selected from each dataset to illustrate the difference of each mesh graph segmentation method. It's worth noting that the graph will be partitioned only once during the training and testing phase for each simulation, and this partitioning will remain consistent across all time steps. This is because the segmentation is based solely on the system's properties and initial conditions prior to the start of the simulation.

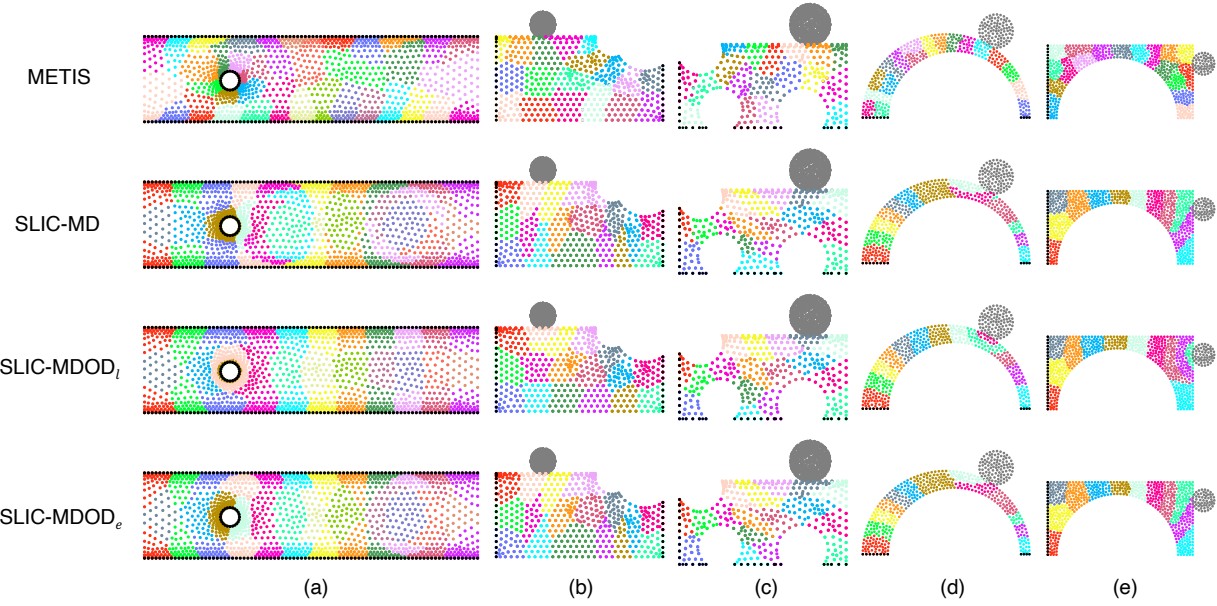

Figure 4: Illustration of different segmentation methods under various cases: (a):CylinderFlow; (b)(c): DeformingPlate; (d)(e): DeformingBeam. Mesh nodes are colored based on segment id and all boundary nodes are colored in black.

The pseudo code of the hybrid segmentation module proposed in this work can be found in Algorithm 2. Here, METIS (Karypis & Kumar, 1998) is a graph partitioning technique that efficiently divides meshes into approximately equal-sized partitions. It leverages multilevel partitioning algorithms to minimize the edge-cut or communication costs between the resulting partitions. We employ METIS due to its versatility in creating a user-specified number of equal-sized mesh segments. SLIC (Achanta et al., 2012) is a clustering algorithm employed for partitioning data. In our approach, we adapt SLIC to segment the mesh based on physics-informed features. These features could guide SLIC to create a segmentation that captures the underlying physics of the system. The consequent mesh segments can potentially enable efficient macro-level information exchange tailored to the system's dynamics. Concretely, for each node $i$, we incorporate physics-aware feature $f_i^{md}$ derived from modal decomposition. Additionally, we augment these features by concatenating a measure of the shortest distance to obstacle nodes $d_i^{obs}$. To ensure that this measure dominates when $d_i^{obs}$ is small, we apply either an exponential or logarithmic transformation, defined as:

$$f_{\exp}(d) = \exp(-d), \quad f_{\log}(d) = \log(d). \tag{9}$$

Depending on the selection of features and the transformation function, we design 6 variants of SLIC:

- SLIC-OD: $f_i = d_i^{obs}$
- SLIC-OD$_l$: $f_i = f_{\log}(d_i^{obs})$

---

**Algorithm 1:** Modal Decomposition

---

1: Case Type: `solid` or `fluid`
2: **Input:** Finite element mesh, boundary conditions, material properties for solid (e.g. $E, \nu, \rho$), number of modes $m$
3: **Build Finite Element Basis:**
4:     Define shape functions on each element using the node connectivity
5:     Enumerate degrees of freedom (DOFs) for each node/component
6: **if** Case Type = `solid` **then**
7:     **Structural Modal Analysis**
8:     Assemble stiffness matrix $\mathbf{K}$ (using elasticity)
9:     Assemble mass matrix $\mathbf{M}$ (using density)
10:     Apply boundary conditions to eliminate fixed DOFs
11:     Solve $\mathbf{K}\boldsymbol{\phi} = \lambda \mathbf{M}\boldsymbol{\phi}$ for the first $m$ modes
12:     **Output:** Eigenpairs $\{(\lambda_i, \boldsymbol{\phi}_i)\}_{i=1}^m$ (*structural modes*)
13: **else if** Case Type = `fluid` **then**
14:     **Laplacian Eigenfunctions**
15:     Assemble Laplacian matrix
16:     Assemble $L^2$-type matrix
17:     Apply Dirichlet constraints on boundary nodes
18:     Solve $-\nabla^2\phi = \lambda\,\phi$ for the first $m$ modes
19:     **Output:** Eigenpairs $\{(\lambda_i, \phi_i)\}_{i=1}^m$ (*harmonic modes*)
20: **end if**
21: **Return:** $m$-dimensional feature vector $f_i^{md} = (\phi_1(i), \phi_1(i), \dots \phi_m(i))$ at each mesh node $i$

---

- SLIC-OD$_e$: $f_i = f_{\exp}(d_i^{obs})$

- SLIC-MD: $f_i = f_i^{md}$

- SLIC-MDOD$_l$: $f_i = \left[f_{\log}(d_i^{obs}), f_i^{md}\right]^T$

- SLIC-MDOD$_e$: $f_i = \left[f_{\exp}(d_i^{obs}), f_i^{md}\right]^T$

After we have the physics-aware feature, we can apply the SLIC algorithm to get the mesh node segments.

### B.5 Mesh Segment Hyperparameter Selection

The compactness parameter $\tau$ in the SLIC algorithm controls the trade-off between physics-guided feature similarity and spatial proximity. Our goal is to choose $\tau$ such that the resulting segmentation captures both underlying physical patterns and spatial coherence (i.e., grouping nodes that are close to each other). For CylinderFlow and DeformingPlate, we set $\tau = 1.0$, which provides a balanced segmentation. For DeformingBeam, we set a lower $\tau$ at 0.5 to promote a better alignment with physical features. The cluster size $S$ is determined such that the domain area satisfies: Domain Area = $KS^2$, where $K$ is the number of segments and the domain area is given by $(x_{\max} - x_{\min})(y_{\max} - y_{\min})$. The average cluster size $S$ for CylinderFlow, DeformingPlate and DeformingBeam is set to be $\sqrt{0.656/K}$, $\sqrt{0.125/K}$ and $\sqrt{0.005/K}$, respectively.

Systematically choosing the optimal number of segments $K$ requires both domain insight and practical experimentation. In our experience, two main factors drive the choice of $K$: (1) the total mesh size ($N$) and (2) local variations in mesh density. For instance, CylinderFlow is particularly dense near boundaries, which benefits from a larger $K$, whereas DeformingBeam/DeformingPlate have more uniformly distributed nodes, so a smaller $K$ can suffice.

To make this selection concrete, we typically perform a short hyperparameter sweep over a small set of candidate values for $K$. A simple heuristic is to pick $K$ values on a roughly geometric or linear scale, for instance: $K \in \{\sqrt{N}/2, \sqrt{N}, 2\sqrt{N}, ....\}$ up to a point where adding more segments no longer improves

---

**Algorithm 2:** Hybrid Mesh Segmentation

---

1: **Input:**
2:     Initial mesh graph $G = (\mathcal{V}, \mathcal{E})$
3:     Perform modal decomposition and compute mesh node feature $f_i$
4:     Number of segments $K$, compactness parameter $\tau$, average cluster size $S$
5: **Output:** Mesh node segmentation $\{\mathcal{V}_{S_k}\}_{k=1}^{K}$

---

$\Rightarrow$ *Graph-based Mesh Segment Initialization*
6: **Coarsening Phase:**
7: $G_{\text{coarse}} \leftarrow G$
8: **while** size of $G_{\text{coarse}}$ is larger than threshold **do**
9:     Combine pairs of connected nodes in $G_{\text{coarse}}$ to form a coarser graph
10:     $G_{\text{coarse}} \leftarrow$ coarsened graph
11: **end while**
12: **Initial Partitioning:**
13: Partition $G_{\text{coarse}}$ into $K$ segments using a standard partitioning method (e.g., spectral partitioning)
14: **Uncoarsening and Refinement Phase:**
15: **while** $G_{\text{coarse}} \neq G$ **do**
16:     Expand $G_{\text{coarse}}$ to the next finer graph $G_{\text{fine}}$
17:     Project partitions onto $G_{\text{fine}}$
18:     Refine the partitioning on $G_{\text{fine}}$ by iteratively moving a vertex to neighboring segments **iff** the move
        lowers the total number (or weight) of edges that cross between segments, provided each segment
        remains roughly balanced in size.
19:     $G_{\text{coarse}} \leftarrow G_{\text{fine}}$
20: **end while**
21: Obtain initial clusters $\{\mathcal{V}_{S_k}\}_{k=1}^{K}$ from the final partitioning, which will be updated next

---

$\Rightarrow$ *Superpixel-based Mesh Segment Refinement*
22: **repeat**
23:     **for** each mesh segment centroid $C_k$ **do**
24:         Update $C_k$ by averaging over all mesh nodes assigned to it:

$$C_k = [x_{C_k}, f_{C_k}]^T = \frac{1}{|\mathcal{V}_{S_k}|} \sum_{i \in \mathcal{V}_{S_k}} [x_i, f_i]^T$$

        where $\mathcal{V}_{S_k}$ is the set of mesh nodes assigned to segment $S_k$
25:     **end for**
26:     **for** each mesh node $i \in V$ **do**
27:         Compute the distance measure $d(i, C_k)$ to each cluster center $C_k$ using:

$$d(i, C_k) = \|f_i - f_{C_k}\| + \tau \|x_i - x_{C_k}\|$$

        where $x_i$ and $x_{C_k}$ are the spatial coordinates, $f_i$ and $f_{C_k}$ are the physics-guided features.
28:         Assign mesh node $i$ to the nearest segment centroid $C_k$ if $d(i, C_k) \leq S$
29:     **end for**
30: **until** convergence or a maximum number of iterations is reached

---

validation metrics (e.g., prediction accuracy, mesh quality). In practice, testing each candidate $K$ on a subset (e.g., 10%) of the training data is typically enough to identify a near-optimal configuration, and then we finalize training with that $K$ on the full dataset. This strategy is computationally manageable and provides a principled way to tailor $K$ to new domains.

## C  Additional Evaluation Metrics and Results

### C.1  Metrics for Mesh Quality Measure

**Hausdorff Distance** – The Hausdorff Distance measures how well the mesh with the predicted node positions conforms to the system's true geometry. It is defined as:

$$\text{GF}_h(\mathcal{V}, \hat{\mathcal{V}}) = \max\Big\{ h(\mathcal{V}, \hat{\mathcal{V}}), h(\hat{\mathcal{V}}, \mathcal{V}) \Big\}, \tag{10}$$

where $h(\mathcal{V}, \hat{\mathcal{V}}) = \sup_{\mathbf{x} \in \mathcal{V}} \inf_{\hat{\mathbf{x}} \in \hat{\mathcal{V}}} \|\mathbf{x} - \hat{\mathbf{x}}\|$ is the directed Hausdorff distance (Huttenlocher et al., 1993) from the ground-truth node set $\mathcal{V}$ to the predicted node set $\hat{\mathcal{V}}$.

**Chamfer Distance** – The Chamfer Distance (Wu et al., 2021) measures the average distance between points on the predicted mesh and the true mesh, providing a balanced assessment of $\text{GF}_h$. Unlike the Hausdorff Distance, which focuses on the maximum deviation, the Chamfer Distance is sensitive to the overall distribution of errors across the mesh surfaces. As both Chamfer and Hausdorff distance are measures for GF, we name them as $\text{GF}_c$ and $\text{GF}_h$ for simplicity, respectively. The Chamfer distance is mathematically defined as:

$$\text{GF}_c(\mathcal{V}, \hat{\mathcal{V}}) = \frac{1}{|\mathcal{V}|} \sum_{\mathbf{x} \in \mathcal{V}} \min_{\hat{\mathbf{x}} \in \hat{\mathcal{V}}} \|\mathbf{x} - \hat{\mathbf{x}}\|^2 + \frac{1}{|\hat{\mathcal{V}}|} \sum_{\hat{\mathbf{x}} \in \hat{\mathcal{V}}} \min_{\mathbf{x} \in \mathcal{V}} \|\hat{\mathbf{x}} - \mathbf{x}\|^2, \tag{11}$$

where $\mathcal{V}$ and $\hat{\mathcal{V}}$ are the set of vertices in the ground-truth and predict mesh, respectively. $|\mathcal{V}|$ and $|\hat{\mathcal{V}}|$ denote the number of vertices in each mesh.

**Mesh Continuity** – Mesh Continuity (Knupp, 2007) evaluates the uniformity of predicted mesh cell sizes to ensure stability and is defined as

$$\text{MC} = \frac{1}{C} \sum_{i=1}^{C} \frac{\max_{c_j \in \text{Adj}(c_i)} V(c_j)}{\min_{c_j \in \text{Adj}(c_i)} V(c_j)}, \tag{12}$$

where $\text{Adj}(c_i)$ is the neighboring cells of cell $c_i$, and $V(c_i)$ calculates the volumetric area for $c_i$.

**Aspect Ratio (error)** – The Aspect Ratio (Zienkiewicz & Taylor, 2005) assesses the shape quality of individual 2D or 3D mesh elements and is widely used in finite element method (FEM) literature to evaluate how closely each element approaches the ideal shape, such as an equilateral triangle or a regular tetrahedron. For example, for triangular meshes, the aspect ratio is defined as $\frac{L_{\max}}{2\sqrt{\sqrt{3}A}}$, where $L_{\max}$ is the longest edge length, $A$ is the area of the triangle. For tetrahedra mesh, it is defined as $\frac{\sqrt{6}L_{\max}}{V^{1/3}}$, where $V$ the volume of the tetrahedron. High aspect ratios indicate elongated or distorted elements, which can cause numerical instability and reduce simulation accuracy. By analyzing the aspect ratios across all elements, we can assess the overall uniformity and regularity of the mesh. To evaluate the accuracy of the predicted mesh compared to the ground truth, we calculate the aspect ratio for both the predicted and actual meshes. The Aspect Ratio Error is then determined as the $L_1$ distance between these two values. This error metric quantifies the deviation in shape quality between the predicted and true meshes, providing a direct measure of how well the prediction preserves the ideal element shapes. Incorporating the Aspect Ratio Error allows for a more precise evaluation of mesh quality and prediction accuracy, ensuring that the segmented meshes maintain the necessary geometric properties for reliable simulations.

### C.2  Segmentation Quality Metrics

In order to rigorously evaluate the quality of our hybrid mesh segmentation and its impact on the prediction of system dynamics, it is essential to consider metrics that assess both inter-segment and intra-segment

characteristics. We introduce three such metrics — *Conductance*, *Edge Cut Ratio*, and *Silhouette Score* — which provide a comprehensive assessment of segmentation quality by quantifying the cohesion within segments and the separation between segments. The necessity of these metrics arises from the need to ensure that segments are well-separated, minimizing unnecessary interactions between dissimilar regions (inter-segment quality), and that nodes within the same segment share similar properties or behaviors (intra-segment quality).

Moreover, in our hierarchical model architecture, the intra-segment quality pertains to the micro-level information exchange stage. High intra-segment quality facilitates accurate modeling of local dynamics within each segment by ensuring that nodes are cohesive and share similar dynamic behaviors. Conversely, the inter-segment quality directly relates to the macro-level information exchange stage. High inter-segment quality ensures efficient communication between segments by reducing redundant or irrelevant interactions, which is crucial for capturing global dynamics across the entire mesh. Below are the details of three metrics to measure segmentation quality.

**Conductance** – Conductance measures the fraction of total edge connections that cross between different segments relative to the total connections of the segments. It assesses how well the segmentation minimizes inter-segment connections while maintaining intra-segment cohesion. Let $G = (\mathcal{V}, \mathcal{E})$ as an undirected graph representing the mesh, where $\mathcal{V}$ is the set of nodes and $\mathcal{E}$ is the set of edges. Let $S$ be a segment and $\bar{S} = G \setminus S$ be its complement. The conductance of segment $S$ is defined as:

$$\text{Conductance} = \frac{\left|\{(u,v) \in \mathcal{E} \mid u \in S, \ v \in \bar{S}\}\right|}{\min\left(\text{vol}(S), \ \text{vol}(\bar{S})\right)}, \tag{13}$$

where the numerator is the number of edges crossing between $S$ and $\bar{S}$. The volumn of segment $S$ is given by $\text{vol}(S) = \sum_{u \in S} \deg(u)$, where $\deg u$ is the degree of node $u$ (the number of edges connected to $u$).

**Edge Cut Ratio** – The Edge Cut Ratio quantifies the proportion of edges that are cut by the segmentation relative to the total number of edges in the mesh. It is defined as:

$$\text{Edge Cut Ratio} = \frac{\left|\{(u,v) \in \mathcal{E} \mid \text{Seg}(u) \neq \text{Seg}(v)\}\right|}{E}, \tag{14}$$

where the denominator is the number of edges that connect nodes in different segment. $\text{Seg}(u)$ denotes the segment to which node $u$ belongs and $E = |\mathcal{E}|$ is the total number of edges.

**Silhouette Score** – For each node $i$, the Silhouette Score evaluates how similar $i$ is to nodes in its own segment compared to nodes in other segments. It is defined as:

$$\text{Silhouette Score} = \frac{1}{N} \sum_{u=1}^{N} \frac{b(i) - a(i)}{\max\{a(i), \ b(i)\}}, \tag{15}$$

where $N$ is the total number of nodes, $a(i)$ is the average dissimilarity of node $i$ with all other nodes in the same segment and $b(i)$ is the lowest average dissimilarity of node $i$ to any other segment to which $i$ does not belong. To be more specific $a(i) = \frac{1}{|S_i|-1} \sum_{\substack{j \in S_i \\ j \neq i}} d(i,j)$, $b(i) = \min_{S' \neq S_i} \left(\frac{1}{|S'|} \sum_{j \in S'} d(i,j)\right)$, where $S_i$ is the segment containing node $i$ and $d(i,j)$ can be any appropriate distance metric, such as Euclidean distance based on node features or positions.

By combining these metrics, we achieve a comprehensive evaluation of segmentation quality that covers both the internal cohesion of segments and their external separation. Having these metrics, along with prediction result metrics, can better help us understand the effect of segmentation on the predicted system dynamics. These metrics can be used to help finding better physics-aware segment features and determining the optimal segmentation number (results and discussion in Appendix D.5).

### C.3 Evaluation on Supplementary Datasets

To evaluate the robustness of our approach across distinct physical regimes, we retrained all models on two additional datasets—*EAGLE* and *FlagSimple*—and compared the proposed M4GN architecture with the

Table 5: Comparison of results with state-of-the-art methods across two additional datasets, where each model is trained independently for each dataset. Prediction accuracy is evaluated using Root Mean Square Error (RMSE), with the output being the 2D velocity and pressure field for EAGLE and the 3D position for FlagSimple. Results are averaged over three experiments with different random seeds and presented as mean and standard deviation.

| | | Mesh Quality Metrics ↓ | | | | Prediction and Computation Metrics ↓ | | |
|---|---|---|---|---|---|---|---|---|
| Dataset | Model | $GF_h$ $(\times 10^{-2})$ | $GF_c$ $(\times 10^{-5})$ | MC $(\times 10^{-2})$ | AR $(\times 10^{-2})$ | RMSE-all | Train Memory [MB] | Test Time per step [ms] |
| EAGLE | MGN | - | - | - | - | $4.13 \pm 0.05$ | 10525 | 35.8 |
| | EAGLE | - | - | - | - | $4.24 \pm 0.06$ | 7254 | 35.2 |
| | M4GN (Ours) | - | - | - | - | $\mathbf{3.95 \pm 0.05}$ | **5308** | **28.4** |
| Flag Simple | MGN | $1.82 \pm 0.08$ | $5.01 \pm 0.34$ | $6.02 \pm 0.59$ | $4.16 \pm 0.65$ | $0.25 \pm 0.11$ | 1060 | 36.5 |
| | EAGLE | $1.73 \pm 0.08$ | $5.22 \pm 0.50$ | $6.71 \pm 0.55$ | $5.49 \pm 1.05$ | $1.01 \pm 1.14$ | 1336 | 41.9 |
| | M4GN (Ours) | $\mathbf{0.98 \pm 0.05}$ | $\mathbf{2.23 \pm 0.19}$ | $\mathbf{4.06 \pm 0.40}$ | $\mathbf{3.11 \pm 0.62}$ | $\mathbf{0.15 \pm 0.01}$ | **549** | **30.6** |

Table 6: Transfer-segmentation experiment on the *DeformingBeam* dataset. Replacing EAGLE's native segmentation with the proposed hybrid M4GN segmentation (second row) isolates the effect of segmentation quality, while the full M4GN row (third row) shows the additional benefit of our modified hierarchical architecture. Lower values indicate better performance.

| | | | Mesh Quality Metrics ↓ | | | | Prediction Error Metrics ↓ | | |
|---|---|---|---|---|---|---|---|---|---|
| Dataset | Model | Segmentation | $GF_h$ $(\times 10^{-3})$ | $GF_c$ $(\times 10^{-6})$ | MC $(\times 10^{-3})$ | AR $(\times 10^{-3})$ | RMSE-1 $(\times 10^{-5})$ | RMSE-50 $(\times 10^{-4})$ | RMSE-all $(\times 10^{-4})$ |
| Deforming Beam | EAGLE | EAGLE | $0.64 \pm 0.04$ | $0.17 \pm 0.01$ | $5.98 \pm 0.43$ | $5.17 \pm 0.37$ | $1.51 \pm 0.04$ | $0.67 \pm 0.12$ | $4.22 \pm 0.30$ |
| | EAGLE | M4GN | $0.46 \pm 0.02$ | $0.11 \pm 0.00$ | $5.43 \pm 0.09$ | $4.02 \pm 0.10$ | $1.28 \pm 0.01$ | $0.57 \pm 0.04$ | $3.27 \pm 0.25$ |
| | M4GN | M4GN | $\mathbf{0.31 \pm 0.01}$ | $\mathbf{0.05 \pm 0.00}$ | $\mathbf{5.26 \pm 0.04}$ | $\mathbf{3.08 \pm 0.06}$ | $\mathbf{1.17 \pm 0.01}$ | $\mathbf{0.34 \pm 0.02}$ | $\mathbf{1.87 \pm 0.12}$ |

strongest baselines identified earlier, MGN and the task-specific EAGLE solver. As summarized in Table 5, M4GN attains the lowest prediction error on both datasets, lowering RMSE by 4% on EAGLE and by 40% on FlagSimple comparing to the second best model. For FlagSimple, M4GN also yields the most faithful meshes, reducing geometric-quality defect metrics by 25–57% relative to both baselines. Moreover, it halves peak training memory consumption and accelerates inference by up to 19%. Collectively, these results demonstrate that M4GN sustains its performance advantages on the supplementary datasets, delivering higher predictive accuracy, better mesh quality, and more efficient computation.

## C.4 Evaluation on the Effectiveness of Segmentation Algorithm

To isolate the impact of the proposed hybrid segmentation algorithm, we replace EAGLE's original segmentation with those generated by our method while holding all other settings unchanged. Table 6 shows a clear gain: mesh-quality metrics improve by 28%–35% and prediction errors fall by 15%–23%. When the same segmentation is paired with the full M4GN architecture, prediction errors are reduced even further and every mesh-quality metric reaches its best value. These results indicate that (i) the proposed hybrid mesh segmentation alone contributes a portion of the improvement, and (ii) the architectural changes in M4GN provide an additional, complementary boost. Hence both components—better segmentation and the modified model—are necessary for the overall performance gains.

## D Hyperparameter Sensitivity Analysis

This section presents a systematic sensitivity study, detailing how each hyperparameter affects performance and offering practical tuning guidelines. Importantly, the model already outperforms all baselines with default or minimally adjusted settings; additional tuning only refines an existing advantage rather than creating it. The following subsections analyze each hyperparameter in detail and Table 7 summarizes the key findings.

| Hyper-parameter | Ranges/ Variants | Observed impact on performance | Practical tuning guideline |
|---|---|---|---|
| Message–passing steps ($n_{\mathrm{MP}}$) | 1–8 | Few steps → under-reach, large RMSE and discontinuities; moderate steps → improve both RMSE and mesh quality; very high steps → over-smoothing and slight accuracy decay. | Start with 3-5 steps; increase while RMSE drops, stop when gains plateau or mesh quality stalls. |
| Hybrid segmentation flavour | METIS, SLIC variants | SLIC variants outperform METIS; physics-aware features give 7–35 % lower RMSE and better segment metrics; best overall: *SLIC-MDOD$_e$*. | Use *SLIC-MDOD$_e$* by default; avoid single-cue variants unless domain knowledge shows otherwise. |
| Number of segments ($N_{\mathrm{seg}}$) | 3–51 | Accuracy fairly stable; too few segments → coarse resolution, RMSE rise; too many segments → diminishing returns followed by possible degradation and added overhead. | Compute Silhouette over candidate $N_{\mathrm{seg}}$, then train 3–4 values near the peak; pick the point where RMSE/Chamfer stop improving. |
| Positional encoding (PE) | off / on | Improves RMSE when $N_{\mathrm{seg}}$ small and geometry simple; neutral or harmful when segments already fine or geometry complex. | Enable PE for coarse segmentations ($N_{\mathrm{seg}} < 15$) or 2-D flows; disable for high-resolution or highly deformable 3-D meshes. |
| Segment overlap ($\delta$) | $\delta = 0$ (none), $\delta = 1$ (one-ring) | Helps Eulerian or directional meshes at high $N_{\mathrm{seg}}$ (smoother transitions); can add redundancy and hurt Lagrangian cases. | Use $\delta = 1$ for high $N_{\mathrm{seg}}$ *and* the mesh is fixed; keep $\delta = 0$ for low $N_{\mathrm{seg}}$. |

Table 7: Summary of key hyperparameters, their tested ranges, performance effects, and tuning recommendations derived from the sensitivity study in Appendix D.

## D.1 Message Passing Steps in Micro-level Module

According to Figure 5, with fewer message passing steps, each node updates only based on immediate neighbors, resulting in higher prediction errors and mesh discontinuities. As more steps are introduced, nodes gather information from a broader neighborhood, leading to more accurate predictions and smoother mesh transitions. The early iterations of message passing yield the most noticeable improvements, as nodes rapidly gather useful information from their surrounding environment. Later iterations primarily serve to fine-tune the mesh continuity and reduce local errors, but the impact on overall accuracy diminishes. Interestingly, increasing the number of message-passing steps beyond a certain point continues to improve mesh quality, but prediction accuracy may degrade. This suggests the occurrence of oversmoothing, where the model excessively homogenizes node features, or overfitting, where the model starts to memorize local information rather than generalize. This phenomenon highlights the importance of carefully selecting the number of message-passing steps during the micro-level information exchange step to strike the right balance between improving prediction accuracy and maintaining mesh quality.

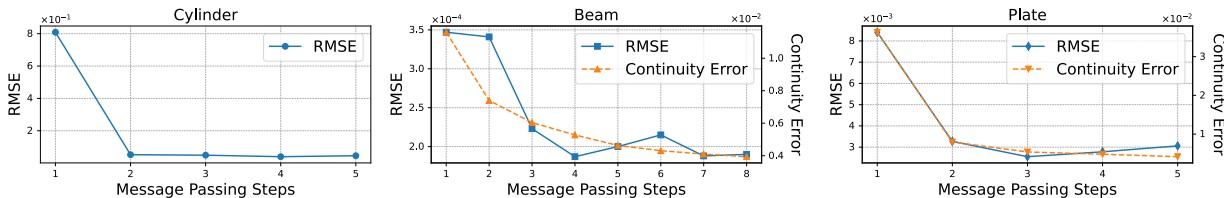

Figure 5: Ablation study on the impact of varying message-passing steps in the micro-level information exchange on prediction performance across three datasets.

## D.2 Variations of Hybrid Segmentation

As detailed in Appendix B.4, the hybrid segmentation admits six variants depending on the selected feature set and transformation. Table 8 summarizes the resulting model performance. Compared with METIS alone, SLIC-based variants lower $RMSE_{all}$ by 16 %, 27 %, and 7 % on *CylinderFlow*, *DeformingPlate*, and *DeformingBeam*, respectively, with SLIC-MDOD$_e$ emerging as the best choice across all three datasets. In terms of the impact of modal decomposition features in SLIC, for the two solid-mechanics cases (DeformingPlate and DeformingBeam), modal-decomposition features enrich the descriptor space with physics-relevant mode shapes, and every OD+MD variant—SLIC-MDOD$_l$ and SLIC-MDOD$e$—beats its OD-only counterpart across all metrics, delivering 24–35% lower RMSE and up to 25% better mesh-quality scores. These consistent gains confirm that modal information and boundary-distance cues are complementary for solid mechanics. On CylinderFlow, however, the flow is dominated by rapidly varying vortical patterns; the MD basis, derived purely from geometry in this Eulerian setting, adds little new information and can perturb the SLIC clusters, so SLIC-MD alone shows a slight RMSE rise. When MD is combined with the (exponentially weighted) distance cue in SLIC-MDOD$e$, the boundary-aware term stabilizes the segmentation while the modal vectors still provide complementary detail, giving a net improvement over distance or modal information used in isolation.

Moreover, refining the coarse METIS partitions with SLIC improves accuracy only when the added SLIC features better align local cuts with the true physics; otherwise, the refinement can fragment physically coherent regions and hurt performance. SLIC-OD, for example, uses only geometric distance to boundaries; on DeformingPlate this over-weights proximity and splits mode-consistent areas, so RMSE increases relative to the original METIS segmentation. Likewise, on DeformingBeam the distance-only (SLIC-OD) and modal-only (SLIC-MD) variants either ignore contact boundaries or long-range bending modes, producing finer—but less meaningful—segments and therefore higher error than METIS. Only the combined SLIC-MDOD$_e$, which couples modal information with an exponentially weighted distance term, strikes the right balance between global coherence and local adaptation.

To thoroughly evaluate the different segmentation methods, we utilize the three metrics -Conductance, Edge Cut Ratio, and Silhouette Score - introduced in Appendix C.2 to assess both inter-segment and intra-segment qualities of mesh partitions, providing a comprehensive understanding of each method's effectiveness. We then analyzed the correlation between these segmentation metrics and overall dynamic system performance, including mesh quality and prediction error, as illustrated in Figure 2 (a-c). Our findings indicate that segmentation methods incorporating physics-aware features, particularly those utilizing obstacle distances with exponential transformations, generally enhance model performance across various datasets. This improvement can be attributed to three key factors: (1) *Alignment with Dynamics*, where segmentation reflecting physical influences enables more effective learning of the system's dynamics; (2) *Enhanced Segment Quality*, achieved through improved intra-segment cohesion and minimized inter-segment interactions, facilitating better learning of localized patterns; and (3) *Benefit to Learning*, where emphasizing critical regions via exponential transformations allows the model to focus on areas with significant dynamic changes, thereby enhancing prediction accuracy. These results demonstrate that the choice of segmentation method impacts the model's ability to learn dynamic behaviors, and the introduction of additional metrics reveals that physics-aware segmentation effectively aligns mesh partitions with the system's inherent physical properties, thereby benefiting the learning process.

## D.3 Number of Segmentation

Table 9 and Table 10 present the RMSE-1, RMSE-all, and various mesh quality metrics as the total number of mesh segments is varied during training on three different datasets. In general, M4GN maintains stable performance with relatively low variance, indicating that results are not highly sensitive to segment count. This robustness ensures reliable accuracy across different mesh granularities. However, increasing the number of segments—thereby reducing finite elements per segment—can lead to slight decreases in accuracy and performance.

To comprehensively evaluate the effect of segment number and determine the optimal segmentation for a given dataset, we analyzed prediction accuracy across a wide range of segment counts (from 3 to 51)

Table 8: Ablation study on different segment extraction methods over different dataset.

| Segmentation Method | Dataset | $GF_h \downarrow$ | $GF_c \downarrow$ | MC $\downarrow$ | Aspect Ratio $\downarrow$ | RMSE-1 | RMSE-all |
|---|---|---|---|---|---|---|---|
| METIS | Cylinder | - | - | - | - | 3.44E-03 | 4.59E-02 |
| | Plate | 5.32E-03 | 1.36E-05 | 5.33E-03 | 2.97E-03 | 2.67E-04 | 3.29E-03 |
| | Beam | 3.88E-04 | 5.61E-08 | 5.18E-03 | 3.09E-03 | 1.15E-05 | 2.16E-04 |
| SLIC-OD | Cylinder | - | - | - | - | 3.28E-03 | 4.40E-02 |
| | Plate | 5.24E-03 | 1.50E-05 | 5.16E-03 | 3.04E-03 | 2.69E-04 | 3.70E-03 |
| | Beam | 3.78E-04 | 5.41E-08 | 5.39E-03 | 3.60E-03 | 1.21E-05 | 2.31E-04 |
| SLIC-OD$_l$ | Cylinder | - | - | - | - | 3.33E-03 | 4.37E-02 |
| | Plate | 5.11E-03 | 8.01E-06 | 5.09E-03 | 2.90E-03 | 2.81E-04 | 3.44E-03 |
| | Beam | 3.95E-04 | 5.44E-08 | 5.33E-03 | 3.30E-03 | 1.18E-05 | 2.68E-04 |
| SLIC-OD$_e$ | Cylinder | - | - | - | - | 3.21E-03 | 3.95E-02 |
| | Plate | 5.27E-03 | 1.31E-05 | 4.58E-03 | 2.54E-03 | 2.61E-04 | 3.51E-03 |
| | Beam | 3.81E-04 | 5.68E-08 | 5.40E-03 | 3.32E-03 | 1.20E-05 | 2.51E-04 |
| SLIC-MD | Cylinder | - | - | - | - | 3.16E-03 | 5.62E-02 |
| | Plate | 5.10E-03 | 8.38E-6 | 4.67E-03 | 2.53E-03 | 2.74E-04 | 3.02E-03 |
| | Beam | 3.81E-04 | 5.45E-08 | 5.32E-03 | 3.20E-03 | 1.17E-05 | 2.32E-04 |
| SLIC-MDOD$_l$ | Cylinder | - | - | - | - | 4.16E-03 | 5.29E-02 |
| | Plate | 4.84E-03 | 7.23E-06 | 4.56E-03 | 2.47E-03 | 2.68E-04 | 2.82E-03 |
| | Beam | 3.53E-03 | 5.10E-08 | 5.29E-03 | 3.40E-03 | 1.22E-05 | 2.25E-04 |
| SLIC-MDOD$_e$ | Cylinder | - | - | - | - | 3.09E-03 | 3.86E-02 |
| | Plate | 4.26E-03 | 6.49E-06 | 4.73E-03 | 2.58E-03 | 2.71E-04 | 2.40E-03 |
| | Beam | 3.02E-03 | 4.47E-08 | 5.31E-03 | 3.08E-03 | 1.17E-05 | 2.01E-04 |

during training on the DeformingBeam dataset. The impact of varying the number of mesh segments on prediction accuracy is illustrated in Figure 6 and Figure 7. According to the plots, we identify 19 segments as the optimal number. At this segmentation level, the model achieves the lowest RMSE and Chamfer Distance, indicating high prediction accuracy and precise shape representation. The Hausdorff Distance is also minimized, reflecting excellent alignment between the predicted and true meshes. While the Silhouette score peaks at 9 segments—suggesting well-defined and compact clusters—the slight decrease at 19 segments is offset by significant gains in other performance metrics. Choosing a lower number of segments, such as 3 or 9, may result in higher Silhouette scores but can compromise mesh detail and prediction accuracy due to insufficient spatial granularity. Conversely, selecting a higher number of segments beyond 19 shows diminishing returns, with only marginal improvements or slight degradations in some metrics and a continued decline in Silhouette scores, potentially indicating over-segmentation and unnecessary computational complexity.

In conclusion, when presented with a new dataset, the optimal number of segments can be determined by first computing Silhouette scores for various segment counts to assess cluster cohesion and separation without requiring model training. This provides initial guidance on meaningful segmentation levels. Subsequently, training the model with different segment numbers and evaluating performance metrics like RMSE, Hausdorff Distance, and Chamfer Distance will help identify the point where performance improvements plateau or begin to reverse, indicating the optimal balance between segmentation detail and model efficacy.

## D.4 Influence of Positional Encoding

Table 10 and Figure 8(a) show the effect of adding positional encoding for small and large numbers of segments across three datasets. According to the results, we identified several key findings. Firstly, the effectiveness of PE depends on the number of segments: in the CylinderFlow and Deforming Plate datasets, incorporating PE with fewer segments improves performance across multiple metrics by reducing positional ambiguity. With low segment counts, each segment covers larger, more diverse areas, limiting the model's spatial detail and understanding of segment relationships. PE provides explicit positional information, allowing the model to distinguish distinct regions within the same segment and better comprehend their interactions. However, as the number of segments increases and spatial resolution improves, the benefits of PE diminish and may even introduce unnecessary complexity that hinders performance. Additionally, dataset-specific factors influence PE's effectiveness; for example, the DeformingBeam dataset, with its complex geometry and deformation, did not benefit from PE. This indicates that PE's success depends not only on segment count but also on

how well the PE implementation aligns with the dataset's unique characteristics. Consequently, tailored PE approaches that consider specific geometry and deformation patterns are necessary for complex systems to achieve performance gains. In summary, while PE enhances the performance of graph-based networks, further advancements are needed to develop optimal encoding strategies that consistently improve performance across diverse dynamic systems.

### D.5 Influence of Segment Overlap

Table 9 and Figure 8(b) illustrate the effect of adding segment overlap for small and large number of segments across three datasets. According to the results, the effectiveness of adding overlap between segments ($\delta = 1$) depends on both the segment count and the characteristics of the dataset, such as dimensionality, mesh type, and system dynamics. Overlapping segments are more beneficial with higher segment counts where discontinuities are more prevalent. In Eulerian systems, overlaps enhance the capture of complex interactions and smooth transitions on fixed meshes, leading to improved representation of fluid dynamics. Conversely, in Lagrangian systems where meshes move with the material, overlaps can create redundancy and complicate connectivity, with their impact on model performance varying based on mesh structures and deformation behaviors. For example, in the Deforming Beam dataset, which uses a prism mesh suited for directional deformation, overlapping segments improve performance by facilitating smooth transitions along its mesh surface, especially with a higher number of segments. In contrast, the Deforming Plate dataset employs a tetrahedral mesh with complex, isotropic deformations, where overlaps introduce unnecessary complexity and redundancy, resulting in decreased performance. Therefore, despite both being 3D Lagrangian systems, the different mesh types and deformation patterns explain why overlapping segments benefit the Deforming Beam but not the Deforming Plate.

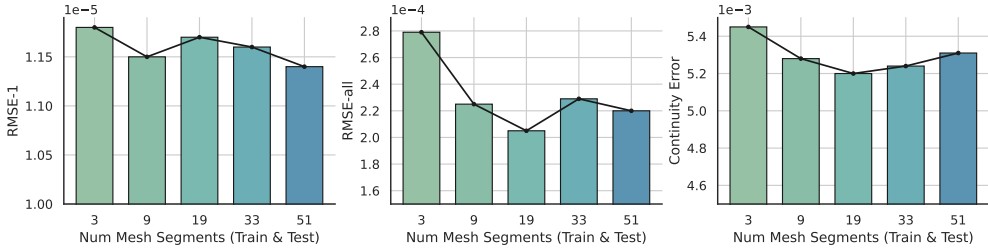

Figure 6: Impact of varying mesh segment numbers during training on prediction accuracy under the DeformingBeam dataset. The number of mesh segments remains consistent during both training and testing. In general, M4GN maintains stable performance with relatively low variance, indicating that results are not highly sensitive to segment count. This robustness ensures reliable accuracy across different mesh granularities. However, increasing the number of segments—thereby reducing finite elements per segment—can lead to slight decreases in accuracy and performance. More detailed analysis on the effect of segmentation numbers on various metrics can be found in Figure 7.

## E  Generalization Studies

To evaluate the generalizability of our M4GN model, we created a larger-scale DeformingBeam dataset, detailed in Appendix A.

### E.1  Performance on Larger-Scale Datasets

Table 11 summarizes the generalization performance of various models trained on the DeformingBeam dataset and directly applied to DeformingBeam(large), a scaled-up version. The results demonstrate that M4GN consistently outperforms all other models across all metrics. In terms of mesh quality, M4GN achieves a 53% improvement over both EAGLE and BSMS for Geometric Fidelity (GF). Similarly, for Mesh Continuity (MC), M4GN achieves the best performance with a value of 1.08e-02, representing a 45% improvement over EAGLE, the next-best model. For the RMSE metrics, M4GN delivers the lowest RMSE-1, RMSE-50, and

Table 9: Ablation study of number of segments, and effect of adding segment overlap.

| Dataset | $N_{\text{SEG}}$ | Overlap | $GF_h \downarrow$ | $GF_c \downarrow$ | MC $\downarrow$ | Aspect Ratio $\downarrow$ | RMSE-1 | RMSE-all |
|---|---|---|---|---|---|---|---|---|
| Cylinder | 16 | ✗ | - | - | - | - | 3.16E-03 | 5.03E-02 |
| | 16 | ✓ | - | - | - | - | 3.19E-03 | 5.35E-02 |
| | 36 | ✗ | - | - | - | - | 3.41E-03 | 4.42E-02 |
| | 36 | ✓ | - | - | - | - | 3.09E-03 | 3.86E-02 |
| Plate | 9 | ✗ | 4.98E-03 | 9.58E-06 | 5.01E-03 | 2.83E-03 | 2.77E-04 | 3.88E-03 |
| | 9 | ✓ | 5.32E-03 | 9.87E-06 | 5.24E-03 | 2.95E-03 | 2.83E-04 | 2.98E-03 |
| | 19 | ✗ | 4.51E-03 | 6.91E-06 | 4.73E-03 | 2.58E-03 | 2.71E-04 | 2.40E-03 |
| | 19 | ✓ | 4.76E-03 | 7.01E-06 | 4.81E-03 | 2.85E-03 | 2.77E-04 | 3.59E-03 |
| Beam | 9 | ✗ | 3.46E-04 | 5.23E-08 | 5.17E-03 | 3.31E-03 | 1.14E-05 | 2.39E-04 |
| | 9 | ✓ | 3.38E-04 | 5.07E-08 | 5.31E-03 | 3.30E-03 | 1.15E-05 | 2.40E-04 |
| | 19 | ✗ | 3.57E-04 | 4.92E-08 | 5.24E-03 | 3.29E-03 | 1.18E-05 | 2.28E-04 |
| | 19 | ✓ | 3.19E-04 | 4.73E-08 | 5.31E-03 | 3.08E-03 | 1.17E-05 | 2.01E-04 |

Table 10: Ablation study of number of segments and whether to add PE or not.

| Dataset | $N_{\text{SEG}}$ | PE | $GF_h \downarrow$ | $GF_c \downarrow$ | MC $\downarrow$ | Aspect Ratio $\downarrow$ | RMSE-1 | RMSE-all |
|---|---|---|---|---|---|---|---|---|
| Cylinder | 16 | ✗ | - | - | - | - | 3.19E-03 | 5.35E-02 |
| | 16 | ✓ | - | - | - | - | 3.34E-03 | 4.76E-02 |
| | 36 | ✗ | - | - | - | - | 3.09E-03 | 3.86E-02 |
| | 36 | ✓ | - | - | - | - | 3.00E-03 | 3.80E-02 |
| Plate | 9 | ✗ | 4.81E-03 | 9.33E-06 | 5.01E-03 | 2.83E-03 | 2.77E-04 | 3.88E-03 |
| | 9 | ✓ | 4.24E-03 | 6.72E-06 | 5.04E-03 | 2.81E-03 | 2.84E-04 | 2.72E-03 |
| | 19 | ✗ | 5.10E-03 | 6.51E-06 | 4.73E-03 | 2.58E-03 | 2.71E-04 | 2.40E-03 |
| | 19 | ✓ | 5.13E-03 | 6.78E-06 | 4.74E-03 | 2.64E-03 | 2.68E-04 | 2.91E-03 |
| Beam | 9 | ✗ | 3.75E-04 | 5.89E-08 | 5.31E-03 | 3.30E-03 | 1.15E-05 | 2.40E-04 |
| | 9 | ✓ | 3.51E-04 | 5.33E-08 | 5.18E-03 | 3.31E-03 | 1.17E-05 | 2.56E-04 |
| | 19 | ✗ | 3.22E-04 | 4.86E-08 | 5.31E-03 | 3.08E-03 | 1.17E-05 | 2.01E-04 |
| | 19 | ✓ | 3.19E-04 | 4.84E-08 | 5.27E-03 | 3.18E-03 | 1.15E-05 | 2.21E-04 |

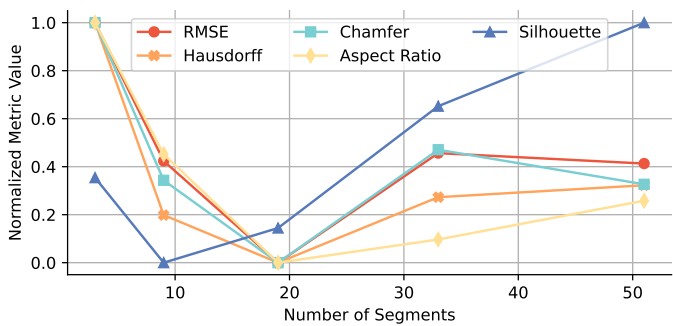

Figure 7: Dependence of various performance metrics on the number of segments in M4GN under Deforming Beam dataset. The plot illustrates how the normalized values of several performance metrics vary with the number of segments. Each metric is represented by a distinct curve, demonstrating the relationship between segment number and overall performance. This figure evaluates the effect of segment number and guides the selection of the optimal number of segments for balanced performance across all metrics.

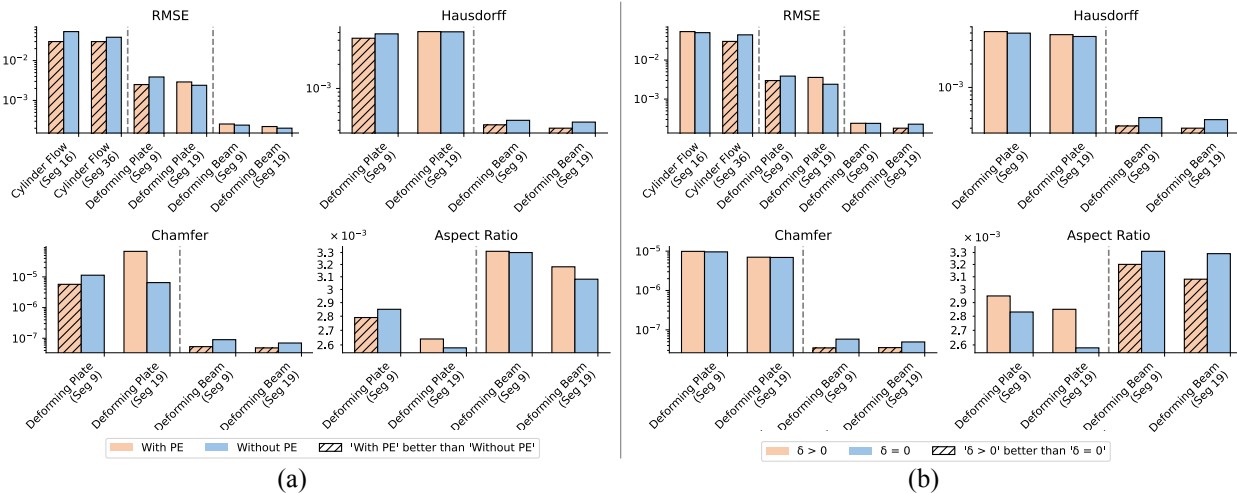

Figure 8: Ablation study on the effects of position encoding and segment overlap across datasets with varying segment numbers. The figure presents the performance metrics for models both with and without the position encoder (a), and with and without considering segment overlap (b) across three distinct datasets, each characterized by a different number of segments. By comparing these conditions, the study highlights how the inclusion of position encoding and the handling of segment overlap influence overall performance, thereby informing the selection of optimal model configurations.

RMSE-all. Notably, M4GN's RMSE-all is 46% lower than EAGLE. These findings suggest that M4GN not only preserves prediction accuracy but also enhances mesh quality when generalizing to larger-scale data, significantly surpassing state-of-the-art models in both accuracy and mesh quality. This demonstrates M4GN's robust generalization ability, making it highly suitable for complex, large-scale dynamical systems. Figure 9 is a visualization of generalization results on DeformingBeam(large) dataset for different models.

Table 11: Generalization performance of our method and five baseline models on the DeformingBeam(large) dataset. M4GN demonstrates great accuracy and mesh quality when generalizing to an unseen dataset with a denser mesh and more extensive long-range dynamic effects.

| Method | $GF_h \downarrow$ | $GF_c \downarrow$ | MC $\downarrow$ | Aspect Ratio $\downarrow$ | RMSE-1 | RMSE-50 | RMSE-all |
|---|---|---|---|---|---|---|---|
| GCN | 2.18e-02 | 3.28e-05 | 1.21e-01 | 1.69e-01 | 2.57e-04 | 1.95e-03 | 1.11e-02 |
| $g$-U-Net | 1.94e-02 | 2.80e-05 | 4.56e-02 | 7.01e-02 | 1.60e-04 | 1.87e-03 | 1.01e-02 |
| MGN | 2.32e-02 | 1.43e-05 | 2.00e-02 | 2.57e-02 | 1.34e-04 | 1.43e-03 | 6.42e-03 |
| BSMS | 1.72e-02 | 3.34e-05 | 1.35e-01 | 1.17e-01 | 4.47e-04 | 3.19e-03 | 1.03e-02 |
| EAGLE | 1.69e-02 | 2.20e-05 | 1.98e-02 | 5.15e-02 | 8.42e-05 | 1.45e-03 | 8.37e-03 |
| M4GN | **7.96e-03** | **5.35e-06** | **1.08e-02** | **2.24e-02** | **5.47e-05** | **9.20e-04** | **4.58e-03** |

### E.2 Effect of Mesh Segment Count on Generalization

**Generalization with Varying Segment Counts During Testing** – Across three datasets, we perform generalization studies where the model is tested using a varying number of segments. The results in Figure 10 illustrate the generalization performance. Pink columns are the references for regular testing and the others are generalization to different number of segments from training. Overall, the M4GN model can generalize very well to different number of segments during testing.

**Impact of Segment Count During Training and Testing** – Equipped with message passing and transformer mechanisms, M4GN can handle an arbitrary number of segments. Figure 11 shows the generalization performance of our M4GN model to larger domain as heatmaps, where models trained with a specific number of segment under deformingBeam dataset are tested with varying number of segments under deformingBeam

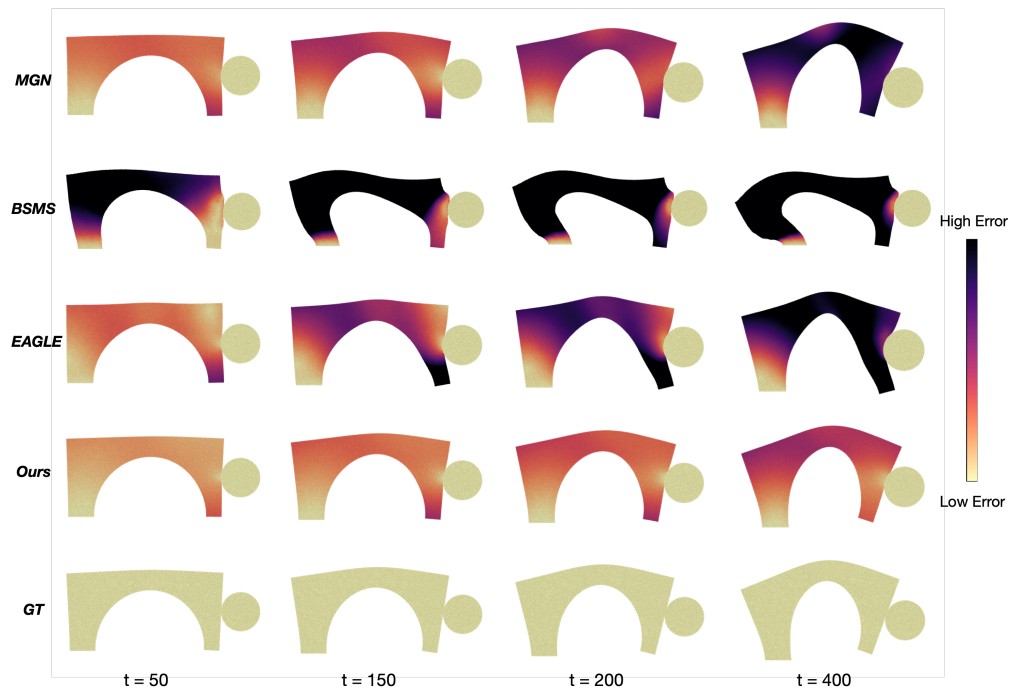

Figure 9: Generalization results for different models under DeformingBeam(large) dataset.

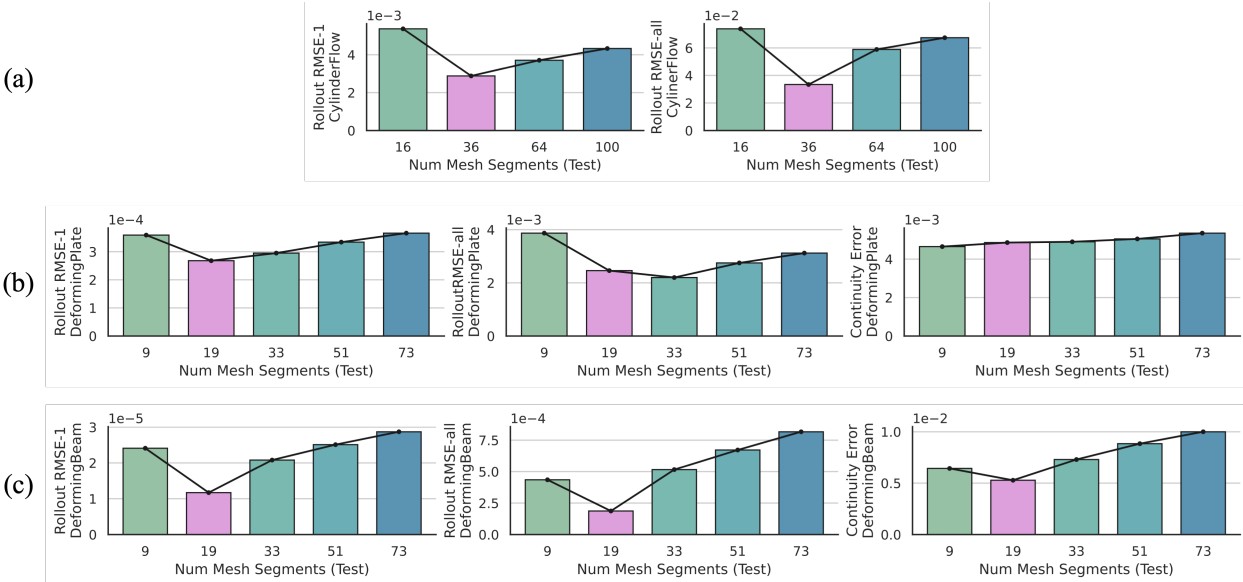

Figure 10: Generalization performance of our method under varying segment counts during testing over three datasets. (a) CylinderFlow: effect of number of segments for test set on different metrics, where model is trained under 36 segments (colored in pink); (b) DeformingPlate: effect of number of segments for test set on different metrics, where model is trained under 19 segments (colored in pink); (c) DeformingBeam: effect of number of segments for test set on different metrics, where model is trained under 19 segments (colored in pink). This figure illustrates that our M4GN model, despite being trained with a fixed number of mesh segments, maintains strong accuracy and mesh quality when tested with varying numbers of mesh segments.

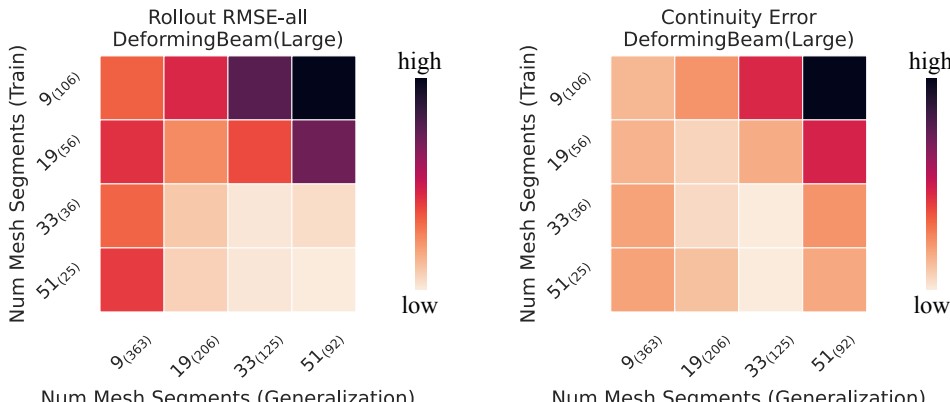

Figure 11: Generalization performance of our method on larger domains under different number of mesh segmentation during training and testing. The subscript of each mesh segment indicating the average number of nodes per segment. M4GN demonstrates robustness and adaptability in handling larger domains with varying mesh segments, making it well-suited for real-world applications involving large and complex mesh structures.

Table 12: Comprehensive evaluation of our method alongside MGN, BSMS, and EAGLE under three datasets. M4GN consistently delivers stable, competitive efficiency while maintaining high accuracy and mesh quality.

| Dataset | Model | RMSE-all | MC ↓ | Train Time per step [ms] ↓ | Train Memory [MB] ↓ | Test Time per step [ms] ↓ | Test Memory [MB] ↓ | Train Time total [h] ↓ |
|---------|-------|----------|------|------|------|------|------|------|
| Cylinder | MGN | 4.81e-02 | - | 66.7 | 698.5 | 20.2 | 67.2 | 37.1 |
| | BSMS | 1.37e-01 | - | 54.7 | 430.3 | 23.8 | 57.9 | 30.4 |
| | EAGLE | 5.83e-02 | - | 69.5 | 618.7 | 28.8 | 230.8 | 38.6 |
| | M4GN | 3.80e-02 | - | 56.2 | 366.6 | 20.0 | 65.0 | 31.2 |
| Plate | MGN | 1.47e-02 | 9.25e-03 | 131.9 | 6021.5 | 36.2 | 445.5 | 73.3 |
| | BSMS | 1.18e-02 | 1.83e-02 | 83.9 | 910.1 | 37.7 | 77.9 | 46.6 |
| | EAGLE | 3.87e-03 | 5.56e-03 | 81.2 | 1090.8 | 32.4 | 362.7 | 45.1 |
| | M4GN | 2.65e-03 | 4.82e-03 | 76.5 | 648.1 | 29.3 | 103.3 | 42.5 |
| Beam | MGN | 4.72e-04 | 1.69e-02 | 79.1 | 1074.4 | 28.6 | 83.8 | 22.0 |
| | BSMS | 4.98e-04 | 3.25e-02 | 61.8 | 213.7 | 30.7 | 35.6 | 17.2 |
| | EAGLE | 4.22e-04 | 5.98e-03 | 53.5 | 410.3 | 26.0 | 153.5 | 14.9 |
| | M4GN | 1.87e-04 | 5.26e-03 | 53.4 | 234.5 | 24.2 | 47.1 | 14.8 |

(large). We observe that better results are seen when the number of nodes per segment during training is less than or equal to that in the generalizing domain, or when the number of segments is greater. Overall, we demonstrate M4GN's robustness and adaptability in generalizing to larger domains with varying mesh segments, making it highly suitable for real-world applications involving large and diverse mesh graphs.

Table 13: The per-step timing of our model against ground-truth simulators across datasets. Since Cylinder-Flow and DeformingPlate are datasets from MGN paper, we adopt their reported values for simulator timing ($t_{GT}$). The time of our model $t_{ours}$ is computed by adding the time used for segmentation and inference on a single NVIDIA Tesla P100 GPU.

| Dataset | solver | $t_{GT}$ ms/step | $t_{ours}$ ms/step | speedup |
|---------|--------|------|------|------|
| CylinderFlow | COMSOL | 820 | 20.4 | 40.2 |
| DeformingPlate | COMSOL | 2893 | 29.7 | 97.4 |
| DeformingBeam | OpenFOAM | 3261 | 24.6 | 132.6 |

## F  Computational Efficiency Analysis

### F.1  Performance Comparison

Table 12 lists the training time, test time, and memory usage for four models MGN, BSMS-GNN, EAGLE, and M4GN across three datasets. The RMSE-all is also listed as a performance reference. Our M4GN model has comparable or better efficiency compared with other models. Notably, the M4GN model has the best efficiency with RMSE-all compared to other baselines. We also compare the per-step timing of our model against ground-truth simulators across datasets in Table 13. Since CylinderFlow and DeformingPlate are datasets from the MGN paper, we adopt their reported values for simulator timing ($t_{GT}$). The time of our model $t_{ours}$ is computed by adding the time used for segmentation and inference on a single NVIDIA Tesla P100 GPU.

### F.2  Complexity Analysis

M4GN is composed of four key components: an Encoder-Process-Decoder (EPD) network operating on mesh graphs, modal decomposition, hybrid mesh segmentation, and a mesh segment transformer. For the learnable part of the model, the computational complexity mainly depends on the EPD and mesh segment transformer components. The complexity of the EPD is: $O(L_1|\mathcal{V}|d^2 + L_1|\mathcal{E}|d^2)$, where $L_1$ is the number of message passing layers, $d$ is the feature dimension, $|\mathcal{V}|$ is the number of mesh nodes, and $|\mathcal{E}|$ is the number of mesh edges. The complexity of the mesh segment transformer is $O(L_2K^2d + L_2Kd^2)$, where $L_2$ is the number of multi-head attention layers, $K$ is the number of segments, and $d$ is the feature dimension. The overall time complexity is $O(L_1|\mathcal{V}|d^2 + L_1|\mathcal{E}|d^2 + L_2K^2d + L_2Kd^2)$.

Modal decomposition and mesh segmentation are performed only once at the initial time step. For modal decomposition, with sparse finite–element matrices, the setup steps—basis construction, matrix assembly, and application of boundary conditions—each require $O(|\mathcal{V}|)$ time. The dominant cost is extracting the first $m$ eigenmodes via a Lanczos/Arnoldi solver; because each Krylov iteration involves one sparse matrix–vector product, the eigen-solve scales as $O(m\,\kappa\,|\mathcal{V}|)$, with $\kappa \approx 10$–$100$ iterations per converged mode and $m$ the number of modes requested. Hence, for a fixed number of modes, the overall algorithm is linear in mesh size. Our hybrid segmentation approach consists of a graph-based method for initial mesh segmentation (METIS) and a superpixel-based method guided by features for refinement (SLIC). In the case of Lagrangian systems where segmentation varies with time, only the refinement part is needed since the initialization segmentation is only based on connectivity and is invariant to feature and coordinate variations. SLIC is based on a local search in k-means clustering, resulting in a time complexity of $O(|\mathcal{V}|)$.

## G  Qualitative Results

Figure 12, 13, 14, and 15 illustrate selected rollout results for all three datasets under different models.

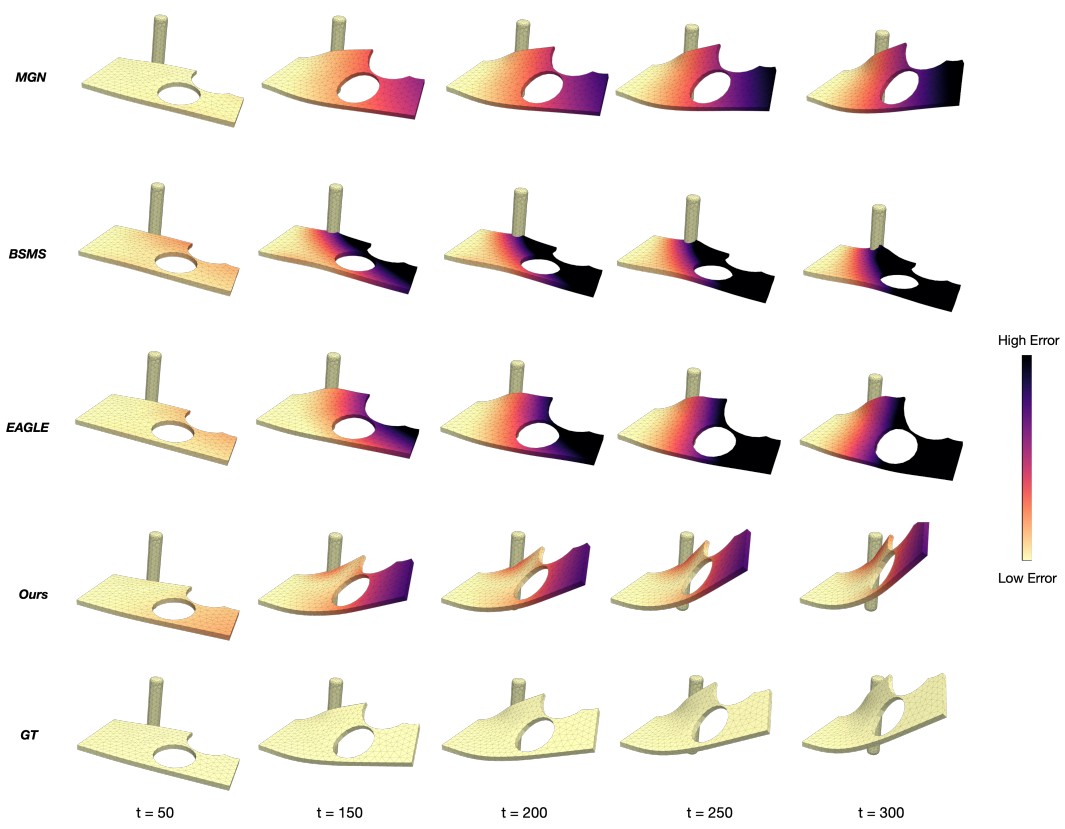

Figure 12: Additional simulation results for different models under DeformingPlate dataset.

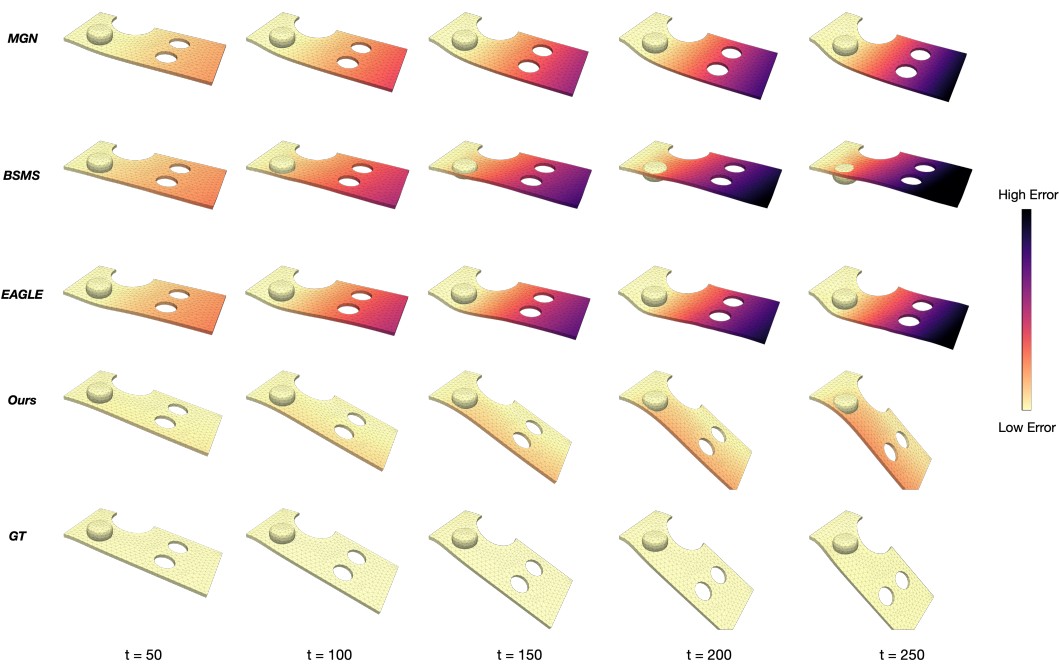

Figure 13: Additional simulation results for different models under DeformingPlate dataset.

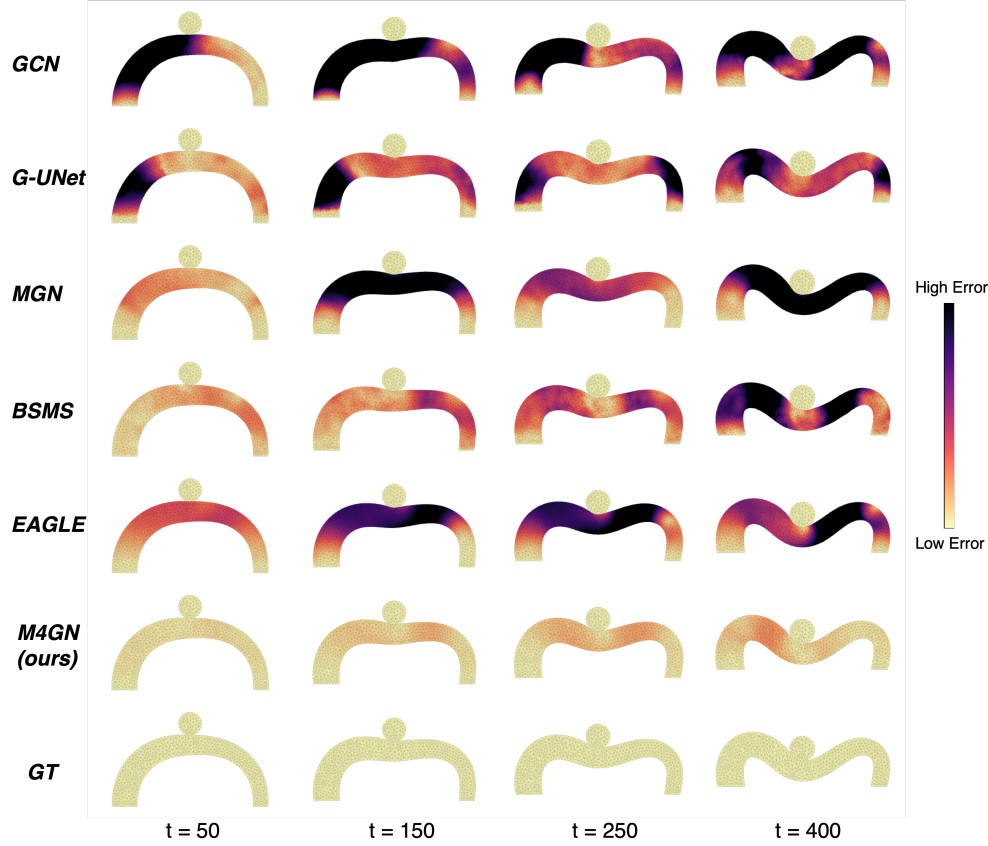

Figure 14: Additional simulation results for different models under DeformingBeam dataset

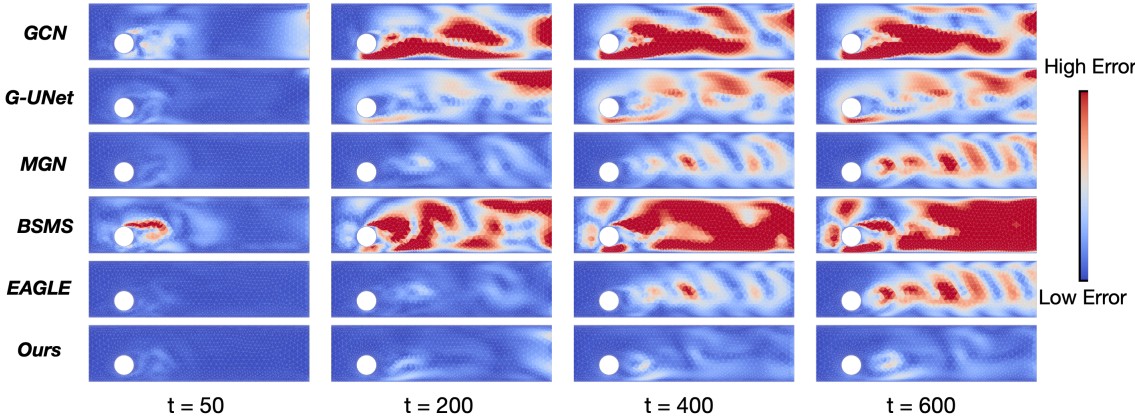

Figure 15: Additional simulation results for different models under CylinderFlow dataset

