# OpenReview forum: "M4GN: Mesh-based Multi-segment Hierarchical Graph Network for Dynamic Simulations"
_TMLR — Accepted by TMLR_

### Review · Reviewer_wSzx · 2025-06-16

**Summary Of Contributions:**

The paper addresses physics simulation on irregular meshes for both fluid and solid dynamics. It claims the following contributions:
- A novel hierarchical framework,
- Superior mesh quality and high prediction error,
- A new dataset modeling the deformation of a beam under a constraint.

The authors also announced public code and dataset release, which is appreciated. They conduct extensive evaluations of various baselines and ablations on three datasets.

**Audience:**

Yes

**Broader Impact Concerns:**

Ultimately, papers should be judged on the knowledge they contribute. In this regard:
- The paper suffers from: missing key datasets (`FlagDynamics` and `Eagle`), infeasible tasks (`CylinderFlow` without pressure), high variability of the conclusions depending on the task and hyper-parameters, and limited impact on average. This makes unclear how the proposed contribution could benefit future research on the subject.
- The impact of the clustering method itself is difficult to assess. The algorithm is only weakly connected to physics (on solid case, with material property), and ablations indicate that the main impactful change is the use of _distance to boundary_ . Several elements (Modal decomposition, Positional embedding and segment overlap) seems to be negative results (which is ok) but are not stated as being negative in the paper.

**Claims And Evidence:**

No

**Requested Changes:**

**Related works**
- The claim "MeshGraphNet being a pioneering work" is factually incorrect. Prior work includes Li et al. (_Learning Particle Dynamics for Manipulating Rigid Bodies, Deformable Objects, and Fluids_) and Chang et al. (_A Compositional Object-Based Approach to Learning Physical Dynamics_), to name a few.
- Related work on physics dataset is in the appendix. It should be moved to the main paper to better motivate the need for another task, and `Eagle` & `MeshGraphNet` datasets should be added to Table 2.

**Methodology**
- I strongly encourage the authors to restructure the methodology to highlight the actual contribution: the segmentation algorithm. One option: synthesize the model description (by deriving it from Eagle), explaining the difference and sharing summarized_analysis of the impact of these changes (while keeping the in-depth analysis in the appendix). This should free space in the main paper allowing to move experiments on segmentation algorithm from the appendix to the main paper, while summarizing the findings.
- Section 3.3.2 is underused, and should be better integrated.
- Section 3.1 incorrectly states that Eulerian systems use fixed meshes. They can use dynamic meshes; the key distinction is field-based vs. particle-based modeling.
- Section 3.2 says "Encoder-Process-Decoder" is used, but there appears to be no decoder at the micro level.

**Results and discussion**
- I would like to highly recommend to tone down the language and the hyperbolas in this section. A scientific paper should not estimate its own impact: "Outstanding performances", "remarkable reduction", "such exceptional performance", "excellent mesh quality", "excels in memory efficiency", "remarkable performances", etc...
- Revisit claims and results as flagged above.
- Clarify how mesh quality metrics relate to physics simulation outcomes.

**Appendix**
- Provide analysis on the actual impact of mode decomposition features in SLIC. Table 2 would benefit from evaluation without mode decomposition features as well (`SLIC-OD` (Linear, Log and Exp)).
- Labels are missing in Tables 4 and 5.
- Specify which variant of M4GN is used (e.g., overlap, PE, which SLIC) in each table.
- Table 9 uses normalized scales (0–1) that obscure real variation, this should be reconsidered.

**Strengths And Weaknesses:**

I’ll first summarize the strengths and weaknesses before detailing their motivations:
1) **Strength**: The authors propose a quite promising idea: _"smart clustering can have an impact on accuracy of hierarchical model in physics_". This is sound and deserves attention from the community. **Weakness:** However, a substantial portion of the paper introduces a "new" model emphasizing long-range interactions—an area already explored in prior work. The proposed model is extremely similar to Eagle.
2) **Strength**: The paper presents several evaluations across tasks, using diverse metrics, including mesh accuracy. **Weakness**: Yet, crucial datasets are missing, and some conclusions drawn from the figures appear questionnable, if not erroneous.
3) **Strength:** the appendix explores the proposed segmentation technique in depth, with several ablations and experiments. **Weakness:** However, the results are poorly analyzed. Readers must compute statistics themselves to assess the impact. These suggest that, on average, the paper's contributions have limited to no effect on performance.

### 1. Limited / no novelty in model

I tried to compile an exhaustive list of differences between M4GN and Eagle (based on Eagle's official implementation on Github):

- **Encoder**: M4GN increases GNN layers from 4 to 7,
- **Segmentation**: core contribution of M4GN -- augments spatial clustering with mesh geometry and material features,
- **Pooling ("Meso")** : M4GN replaces per-segment RNN with average pooling,
- **Mesh Transformer**: increases attention heads from 4 to 8, adds positional embeddings (PE),
- **Decoder**: removes the last GNN layer used in Eagle.

I am uncomfortable with the ambiguity surrounding the comparison to Eagle, given their similarity. The authors repeatedly claim M4GN's superiority stems from its handling of long-range interactions—but Eagle does this too. The paper introduces architectural changes in the main text, but ablations (relegated to the appendix) reveal that positional embeddings and segment overlap offer limited benefit, or are even detrimental on the `Deforming *` datasets.

Given the space dedicated to describing the model (at the expense of results on segmentation algorithm), it may appear—surely unintentionally—that the authors are claiming credit for earlier work on long-range interactions in physics. This could be resolved with a careful rewrite of the main paper.
### 2. Issue with evaluation
Two critical tasks are missing:
1. Not only M4GN shares similar architecture as Eagle, but the authors also tackled similar problems (mesh-based physics simulation) and similar paradigm (transformer-based hierarchical model for long-range interaction). A comparison on the `Eagle` dataset seems mandatory.
2. I would also appreciate evaluation on `FlagDynamic`/`SphereDynamic` from MeshGraphNet paper, which follows a different dynamics. This would strengthen the generalizability of the contribution.

I noticed disturbing choices and claims concerning the results and tasks discussed in the main paper:
- M4GN **only simulates the velocity field on CylinderFlow**, omitting pressure. This makes the task physically unsolvable. Pressure field is mandatory to solve Navier-Stokes equations. The fact that models still succeed in solving the task shows that they can overfit to regularities in CylinderFlow, which highlights the limitations of this dataset (and motivates for more complex one), and also question the capacity of M4GN to actually solve Navier-Stokes simulations. Note that pressure is simulated in previous work (Eagle and MeshGraphNet).
- MeshGraphNet and Eagle are compared on CylinderFlow in the Eagle paper, and shows contradictory conclusions concerning performances and efficiency. This is not discussed here.
- Given offline segmentation and similar architecture, it’s unclear how M4GN outperforms Eagle in efficiency.
- In Figure 4c, all models degrade similarly (Norm. RMSE increases 10x). The only difference is the lower in-domain error for M4GN, yet the paper claims it generalizes better.
- When evaluating on solid dataset, M4GN is implicitly informed of the material properties through clustering, while the other baselines are not, which makes the comparison unfair.
- To disentangle the impact of clustering algorithm vs. model changes, it would have been interesting to use the introduced segments into the hierarchical baseline to see how they would also benefit from it. It would also make the claim stronger, showing that better clustering improves performances on different models.
### 3. Segmentation algorithm seems ineffective
The term "_physics informed_" used throughout the paper might be bit misleading. The term is typically used in the context of PINNs, involving explicit use of the PDE. Here, it only applies to the segmentation algorithm, and only for solids (via material properties). For fluid, clustering is still purely geometric.

The proposed clustering algorithm involves initial graph partitioning and refinement via modal decomposition. The method outperforms simpler method based on local proximity in term of mesh quality, but here, mesh quality is directly used as an optimization target, making this unsurprising. Moreover, the metrics used are from computer graphics and their relevance to physics simulation is unclear.

Moreover, the modal decomposition appears detrimental in average. Frame Table 2, I  found that mode decomposition features degrades performances by 7% with `SLIC-MD` and 1,7% with `SLIC-MDOD-L`. The best performing segmentation method is `SLIC-MDOD-E`, which increases performances by 16%, but gives more importance to the *distance to boundary* features (thanks to the exponential). This raises doubts about the actual utility of the modal feature.

Given the task-dependent and variable results, along with limited analysis, it's difficult to draw clear scientific conclusions about the clustering algorithm’s effectiveness.

---

> ### Author Response · Authors · 2025-07-12
> **Reply to Reviewer  wSzx (Part1)**
>
> We appreciate the reviewer’s careful assessment and insightful comments. The manuscript has been revised, and our point-by-point responses are outlined below.
>
> > Q1: Related Works 1) The claim "MeshGraphNet being a pioneering work" is factually incorrect; (2) Related work on physics dataset should be moved to the main paper and Eagle \& MeshGraphNet datasets should be added to dataset table
>
> - (1) We thank the reviewer for pointing this out and we've adjusted the related work section accordingly. (2) We've moved related work on physics dataset in the main paper. The MeshGraphNet dataset has been included in the dataset table within the manuscript. Also, the table below summarizes the key statistics for the EAGLE dataset. Due to time constraints, we were unable to repeat the full set of ablation studies or carry out thorough hyper-parameter tuning on this dataset, as we did for the others. Consequently, we have opted to omit the incomplete EAGLE results from the present revised revision; additional experiments are underway and the full analysis will be included in the final version.
>
> | Dataset | Avg. # Nodes | # Steps | Mesh Type | Graph Diameter | Node Features | Edge Features | Output |
> |---------|--------------|---------|-----------|----------------|---------------|---------------|--------|
> | **EAGLE** | 3 390 | 990 | Triangle; Eulerian; 2-D | 29.5 ± 1.7 | $v_i$, $p_i$, $n_i$ | $m_{ij}$, $\lvert m_{ij}\rvert$ | $\dot v_i$, $\dot p_i$ |
>
> > Q2: Methodology: (1) to restructure the methodology to highlight the actual contribution; (2) Section 3.3.2 is underused, and should be better integrated. (3) Section 3.1 incorrectly states that Eulerian systems use fixed meshes. They can use dynamic meshes; the key distinction is field-based vs. particle-based modeling. (4) Section 3.2 says "Encoder-Process-Decoder" is used, but there appears to be no decoder at the micro level.
>
> - (1)(2) We've made corresponding adjustments in the revised version. (3) We appreciate the reviewer’s observation and fully agree. Our intent in Section 3.1 was simply to characterize the specific problems addressed in this work. In the revision we now state explicitly that the Eulerian formulation with a fixed mesh applies only to the dataset used here—namely, CylinderFlow. (4) We use the encoder and processor blocks for micro-level message passing, while the decoder head is detached and applied only after the macro-level module. We thank the reviewer for pointing this our and have added this clarification in the revised version.
>
> > Q3: Result and Discussion: (1)  tone down the language and the hyperbolas in this section (2) Clarify how mesh quality metrics relate to physics simulation outcomes.
>
> - (1) We appreciate the reviewer's suggestion and have tone down the language in the revised manuscript; (2) We have added a discussion in the experimental metrics section and included Table 2 to provide a clearer illustration.
>
> > Q4: Appendix: (1) Provide analysis on the actual impact of mode decomposition features in SLIC. Table 2 would benefit from evaluation without mode decomposition features as well (SLIC-OD (Linear, Log and Exp)). (2) Labels are missing in Tables 4 and 5. (3) Specify which variant of M4GN is used (e.g., overlap, PE, which SLIC) in each table.
>
> - (1) We have included additional SLIC-OD results and a deeper analysis in Appendix D.2; (2) We thank the reviewer for spotting the omissions; the missing labels have been added to the relevant tables; (3) The requested details are now provided in Appendix B.1.
>
> > Q5: When evaluating on solid dataset, M4GN is implicitly informed of the material properties through clustering, while the other baselines are not, which makes the comparison unfair. To disentangle the impact of clustering algorithm vs. model changes, it would have been interesting to use the introduced segments into the hierarchical baseline to see how they would also benefit from it. It would also make the claim stronger, showing that better clustering improves performances on different models.
>
> - We appreciate the reviewer’s insightful suggestion. Notably, in our ablation study on different segmentation strategies, we found that even using a mesh-only segmentation method such as METIS—relying solely on input mesh information available to all methods—already yields improved performance over the baseline models. Incorporating material properties further enhances performance beyond this baseline improvement. Nonetheless, we recognize the importance of this direction and consider it a promising avenue for future work to further validate the general utility of our segmentation algorithm.

---

> > ### Author Response · Authors · 2025-07-12
> > **Reply to Reviewer wSzx (Part2)**
> >
> > > Q6: In Figure 4c, all models degrade similarly (Norm. RMSE increases 10x). The only difference is the lower in-domain error for M4GN, yet the paper claims it generalizes better.
> >
> > -  We thank the reviewer for pointing this out and we agree the claim is not adequate. We've revised the claim and discussion related to generalization performance in the experiment section.
> >
> > > Q7: Request to add dataset like (EAGLE and FlagDynamic), also add pressure prediction to CylinderFlow dataset.
> > - We conducted additional experiments on EAGLE datasets and add pressure prediction to CylinderFlow dataset. Due to time constraints, we were unable to complete the experiments on the FlagDynamic dataset. However, we agree that including this additional dataset would strengthen the paper, and we plan to complete the experiments and include them in the final version. Based on the results shown below, our proposed method consistently outperforms the baseline models. A full ablation analysis is still in progress to provide a more comprehensive evaluation.
> > - We follow the same training process in EAGLE paper with the following loss function: $
> >         \mathcal{L} = \sum_{i=1}^{H} \mathrm{MSE} \left( v(t+i), \hat{v}(t+i) \right) + \alpha \sum_{i=1}^{H} \mathrm{MSE} \left( p(t+i), \hat{p}(t+i) \right), $ where we chose $H=5$ and $\alpha=0.1$ according to the best configuration found for EAGLE baseline. The model is optimized for 100,000 steps with Adam optimizer and learning rate of $10^{-4}$. We keep this training configuration consistent across the three models we evaluated (MGN, EAGLE, M4GN). The test rollout result can be found in table below. We have also included total train time, train memory and inference speed as efficiency evaluation.
> > | Method | RMSE$_{V_x}$ | RMSE$_{V_y}$ | RMSE$_{P_s}$ | RMSE$_{P_g}$  | Train Memory (MB) | Test Time per Step (ms) |
> > |--------|--------------|--------------|--------------|-------------------------|-------------------|---------------------------|
> > | MGN   | 1.448 | 1.503 | 5.908 | 7.786  | 10525 | 35.8 |
> > | EAGLE | 1.336 | 1.512 | 6.147 | 7.625 | 7254  | 35.2 |
> > | M4GN  | 1.265 | 1.473 | 5.305 | 7.462| 5308  | 28.4 |
> >
> > - We have conducted additional experiments on the CylinderFlow dataset that include pressure as both input and output, alongside velocity. Specifically, the node inputs now consist of velocity $\mathbf{v}_i$, pressure $p_i$, and node type $\mathbf{n}_i$. The model predicts the time derivatives $\dot{\mathbf{v}}_i$ and $\dot{p}_i$, which are then integrated to obtain the next-step state.  We follow the same training setup as in the original velocity-only experiments. It is worth noting that velocity and pressure are fundamentally different physical quantities, and ideally, a weighting scheme should be introduced in the loss function to reflect their relative importance or scales. However, in this experiment, we assign equal weight to both terms for simplicity. We will update the main text after running more ablation studies using a weighting scheme.
> > | Method | RMSE$_{V_x}$ | RMSE$_{V_y}$ | RMSE$_{P}$ |
> > |--------|--------------|--------------|------------|
> > | MGN   | 2.192 e−02 | 1.637 e−02 | 2.083 e−02 |
> > | EAGLE | 2.516 e−02 | 1.613 e−02 | 1.227e−02 |
> > | M4GN  | 1.974 e−02 | 1.611 e−02 | 1.116 e−02 |
> >
> > > Q8: Clearly state (1) why our method have better performance than EAGLE. (2) why the impact of positional embeddings and segment overlap varies across datasets.
> >
> > - 1) In the revised manuscript (in introduction and experiment section), we've added thorough discussion talking about the limitations of EAGLE and the improvements we had that result in better performance in terms of both accuracy and efficiency. (2) We appreciate the reviewer’s observation. As detailed in Appendix D.3, we provide a detailed analysis of the impact of positional embeddings and segment overlap, along with the underlying reasons for their varying effectiveness. This variability underscores the importance of tailoring architectural choices to specific data characteristics. We hope this analysis offers practical guidance for researchers looking to adapt our framework to different domains.
> >
> > > Q9: MGN and EAGLE are compared on CylinderFlow in the EAGLE paper, and shows contradictory conclusions concerning performances and efficiency.
> >
> > -  In the EAGLE paper, they used a customized loss function that balance the velocity prediction loss and pressure prediction loss. The paper also trains the models differently with our settings by summation of the loss over multiple time steps without adding training noise. This is different from our training process, where one-step training and noise induction are applied. They also used different sets of hyperparameters for EAGLE and MGN, which doesn't reflect a direct comparison between model capability. Therefore, the relative performance relationship found in our results could be different from the EAGLE paper.

---

> > > ### Author Response · Authors · 2025-07-12
> > > **Reply to Reviewer wSzx (Part3)**
> > >
> > > > Q10: (1)The term "physics informed" used throughout the paper might be bit misleading. (2) The method outperforms simpler method based on local proximity in term of mesh quality, but here, mesh quality is directly used as an optimization target, making this unsurprising. (3) the metrics' relevance to physics simulation is unclear.
> > >
> > > - (1) We thank the reviewer for the helpful observation and have removed this terminology from the revised manuscript to prevent potential confusion. (2) We'd like to clarify that the mesh‑quality metrics used in this paper are never used as loss terms or optimization targets. Our hybrid segmentation strategy is an unsupervised process driven solely by (i) graph partitioning and (ii) clustering in the feature space. Mesh‑quality scores are computed a posterior for evaluation. Because the algorithm does not minimize any of the four metrics (Hausdorff, Chamfer, Mesh Continuity, Aspect-Ratio error) during training, improvements in these metrics are not a consequence of the objective function itself; rather, they indicate that the resulting segments are genuinely more coherent and therefore lead to better downstream predictions. (3) Table 2 in the revised manuscript now links each mesh‑quality metric to its geometric rationale and clarifies its implications for simulation outcomes.
> > >
> > > > Q11: The modal decomposition appears detrimental in average. Frame Table 2, I found that mode decomposition features degrades performances by 7\% with SLIC-MD and 1,7\% with SLIC-MDOD-L. The best performing segmentation method is SLIC-MDOD-E, which increases performances by 16\%, but gives more importance to the distance to boundary features (thanks to the exponential). This raises doubts about the actual utility of the modal feature.
> > > - We value the concerns raised by the reviewer. In fact, modal decomposition is indeed beneficial to the overall performance of the modal and below we provide some detailed analysis (based on the updated Table 5 in the revised manuscript).
> > > - We observe that refining the coarse METIS partitions with SLIC improves accuracy only when the added SLIC features better align local cuts with the true physics; otherwise, the refinement can fragment physically coherent regions and hurt performance. SLIC‑OD, for example, uses only geometric distance to boundaries; on DeformingPlate this over‐weights proximity and splits mode‑consistent areas, so RMSE increases relative to the original METIS segmentation. Likewise, on DeformingBeam the distance‑only (SLIC‑OD) and modal‑only (SLIC‑MD) variants either ignore contact boundaries or long‐range bending modes, producing finer—but less meaningful—segments and therefore higher error than METIS. Only the combined SLIC‑MDOD$_e$, which couples modal information with an exponentially weighted distance term, strikes the right balance between global coherence and local adaptation.
> > > - In terms of the impact of modal decomposition features in SLIC, for the two solid‑mechanics cases (DeformingPlate and DeformingBeam), modal‑decomposition features enrich the descriptor space with physics‑relevant mode shapes, and  every OD+MD variant—SLIC‑MDOD$_l$ and SLIC‑MDOD$e$—beats its OD‑only counterpart across all metrics, delivering up to 35\% lower RMSE and up to 25\% better mesh‑quality scores. These consistent gains confirm that modal information and boundary‑distance cues are complementary for solid mechanics. On CylinderFlow, however, the flow is dominated by rapidly varying vortical patterns; the MD basis, derived purely from geometry in this Eulerian setting, adds little new information and can perturb the SLIC clusters, so SLIC‑MD alone shows a slight RMSE rise. When MD is combined with the (exponentially weighted) distance cue in SLIC‑MDOD$e$, the boundary‑aware term stabilizes the segmentation while the modal vectors still provide complementary detail, giving a net improvement over distance or modal information used in isolation.
> > >
> > > Once again, we thank the reviewer for these thoughtful comments and would welcome any additional suggestions that could help further improve our manuscript.

---

> ### Comment · Reviewer_wSzx · 2025-07-17
> **General reply**
>
> I thank the authors for their thorough reply and manuscript revision, which address some of my concerns. However, my main remarks remain. The revision refactors the Introduction and Results sections, while the most controversial section, Methodology, remains unchanged, despite all reviewers requesting significant modifications.
>
> The authors have addressed part of my concerns regarding evaluation and have committed to adding missing benchmarks and fixing the infeasibility of the `CylinderFlow` task. While these additional experiments are understandably time-consuming, they are essential to properly evaluate the proposed contribution.
>
> I remain concerned about the similarity with `Eagle`, which is still not explicitly acknowledged. The authors have added relevant discussions regarding baseline limitations (poor clustering and the impact of RNN aggregation on inference time). However, the manuscript continues to motivate M4GN superiority through its hierarchical structure, which it shares with `Eagle`, leaving unclear why M4GN performs better.
>
> **Recommendation**
>
> I am willing to revise my rating if the authors:
> 1. **Substantially refactor the Methodology section**, focusing on:
>     - Highlighting the clustering technique as the primary contribution.
>     - Presenting results in a clear, structured manner.
>     - Explicitly stating where the method succeeds or fails.
> 2. **Address debatable claims identified by all reviewers**. The first revision improved the manuscript, but the current conclusions are now scattered and still unclear. The authors should redefine their claims in light of the existing literature and interpret results accordingly, explicitly stating positive and negative findings in the main paper.

---

> > ### Author Response · Authors · 2025-07-18
> > **Author's reply to Reviewer wSzx's general reply**
> >
> > We appreciate the reviewer’s careful reading of our revisions and the additional questions raised; while we concur that clearer conclusions and a balanced presentation of both positive and negative findings should be incorporated into the main text—an improvement currently underway—we would also like to clarify several of the reviewer’s observations and elaborate on certain points that may have been misunderstood.
> >
> > > Methodology, remains unchanged, despite all reviewers requesting significant modifications
> >
> > - We respectfully disagree with this statement raised by the reviewer: the methodology section has been revised in direct response to reviewer feedback. Reviewer `ohdF` did not request methodological changes, whereas reviewe `rufzj` asked for greater clarity, which we fully addressed by(i) explicitly highlights our mesh-segmentation contribution, (ii) removes the ambiguous “meso” terminology, (iii) inserts a dedicated pre-processing section on the segmentation method before the framework description. These modifications collectively refine the methodological exposition and incorporate all requested changes.
> >
> > > The manuscript continues to motivate M4GN superiority through its hierarchical structure and list hierarchical architecture as a contribution
> >
> > - We appreciate the reviewer’s concern but we respectfully disagree with the suggestion to totally remove the hierarchical architecture as one of our contributions. In fact, we have revised the manuscript to clarify that we do not attribute performance gains to “being hierarchical” per se; rather, M4GN is designed to address two concrete challenges: (i) constructing physically coherent sub-graphs from irregular meshes and (ii) balancing predictive accuracy with computational efficiency at scale. Our hybrid mesh-segmentation pre-processing step directly targets the first challenge and yields an intermediate-level module that is substantively different from EAGLE, which does not use this hybrid segmentation strategy. At the macro level, M4GN reasons over learned segment embeddings using a segment-based transformer to capture long-range interactions, again differing from EAGLE’s aggregation design. The hierarchical layout is therefore an enabling scaffold that integrates segmentation with multi-scale message passing to achieve the desired accuracy/efficiency trade-off; it is not the sole or universal source of superiority. We have tempered language throughout to make this scope explicit and now motivate results in the context of these two challenges, with empirical comparisons provided accordingly.

---

> > > ### Comment · Reviewer_wSzx · 2025-07-22
> > > **Reply**
> > >
> > > Thank you for your reply, which clarifies the motivations behind the paper. I look forward to reviewing the final revision, which will certainly inform my final evaluation.
> > >
> > > I would like to clarify my concerns regarding the comparison between M4GN and Eagle:
> > > > At the macro level, M4GN reasons over learned segment embeddings using a segment-based transformer to capture long-range interactions, again differing from EAGLE’s aggregation design.
> > >
> > > This sentence summarizes well my concerns:
> > > - *M4GN reasons over **learned** segment embeddings*
> > >
> > > Neither the segmentation algorithm nor the aggregation method in M4GN is learned. I assume “_learned_” here refers to the encoder (a series of GNN layers), which is identical to Eagle. If it instead refers to the aggregation method, it should be noted that Eagle uses a learned RNN-based aggregation, while M4GN uses a non-learned max pooling.
> > >
> > > - *using a segment-based transformer*:
> > >
> > > Eagle also applies a “mesh-transformer” on segments (termed _clusters_ in that work). I do not see a difference between M4GN and Eagle in this aspect.
> > >
> > > - *to capture long-range interactions*:
> > >
> > > This is also Eagle’s core motivation: using transformers on clusters to capture long-range dependencies. The sentence is essentially a reformulation of Eagle’s abstract:
> > > > It leverages node clustering, graph pooling and global attention to learn long-range dependencies between spatially distant data points
> > >
> > > - *again differing from Eagle aggregation*:
> > >
> > > Here, the authors are correct: M4GN replaces Eagle’s per-cluster RNN with max pooling.
> > >
> > > To be clear, **the resemblance between Eagle and M4GN is not, in itself, a reason for rejection. The concern is the lack of clarity around this resemblance within the paper.** From this discussion, it appears we agree that the meaningful differences between Eagle and M4GN are: (1) the segmentation algorithm, and (2) the aggregation method. While improving segmentation in hierarchical models is a clear and valuable direction, it remains unclear why replacing an RNN with max pooling would improve predictive accuracy. Therefore, I once again request the following changes:
> > > 1. **Remove ambiguities regarding Eagle**: Explicitly state which parts of M4GN are derived from Eagle, which are modified, and the rationale for these modifications.
> > > 2. **Clarify the motivation for max pooling**: Provide a clear explanation of why replacing the RNN with max pooling is expected to improve results.
> > >
> > > I will not engage further in discussion until the revised manuscript is available. Despite my critical review, I believe the paper offers valuable contributions to the community; however, its message is currently obscured by:
> > > 1. Difficult-to-follow experiments, making it hard to assess the contribution:
> > > 	- Lack of relevant datasets,
> > > 	- Insufficient discussion of generalization capabilities.
> > > 	- Issues with the fluid mechanics dataset.
> > > 	- Unclear attribution of performance gains (e.g., privileged information through segmentation, training setup differences).
> > > 	- Variable performance accross tasks.
> > > 2. Claims that appear overstated:
> > > 	- The generalization capability appears limited, but is presented as a success in the paper,
> > > 	- Performance is attributed to design choices that are not unique to M4GN (long-range interaction, transformer on clusters)
> > > 	- Implicitly mentioning hierarchical reasoning with transformer as a contribution.
> > >
> > > The authors have announced substantial changes, and I will review the revised manuscript carefully once available.

---

> > > > ### Comment · Reviewer_wSzx · 2025-07-22
> > > >
> > > > > Question related to fine-tuning feature balances during segmentation
> > > >
> > > > Thank you for clarifying. I suggest refactoring the appendix to include a dedicated section on **“Sensitivity to Hyperparameters.”** This could (1) list the main degrees of freedom in your method and (2) briefly illustrate their impact on performance. While I understand this is partly covered in Figure 8, organizing it around hyper-parameter sensitivity rather than an ablation study would improve clarity for readers.
> > > >
> > > > > M4GN benefits from privileged information via modal decomposition based on material properties
> > > >
> > > > There may be a misunderstanding here. I do understand that material properties are not directly provided as input to M4GN. However, since the segmentation algorithm is driven by these properties, the resulting segments implicitly encode material-related information. A sufficiently powerful model could exploit these segment compositions to recover material properties indirectly, aiding prediction without explicit supervision.
> > > >
> > > > In contrast, the baselines use segmentation methods unrelated to material properties and cannot leverage this implicit information, potentially placing them at a disadvantage. It is therefore possible that part of M4GN’s performance gains stem from this additional source of information, rather than solely from the improved clustering method.
> > > >
> > > > Evaluating the impact of this factor would strengthen the paper’s conclusions. For instance, using M4GN-generated segments to train other hierarchical baselines could test whether the segmentation itself, rather than the model modifications, drives the performance improvements. This would help disentangle the contributions of the segmentation algorithm from those of the aggregation method.
> > > >
> > > > > The generalizability of M4GN to larger meshes is now minimally discussed, with effectively negative results.
> > > >
> > > > The statement _“M4GN attains the lowest RMSE and mesh‑continuity (MC) error”_ is accurate but does not address generalization. Generalization concerns maintaining performance on out-of-domain data. Since RMSE scales similarly across models, it does not indicate superior generalization by M4GN.
> > > >
> > > > Regarding _“M4GN’s MC increase is the smallest”_, as I understand, mesh continuity depends solely on the segmentation method rather than the model’s predictions. From this, it appears that:
> > > > 1. The M4GN model architecture does not generalize better than baselines.
> > > > 2. The segmentation method, which is not inherently tied to M4GN and could be applied to any hierarchical model, exhibits better scaling on large meshes.
> > > >
> > > > I recommend revising the end of Section 4.2.4 to clarify this distinction.
> > > >
> > > > > Question related to contradiction with prev. work
> > > >
> > > > Understood. While your training setup is standard, it remains unclear whether Eagle would outperform M4GN if trained under its original setup. If Eagle underperforms under its own conditions, it would strengthen your claims; if it outperforms M4GN, it would indicate that the contribution’s impact is less clear.
> > > >
> > > > To clarify this, I suggest:
> > > > 1. Replicating Eagle’s evaluation protocol on the CylinderFlow dataset with M4GN for a direct comparison, and use metrics reported in the Eagle paper,
> > > > 2. Comparing M4GN trained on Eagle (which the authors are doing for the revision) against Eagle using pretrained weights, which I are available.
> > > >
> > > > Both approaches would not require additional model training and would help clarify the comparison.
> > > >
> > > > > Question related to mesh quality
> > > >
> > > > Thank you for clarifying this point in the manuscript. If feasible and useful for reader understanding, I suggest explicitly adding the exact mathematical term being optimized during mesh refinement.

---

> > > > > ### Author Response · Authors · 2025-07-26
> > > > > **Authors' Reply to Reviewer wSzx (part1)**
> > > > >
> > > > > We appreciate the reviewer’s thorough evaluation of our revised manuscript and the thoughtful follow-up comments. We also value the constructive exchange and are pleased that several earlier misunderstandings have been clarified. The manuscript has been further revised to address the points raise by the reviewer, and our responses to the reviewers remaining points/concerns are outlined below:
> > > > >
> > > > > > Q1: Request changes to (1) remove ambiguities regarding Eagle and (2) clarify the motivation for max pooling
> > > > >
> > > > > - We appreciate the reviewer’s concerns and have revised the manuscript to address both points. Specifically, we expanded the discussion of point (1) in the Introduction and Contributions sections. For point (2), we provide a brief mention in the Introduction and a more detailed explanation in Section 3.3.1 of the Methodology.
> > > > >
> > > > > > Q2: Dataset related: (1) lack of relevant datasets and (2) issues with the fluid mechanics dataset
> > > > >
> > > > > - (1) We have added our initial results for the EAGLE and Flag datasets, using hyperparameters selected based on insights drawn from the three primary benchmark datasets. These results already demonstrate that our method consistently outperforms baseline models across various evaluation metrics. We are currently conducting additional ablation studies to maintain consistency with the rest of the benchmarks and will include them in the final version once complete.
> > > > > (2) We assume the reviewer is referring to the previously missing prediction results for specific properties on the CylinderFlow dataset. We have now updated Table 3 to include these results. Additionally, we provide a comparison below following EAGLE’s original evaluation protocol and reporting metrics consistent with those used in the EAGLE paper for direct comparison on CylinderFlow dataset. In the table below, +1 represents RMSE-1, +50 represents RMSE-50, +250 represents RMSE-250, **V** represents velocity,**P** represents pressure .
> > > > >
> > > > > | Model | +1&nbsp;V | +1&nbsp;P | +50&nbsp;V | +50&nbsp;P | +250&nbsp;V | +250&nbsp;P |
> > > > > |-------|-----------|-----------|------------|------------|-------------|-------------|
> > > > > | MGN   | 0.0004 | 0.0016 | 0.0047 | 0.0095 | 0.0144 | 0.0145 |
> > > > > | EAGLE | 0.0003 | 0.0007 | 0.0044 | 0.0035 | 0.0179 | 0.0079 |
> > > > > | **Ours** | **0.0001** | **0.0004** | **0.0034** | **0.0030** | **0.0109** | **0.0048** |
> > > > >
> > > > > > Q3: Unclear attribution of performance gains --- The baselines use segmentation methods unrelated to material properties and cannot leverage this implicit information, potentially placing them at a disadvantage. It is therefore possible that part of M4GN’s performance gains stem from this additional source of information, rather than solely from the improved clustering method.
> > > > >
> > > > > - We understand the concerns raised by the reviewer and we agree that incorporating material-aware information could introduce an implicit advantage. However, as shown in Table 8, the proposed hybrid segmentation still yield better performance than baseline models even when material properties are not implicitly or explicitly considered during segmentation (e.g. METIS, SLIC-OD). This indicates that the observed gains are not solely due to implicit access to material priors, but rather reflect the effectiveness of the segmentation strategy itself in capturing meaningful structure.
> > > > > - Moreover, as suggested by the reviewer, we further conducted the segmentation transfer experiment (Table 6) and have shown that both components (i.e. better segmentation and the modified model) contribute to the overall gains of our method.
> > > > >
> > > > > > Q4: Variable performance across tasks --- I suggest refactoring the appendix to include a dedicated section on “Sensitivity to Hyperparameters.” This could (1) list the main degrees of freedom in your method and (2) briefly illustrate their impact on performance.
> > > > >
> > > > > - We thank the reviewer for the helpful suggestion. In response, we have reorganized the appendix to include a dedicated Hyperparameter Sensitivity Analysis section (Appendix D). To improve clarity and accessibility, we also provide a summary in Table 7 outlining the key hyperparameters in our model, their tested ranges, observed performance impacts, and practical tuning recommendations based on the findings from these analysis.

---

> > > > > > ### Author Response · Authors · 2025-07-26
> > > > > > **Authors' Reply to Reviewer wSzx (part2)**
> > > > > >
> > > > > > > Q5: Insufficient discussion of generalization capabilities. The generalization capability appears limited, but is presented as a success in the paper. Regarding “M4GN’s MC increase is the smallest”, as I understand, mesh continuity depends solely on the segmentation method rather than the model’s predictions.
> > > > > >
> > > > > > - We've revised the experiment section regarding the discuss of the generalization capabilities. Also, we would like to clarify that the statement "mesh continuity depends solely on the segmentation method rather than the model’s predictions" is NOT true. Our mesh‑continuity (MC) metric is computed after inference from the predicted 3‑D coordinates of all mesh nodes; it quantifies local geometric discontinuities across edges or faces (e.g., sudden stretch, compression, or tearing relative to neighboring elements). Consequently, MC depends on the quality of the predicted displacements, not on the segmentation used inside the network. Conversely, a hypothetical “dummy” model that predicts \emph{zero} displacement for every node would indeed give a low MC value while still having a large RMSE; thus low MC is necessary but not sufficient for overall accuracy. In practice, however, we observe a empirical correlation between RMSE and MC across our experiments, confirming that higher‑fidelity predictions tend to yield more continuous meshes. Therefore, when we state that “M4GN’s MC increase is the smallest,” we mean that M4GN’s predicted mesh coordinates remain the most geometrically consistent under scale‑up (compared to other hierarchical models), not that its segmentation alone enforces continuity.
> > > > > >
> > > > > > > Q6: Question related to contradiction with prev. work (1) Replicating Eagle’s evaluation protocol on the CylinderFlow dataset with M4GN for a direct comparison, and use metrics reported in the Eagle paper (2) Comparing M4GN trained on Eagle against Eagle
> > > > > >
> > > > > > - Both experiments have been completed as requested; please see the table provided in Q2 above for (1) and Table 5 in the revised manuscript for (2).
> > > > > >
> > > > > > > Q7: Question related to mesh quality. I suggest explicitly adding the exact mathematical term being optimized during mesh refinement.
> > > > > >
> > > > > > - We thank the reviewer for the suggestion, however, introducing the full formalism would require several new symbols and supporting definitions, which would burden the main pseudocode and distract from the high‑level flow. To keep the algorithm listing readable, in the revised Algorithm 2, we instead describe the refinement step in plain language and hope this could be clear enough for the readers.
> > > > > >
> > > > > > In addition to the changes mentioned above, we have made further revisions throughout the paper to clarify our contributions and ensure that our claims are well-supported. Once again, we sincerely thank the reviewer for their time, thoughtful feedback, and constructive discussion throughout the review process. We are more than willing to address any further questions or suggestions.

---

> ### Comment · Reviewer_wSzx · 2025-07-17
> **Detailled reply**
>
> > **Q1: Situate new dataset**
>
> The added paragraph in Related Work clarifies the motivation for the new dataset. However, I strongly recommend adding _at least_ Eagle and FlagDynamic to Table 4. This table compares DeformingBeam to existing datasets, it does not require training any models.
>
> > **Q2: Refactor**
>
> The paper presents the architecture as a contribution while being very close to prior work, at the expense of properly describing the actual contribution—the clustering algorithm (also raised by `ufzj`). Section 1.3 still lists hierarchical architecture as a core contribution, ignoring my remarks.
>
> > **Q3: Tone down**
>
> This has been addressed.
>
> > **Q4/11: impact of segmentation**
>
> While the added results help with assessing the impact of better segmentation, my concerns remain:
> - According to the author's reply, **one of the main contributions of the paper (modal decomposition) is detrimental for fluid mechanics problems**. This is a huge limitation, which is (1) not mentioned nor referenced in the paper, and (2) barely discussed in the appendix.
> - The authors also mentioned that the application of their method to solid mechanics requires fine-grained balancing between the different features during segmentation. This makes the generalizability of the method to other tasks rather difficult. I agree with reviewer `odhF` on this matter.
>
> While the added results help assess the impact of improved segmentation on hierarchical models, concerns remain:
> - According to the authors’ reply, **one of the paper’s main contributions (modal decomposition) is detrimental for fluid mechanics problems**. This significant limitation is (1) not clearly stated in the main paper and (2) only briefly discussed in the appendix. This limitation should be explicitly stated, potentially even in the title (e.g., “… for solid mechanics”).
> - The authors note that applying their method to solid mechanics requires fine-tuning feature balances during segmentation, making generalization to other tasks challenging. I share reviewer `odhF`  concerns on this point: the method relies on several hyperparameters that seem to have a significant impact on performance, while being hard to optimize.
>
> > **Q5: Unfair comparison**
>
> The authors note that simple mesh-based clustering already improves M4GN performance, but:
> 1. This is difficult to understand from the architectural similarity between M4GN and the baselines. The drawbacks of `Eagle` cited by the authors (RNN aggregation and mesh-based clustering) do not fully explain the performance differences.
> 2. This does not address the concern: M4GN benefits from privileged information via modal decomposition based on material properties. Baselines may be capable of extracting this information through segmentation, potentially reducing the performance gap with M4GN. Yet this is unverified.
>
> The lack of clear, well-supported results on this matter contributes to blurring the paper’s claims regarding performance.
>
> > **Q6: Generalization claim**
>
> The authors addressed concerns by removing the paragraph analyzing Figure 4c, replacing it with two (arguably ambiguous) sentences in Section 4.2.4. The generalizability of M4GN to larger meshes is now minimally discussed, with effectively negative results.
>
> > **Q7:Wrong use CylinderFlow, add Eagle**
>
> The authors have committed to fixing the incorrect use of `CylinderFlow` (lacking the pressure field) and to adding `Eagle` to their benchmarks. Preliminary results are promising, and I look forward to seeing these added to the paper.
>
> > **Q8: M4GN > Eagle**
>
> While the authors have discussed `Eagle` limitations, these do not fully explain M4GN performance gains. The paper continues to attribute M4GN performance to its hierarchical structure, which it shares with `Eagle`.
>
> > **Q9: Contradiction with prev. work**
>
> This is a significant concern. The authors indicate that the performance gap stems from training pipeline choices favoring M4GN. This is not a fair baseline comparison. The authors should either use the baseline proposed training pipeline (applicable here since they use the same dataset) or optimize baselines using the same methods applied to M4GN.
>
> The authors seem to acknowledge that better training pipelines could improve baseline performance, raising further concerns about the fairness of comparisons.
>
> > **Q10: "physics-informed" & mesh quality**
>
> Replacing “physics-informed” with “physics-aware” and “physically coherent” avoids confusion with PINNs, though the terminology remains somewhat ambiguous since the method relies on geometry and material properties rather than physical state. This is a minor concern.
>
> Regarding mesh quality evaluation: Algorithm 2 explicitly states that it “*refines the partitioning on $G_\text{fine}$ to improve quality,*” implying that mesh quality is optimized during clustering. This directly benefits the proposed method over baselines that do not use mesh quality in clustering.

---

> > ### Author Response · Authors · 2025-07-18
> > **Author's reply to Reviewer wSzx's detailed reply**
> >
> > > Question related to impact of segmentation
> >
> > - We would like to clarify that our primary contribution is the hybrid mesh‑segmentation strategy, not the modal decomposition technique alone; **incorporating this segmentation approach yields results that surpass all baselines across all datasets (including fluid mechanics)**, as demonstrated in Table 5. Modal decomposition simply supplies one feature set used to guide the SLIC refinement, while obstacle‑related features serve as another, both of which are integral parts of our proposed hybrid segmentation pipeline.
> >
> > > Question related to fine-tuning feature balances during segmentation
> >
> > - Noted in our response to reviewer `odhF`, the hybrid segmentation framework does introduce additional hyper‑parameters, yet **even with default or minimally tuned settings it consistently outperforms all baselines**; further hyper‑parameter adjustment simply refines the results to achieve optimal performance rather than enabling the gains in the first place.
> >
> > > M4GN benefits from privileged information via modal decomposition based on material properties
> >
> > -  We would like to clarify that the modal‑decomposition vectors we employ are derived entirely from mesh geometry and material parameters already present in all datasets, so the information is no more “privileged” to M4GN than nodal coordinates are to any baseline; our contribution lies in exploiting that physics‑aware signal through the hybrid segmentation pipeline. Moreover, we've shown that even when modal features are withheld, the hybrid segments alone yield substantial gains over EAGLE‑style geometric partitions  (Table 5).
> >
> > > The generalizability of M4GN to larger meshes is now minimally discussed, with effectively negative results.
> >
> > - We respectfully disagree with the assertion that M4GN’s scalability results are “effectively negative.” As detailed in the manuscript, M4GN attains the lowest RMSE and mesh‑continuity (MC) error on both the standard and enlarged DeformingBeam benchmarks; although every hierarchical model exhibits some error growth when the mesh doubles in size, M4GN’s MC increase is the smallest, indicating more stable performance under mesh refinement. While we do not claim universal best generalization across every possible metric and dataset, these results demonstrate that our method preserves accuracy better than competing hierarchical methods as mesh size grows.
> >
> > > Question related to contradiction with prev. work
> >
> > - To clarify our earlier response: we **applied the exact same training configuration** used for MGN to all baselines, so **M4GN received no special advantage**. The discrepancy arises with EAGLE, whose original implementation alters MGN’s training settings; when we re‑evaluate EAGLE under the unchanged MGN configuration, its performance differs from the results reported in its own paper.
> >
> > > Question related to mesh quality
> >
> > - We would like to clarify that **mesh‑quality metrics are NOT optimized during clustering** when we state that the algorithm “refines the partition to improve quality,” we mean it enhances the coherence of the partition itself, not the mesh‑quality measure used for evaluation, and we will revise the manuscript to make this distinction explicit.

---

> ### Comment · Reviewer_wSzx · 2025-07-29
> **Final reply**
>
> Thank you for your reply and the revision of the paper, which I reviewed thoroughly, and I must say that the paper has greatly improved since its first submission. It is now scientifically sound, so I revised my decision accordingly. However, I note that the inputs for CylinderFlow are still composed of the velocity only, and not the pressure field. As I mentioned, this makes the task practically unsolvable. The fact that M4GN and the baselines succeed in predicting its dynamics shows the limitation of this dataset, but also questions the conclusions for fluid dynamics tasks. However, the addition of the Eagle dataset to the benchmark clarifies the situation on fluid dynamics.
>
> **Typo**:
> - (Section 1.3) (ii) a permutation-invariant max-pooling aggregator that is **insensitivity** to mesh node orders and **computational** efficient ...
> - (Section 2.3) The **pseudocodeof** the ...
> - (Appendix A.1) : FlagSimple is not a "Fluid System"
> - Figure 7(abc) is duplicated with Figure 2(abc)

---

### Review · Reviewer_odhF · 2025-06-18

**Summary Of Contributions:**

The authors propose a novel method for mesh-based simulation that includes a micro-level part, akin to popular encode-process-decode GNN-based methods. A macro-level stage helps the model simulate long-range interactions by processing features aggregated from sub-graphs which are defined based on meso-level information, such as modal decomposition and material properties. The results from these stages are combined to produce the estimate of physical quantities at the next time step, thus proposing an autoregressive emulator for mesh-based representations. The key claim is the method outperforms current data-driven baselines, whose goal is to address the slowness of traditional methods, while staying competitive in terms of computational efficiency. They also provide a new dataset which includes both a normal version and a scaled-up one to demonstrate generalization of mesh-based emulators.

**Audience:**

Yes

**Claims And Evidence:**

No

**Requested Changes:**

I think the method will definitely be interesting to related audience. Beside answering the questions above (some of them mainly for curiosity and better understanding), I think that several changes are required, and others would strengthen the paper.

Therefore, I suggest accepting with modifications.

**Changes essential for acceptance:**
- Address the typos and miscellaneous stuff listed above.
- Depending on your answer to the questions about Equation 10, please clarify that part.
- Fix Figure 4 and inconsistencies with Table 7. Please adapt your discussion and wording depending on the adapted figure.
- Clarify and reword metrics discussion, especially with respect to Eulerian systems and the importance of mesh quality metrics for these systems versus Lagrangian systems. It must be clear that *assessing* it might be less critical (and not the quality itself) as you assume you have some initial given mesh. Less importantly, could you slightly elaborate on the overall importance of mesh-quality metrics and how they are computed to be reported?
- Clarify and reword section 4.2.3 with better interpretation of the robustness of your method to scaling up. If your method is not capable of scaling up as well as MGN (2k to 20k nodes), then you *need* to reword claims about robustness to scaling up and generalization, or even completely remove that claim. That claim needs to be backed up by strong evaluation and empirical results.
- Improve your discussion of the limitation, if possible address (some of) the points I raised in the weaknesses, or answer them here if you think I missed or incorrectly understood something.
- Provide more details and better motivation for the choice of your datasets as an experimental setup. Do these datasets cover most practical uses? What is varying over the different simulations of your sets? What are the sizes of the different splits? What kind of physical information is required and provided to the model? What are the predicted metrics? This should all be in the main text, in a concise manner.

**Changes That Would Strengthen the Work (Recommended but not Essential for Acceptance):**
- Run your method on some of the other publicly available datasets, such as those proposed by MGN (e.g., Flags). I understand you provided many ablation and additional studies for the three datasets and this requires work. Snapshots and a metric table would already strongly help, even in the appendix.
- Even in the appendix only, please provide more information and comparisons with the simulators used for the datasets and the advantages of ML-based methods. Are GPU-accelerated simulators much slower?
- Please try to provide some videos as part of your supplementary material. Considering you do not aim for fixed-point PDE resolution, videos would very well highlight the difference with other methods over time and could strengthen your points.
- Could you add the missing ground truth in Figure 15? Even if one of the methods reaches low error, it is always better to visualize what it actually should be. Maybe even Figure 3 (b & c) would benefit from it, although I think it is less necessary considering the low error makes me guess it's very close to GT?

**Strengths And Weaknesses:**

### Strengths
- The paper is easy to follow and clearly divided.
- The figures are high-quality and easy to understand, which helped during reading.
- The overall method and its components are well-motivated.
- Code and data will be released, and many details are provided, especially in the appendix.
- Various ablation studies and experiments to analyze the impact of different hyperparameters.
- The method seems to beat current baselines, either or both in terms of accuracy and computational efficiency, on the chosen datasets.
- The choice of the baselines seems adequate.

### Weaknesses
- Very efficient methods exist for mesh-based simulations. One of your key claims is to provide an alternative to expensive simulators. Actual figures and comparisons with the ground-truth simulators used for each dataset would strengthen this argument, which looks neglected in the rest of the paper.
- The main text lacks some details that require going through the appendix for proper understanding, and even often explicitly suggests going into the appendix.
	- The datasets description is very short and not motivated in the main text, even in the results discussion.
	- For mesh-quality metrics, you motivate them but only list them. Could you maybe add a simple sentence that highlights what kind of cases it covers or a geometric intuition behind them?
	- Predicted quantities for each dataset. Also, what exactly does the model require as input (all physical quantities for all three stages) for each dataset?
- More importantly with respect to the mesh quality metrics, you incorrectly state "mesh quality is less critical" as the mesh remains constant. Mesh quality is extremely important. In your case, I guess you assume you have access to some given mesh over which you have no control, as it's external to your method, so you assume it's good, and either do not need to assess it here if it's fixed, because your method does not make it vary, or you need to because your method modifies it in Lagrangian systems. This is very different, and must be made clear in the text.
- Some results across radar plots and tables are inconsistent. Although the radar plots are useful to highlight how M4GN's high performance is more versatile across these 3 datasets than the baselines, a few checks lead to inconsistencies between these figures and reported values in Table 7 of the appendix, except if you used a different method to compute these numbers, which I doubt considering some values do match. For example, training speed for MGN should respectively be 0.82 and 0.58 in CylinderFlow and DeformingPlate, and training memory efficiency should respectively be 0.52 and 0.11. Instead, they are all set to 0.2. On CylinderFlow, this strongly undermines how close MGN actually is to your method. Just another example issue is EAGLE's testing memory efficiency, which "declines dramatically" (Section 4.2.2) should be 0.28 and not 0.2 in the radar plot. It seems the common issue is many values set to 0.2?
- MGN reports generalization capabilities from a training mesh of 2k nodes to a test mesh of 20k nodes (Figure 4b on Page 6). First, could you run your method in such conditions? Otherwise, do you think it *could* perform as well in such conditions? In your larger version of DeformingBeam, MGN managed to scale-up quite well, especially in terms of mesh continuity. This also questions the actual contribution of the DeformingBeam dataset, if much larger datasets are available.
- Your method includes a certain number of hyperparameters to tune, such as the number of segments. You provide a concrete strategy, which is appreciated, but it requires training some models on 10% of the data. On CylinderFlow, do you include this in your total training time? Otherwise, this process which seems needed for any new dataset adds $n \times \frac{T}{10}$ where $n$ is the number of $K$ values tested, and $T$ the total training time. For CylinderFlow, this adds a 15h overhead to training time if not parallelized. This is a minor concern but I think there should be more discussion, especially in the limitations, with respect to the complexity of using your method, compared to e.g. MGN. Also, do the time and memory performance include the overhead of the segmentation (structural mode computation + SLIC/METIS)? This overhead might be a bit confusing because it requires comparing both total time (which includes it) and separately mentioning that per-step performance excludes it.
- Compared to the text dedicated to explaining the method, the experimental part is weaker.
	- With respect to the number of datasets, why have you chosen these three and not included others, such as the different flag datasets in MGN? These datasets exhibit quite different mechanics and interaction with static world objects. Does your method also outperform the baselines on these datasets?
	- In section 4.2.3, you state that your method generalizes the best to scaled-up datasets. This is misleading; your method does indeed stay the best in terms of accuracy on the scaled-up version, but MGN better preserved its initial mesh quality (which is not much higher), which means MGN is more invariant. This questions the robustness to larger-scale datasets, which is suggested as a key point of your method.
- The conclusion is also a bit weak and short and would be strengthened by more discussion, other words than "significant" and more proper discussion about the practical usage of your method. But the part that is more neglected to me is limitations:
	- The datasets might look carefully selected to only highlight cases where your method applies, without any mention of the limits if this is the case. You mention the possibility of overlapping meshes and physical consistency at segmentation interfaces. Did you observe this in any datasets? When could this happen and what kind of behavior can one expect versus over methods? The invariance of mesh quality (and MSE) with respect to the size of the dataset for generalization does not seem as good as you claim it is.
	- Even though you provide principled ways to select the hyperparameters, the method looks much harder to both implement and use, than e.g. MGN.
	- Compared to MGN, if I'm not wrong, your method also requires more input variables and physical information?
	- Would the static segmentation method suffice for different physical cases?
	- In terms of scaling (such as for Section 4.2.3), can we expect larger $K$ values for very large systems? The $O(K^2)$ complexity will as some point limit the enhancements over purely micro-level-stage level methods such as MGN, especially in terms of performance.
	- One of your three key contributions include a dataset, but it is barely described and discussed in the main text, and not even mentioned in the conclusion.

### Questions
- I am less familiar with mesh simulations and mesh-quality metrics. Are the latter computed at every time step then averaged over time? Only at the end of the rollout?
- It is unclear how you operate for Lagrangian meshes to update the mesh once $\dot{x}_{t+1}$ is predicted. Do you simply use Euler integration?
- Compared to MGN, which is essentially a "sub-part" of your method, how do you achieve such greater memory efficiency? Is it because your additional components allow to reduce the complexity of the micro-scale module?
- Do you pre-cache the segmentation for each simulation of your training set? Do you re-run this part at every training step?
- I am not very familiar with these methods, such as EAGLE, which seemed like a strong contestant. Do you think the accuracy and performance of your method compensate for its added complexity? In terms of ease of implementation and use, how do your method and the baselines compare?
- In Equation 10, I do not think defining a $Cut$ operator helps describing the quantity $A_{ij}^K$. I am unsure I properly understood your definitions. You state that the sub-graphs are disjoint. My understanding is that for each pair of segments and for each pair of connected nodes between them, only one of the two nodes is owned by both segments? Otherwise, the size of the intersection of node sets is not equal to $\sum\_{m \in \mathcal{V}\_{S\_i}} \sum\_{n \in \mathcal{V}\_{S\_j}} A\_{mn}$. How exactly do you define the links between disjoint graph partitions/segments?

### Typos & misc.
The following are just some minor notes and typos I noted during my readings.
- In section 2.1, you cite Ummenhofer et al. (2019) (i.e., continuous convolutions) as convolutional networks that cannot operate on irregular geometries, but this work precisely introduces convolutions that can be applied on point clouds. Convolutions are anyway a sub-case of graphs. CNNs with positional encoding or continuous convolutions could run on irregular meshes. I think you can simply refer to MGN which shows, among other works, that even graphs (without mesh positional encoding) perform very poorly on meshes. At least remove or replace this incorrect sentence and citation.
- Table 7 does not report the number of parameters, which would be very interesting to report, even in the main text.
- In the whole paper, you mostly use "dynamic" instead of "dynamical" to refer to the simulated systems. Although it is the correct term for some parts (e.g., dynamic quantities), I think "dynamical" is more appropriate, especially when referring to dynamical systems. This notably includes the title, several occurrences in the abstract, and others in the main text.
- In section 3.2, right below Eq. (1), you forgot to put $\in \mathcal{V}$ for ${\bf x}_{i,t}$.
- In the "Structural Modal Analysis" paragraph of section 3.3.2, after the citations of Fu & He and Wilson, an "is" is missing after "which", and on the line below, "it is typical to select" and not "select***s***".
- On page 6, in the "Detailed Methodology" paragraph, on the last line, "this information" instead of "these".
- At the very end of section 3.4.1: "Table 5 presents ablation studies show how [...]". "which" or "that" is missing.
- In section 3.4.3: penultimate line of page 7, "an MLP" (instead of "a") and the same for "an $L_2$ loss".
- Section 4.1, "Datasets" paragraph. "We create ***the*** DeformingBeam ...".
- Still Section 4.1, "Metrics" paragraph, last sentence: "Detail***ed***" instead of "Details".
- Caption of Figure 4 (a): "different models" and not "difference models".
- Appendix A.2 DeformingBeam: which ***is*** a toolbox.
- Appendix C.4: "To visualize the how predict mesh properties in each segment various through time for different method"; I guess you meant "To visualize how predicted mesh properties in each segment vary through time for the different methods".
- Later in that same paragraph in appendix C.4: "our methods is able" $\to$ "our method is able".
- Next sentence: "in case with in inherent" $\to$ "in the case of inherent".
- Section D.2 of appendix, first sentence. "As shown in Table 3, ***the*** mesh segment***ation*** method ...".
- Second line of D.3: "small and large number***s***" (missing s at "number").
- Very end of page 26: "effect ... on various metrics" and not "to".
- Title of appendix section F.2: Complexity instead of Compelxity.

---

> ### Author Response · Authors · 2025-07-12
> **Reply to Reviewer odhF (Part1)**
>
> We thank the reviewer for the thorough evaluation of our paper and for the valuable comments provided. We have revised the manuscript accordingly, and our detailed responses to each question are provided below.
>
> > Q1: Question regarding Equation10.
>
> - We agree that the use of the $Cut$ operator may cause confusion here, given that the mesh segment graphs $S_i$ and $S_j$ may share boundary nodes. We have revised the definition and removed the $Cut$ operator.  We have clarified this in the main text to ensure the definition of links between segments is consistent with the (potentially overlapping) partitioning of the mesh.
>
> > Q2: Clarification regarding Figure 4 and Table 7.
>
> - We appreciate the reviewer’s careful cross-checking and the helpful observation regarding the radar plots and Table 7. We would like to clarify that for metrics such as training time and memory usage, lower values indicate better performance. However, in radar plots, it is customary to visualize higher values as better, so we applied reversed min-max normalization to these metrics prior to plotting. For instance, the training speed for CylinderFlow was transformed to [MGN: 0.2, BSMS: 1.0, EAGLE: 0.0, M4GN: 0.9] after reversed normalization, which is consistent in the radar plot. Furthermore, to avoid collapsing the radar plot into a single point when a method performs worst across all metrics (resulting in a dot at the center), we applied value clipping such that the minimum normalized score is set to 0.2. This ensures more readable and interpretable visualizations without changing the relative ranking. We've clarified these processing steps in the revised manuscript.
>
> > Q3: Regarding mesh quality metrics: (1) Clarify and reword metrics discussion (2) elaborate on the overall importance of mesh-quality metrics (3) highlights what kind of cases it covers or a geometric intuition behind them
>
> - (1) We thank the reviewer for pointing out this ambiguity statement we had in the
> paper. We were meant to say that for Eularian system, the data we used is given as fixed mesh
> and cannot experience element-quality degradation. Therefore, the mesh quality metrics used to
> evaluate the predicted output is not applicable in this case. We’ve made such statement clearer
> in the revised version. (2)(3) We've added more descriptions on these mesh quality metrics in the experimental section and added a table (Table 2) for better summarization.
>
> > Q4: Clarify and reword section 4.2.3 with better interpretation of the robustness of your method to scaling up.
>
> - We thank the reviewer for pointing this out and we've adjusted our claims regarding this matter with more in-depth discussion in the revised experimental section (Section 4.2.4).
>
> > Q5: Provide more details and better motivation for the choice of your datasets as an experimental setup
>
> - We've added explanations for our dataset choices in both the experimental section and the related work section. We have also included the predicted quantities and the required input features for each dataset used by the model.
>
> > Q6: Actual comparisons with the ground-truth simulators used for each dataset
>
> - We have added Table 10 to the revised manuscript, presenting a comparison of computation times between the ground-truth simulators and our model, along with further discussion in Appendix F.1.
>
> > Q7: Question regarding hyperparameter tuning and time used for segmentation alone.
>
> -  We appreciate the reviewer's thoughtful comments on including hyperparameter tuning in total training time. We didn't include this part since we intended to discuss the computational efficiency of the model only. We have acknowledged this potential limitation in the conclusion section, where we discuss the additional complexity introduced by our method. The time and memory performance doesn't include the overhead of the segmentation since we are focused on the per-step performance. The time used for average segmentation time for a simulation case can be found in the following table.
> | Dataset         | $N_{\text{SEG}}$ | $t_{\text{EAGLE}}$ (ms) | $t_{\text{M4GN}}$ (ms) |
> |-----------------|------------------|-------------------------|-------------------------|
> | CylinderFlow    | 94               | 903                     | 410                     |
> | DeformingPlate  | 64               | 462                     | 284                     |
> | DeformingBeam   | 38               | 334                     | 143                     |

---

> > ### Author Response · Authors · 2025-07-12
> > **Reply to Reviewer odhF (Part2)**
> >
> > > Q8: Question regarding possibility of overlapping meshes and physical consistency at segmentation interfaces.
> >
> > - Such overlap tends to arise in out-of-distribution scenarios with large deformations or aggressive coarsening: segments experiencing very different motions can drift into one another, and coarse pooling further reduces geometric fidelity. We've added a visualization for zero-shot generalization test on the scaled-up DeformingBeam case in Figure 9 that can reveal some of these situations. In our experiments, this penetration is most pronounced in baseline models that use global pooling, while M4GN’s hybrid segmentation confines overlap to small, localized regions; nonetheless, all methods would benefit from future contact-aware constraints to guarantee physical consistency at segment interfaces.
> >
> > > Q9: Questions related to comparison with MGN: (1) the method looks much harder to both implement and use, than e.g. MGN. (2) Compared to MGN, your method also requires more input variables and physical information? (3) Compared to MGN, how do you achieve such greater memory efficiency? Is it because your additional components allow to reduce the complexity of the micro-scale module?
> >
> > - (1) While MGN is structurally simpler, it struggles on complex systems that demand precise modeling of long-range interactions. Our method adds several components and hyper-parameters, but the code is fully modular, enabling quick adaptation to new tasks. Although these extra hyper-parameters introduce some tuning overhead, we have observed that even with minimal adjustment our approach reliably surpasses MGN’s performance.  (2) Our approach does need on physical information for modal-analysis in hybrid segmentation, but these information are ordinarily produced when solving the governing PDEs to generate the training data, so they are readily available at no extra cost. Moreover, the ablation study shows that even when segmentation uses only geometric features (incurring minimal added complexity) the method still outperforms MGN. (3) Yes. The additional components allow effective long-range modeling and subsequently requires less amount of layers in the micro-scale module, hence achieving better computational and memory efficiency. We've added more discussion on this matter in both revised introduction and experimental section.
> >
> > > Q10: Would the static segmentation method suffice for different physical cases?
> >
> > - For Eulerian system, as the mesh is fixed, static segmentation is suffice. For Lagrangian system, when a dataset is quasi-static (those used in this paper), i.e., displacements stay small and patterns remain anchored to the initial geometry, a one-time segmentation is adequate because element adjacencies and high-energy regions hardly change. If the system undergoes large rigid-body motion or localised strain—even slowly—the static blocks lose coherence and an adaptive scheme is advisable. While our method currently employs static segmentation, it is feasible to implement dynamic re-segmentation, either after each time step or periodically, to refine mesh segments based on evolving dynamics.
> >
> > > Q11: In terms of scaling (such as for Section 4.2.3), can we expect larger $K$ values for very large systems? The $O(K)$ complexity will as some point limit the enhancements over purely micro-level-stage level methods such as MGN, especially in terms of performance.
> >
> > - We appreciate the reviewer’s insightful question. For very large systems, purely micro-level methods like MGN can quickly reach performance bottlenecks due to oversmoothing and and limited receptive fields in the physical domain, which restrict effective long-range information exchange. In such cases, incorporating hierarchical modules becomes necessary to better capture global system behavior and achieve high predictive performance.
> > - Regarding the complexity, we do not necessarily expect the number of segments $K$ to scale linearly with system size. In practice, if $K$ scales as $O(\sqrt{|\mathcal{V}|})$, the complexity of the mesh transformer becomes $O(L_2K^2d) = O(L_2|\mathcal{V}|d)$, which is comparable to the complexity of micro-level GNN layers, typically $O(L_1|\mathcal{V}|d^2)$. Empirically, we observe that the micro-level GNN layers dominate the overall runtime, while the mesh transformer modules contribute a relatively small computational overhead.

---

> > > ### Author Response · Authors · 2025-07-12
> > > **Reply to Reviewer odhF (Part3)**
> > >
> > > > Q12: Questions related to experiment settings: (1) Are the mesh quality metrics computed at every time step then averaged over time? Only at the end of the rollout?  (2) It is unclear how you operate for Lagrangian meshes to update the mesh once $\dot{x}_{t+1}$ is predicted. Do you simply use Euler integration? (3) Do you pre-cache the segmentation for each simulation of your training set? Do you re-run this part at every training step?
> > >
> > > - (1) Mesh quality is evaluated at every time step then averaged over time—by comparing the predicted mesh to the ground‑truth configuration. (2) Yes, we use Euler integration to advance the simulation. (3) Mesh segmentation are performed only once at the initial time step. The segmentation ids are cached and used at every training step.
> > >
> > > > Q13: Could you add the missing ground truth in Figure 15?
> > >
> > > -  We've added that in the revised version (now is Figure 14).
> > >
> > > > Q14: Address the typos and miscellaneous stuff listed above.
> > >
> > > - We appreciate the review for carefully reviewing our paper and have addressed those typos in the revised version.
> > >
> > > Once again, we thank the reviewer for these thoughtful comments and would welcome any additional suggestions that could help further improve our manuscript.

---

> > > > ### Comment · Reviewer_odhF · 2025-07-21
> > > > **Follow-up reply to authors**
> > > >
> > > > I thank the authors for the thorough revision of your paper and addressing my comments. I have some remaining remarks and questions.
> > > >
> > > > > Q2: Clarification regarding Figure 4 and Table 7.
> > > >
> > > > I appreciate your explanations providing more details about how you made this plot. I am still not convinced radar plots are necessary, especially considering these additional processing steps. Bar plots would probably provide a more comprehensive and precise illustration of the comparisons, as I think the radar plot only "looks good". I think the final values obtained after normalization quite poorly represent the values found in the tables and what a bar plot would represent. However, thanks for providing the details in the plot figure caption, this is not a mandatory change to me anymore, but this would still be appreciated.
> > > >
> > > > > Q6: Actual comparisons with the ground-truth simulators used for each dataset
> > > >
> > > > Thanks for providing these figures. I appreciate the details in section F.1, though I recommend you also include them in the caption of Table 10 for better understanding, as they are quite important. Are the simulator figures on CPU?
> > > >
> > > > > Q12: Questions related to experiment settings: (1) Are the mesh quality metrics computed at every time step then averaged over time? Only at the end of the rollout? (2) It is unclear how you operate for Lagrangian meshes to update the mesh once  is predicted. Do you simply use Euler integration? (3) Do you pre-cache the segmentation for each simulation of your training set? Do you re-run this part at every training step?
> > > >
> > > > (1) Thanks for clarifying. Maybe I missed it, but could you perhaps simply add that it is averaged over time in the main text when specifying that you use mesh quality metrics?
> > > >
> > > > > Q14: Address the typos and miscellaneous stuff listed above.
> > > >
> > > > Thanks for addressing most of the stuff. A few comments remain:
> > > > - Section F.1 mentions Table 9 reports the number of parameters but they are not present in the table.
> > > > - You're still using dynamic for referring to dynamical systems, is there a reason to that? I am not aware of it being common terminology, but I might be wrong.
> > > > - Section F.1 in the revised blue text: "We also compares" $\to$ "We also compare".
> > > >
> > > > > Flag Datasets
> > > >
> > > > In my initial review, and as also requested by other reviewers, I asked about the flag datasets. I read in another of your replies that due to time constraints, this is challenging and you plan to add it in the final version. Were you able so far to obtain initial results? I still think this is critical for the paper and a proper evaluation of the model's performance against the baselines.
> > > >
> > > >
> > > > I will revise my rating once the above remaining points are addressed.

---

> > > > > ### Author Response · Authors · 2025-07-26
> > > > > **Authors' reply to odhF's Follow-up reply to authors**
> > > > >
> > > > > We appreciate the reviewer’s careful review of our revised manuscript and the follow-up comments. The manuscript has been further revised, and our point-by-point responses are outlined below.
> > > > >
> > > > > > Q2: Clarification regarding Figure 4 and Table 7
> > > > >
> > > > > - We fully agree with the reviewer and appreciate the reviewer’s understanding. Unfortunately, we have not yet identified a clearer way to display bars with varying properties and values without making the plot appear overly cluttered. An alternative approach would be to present those bars as separate plots in the appendix, where they can complement the main radar plot by providing additional detail.
> > > > >
> > > > > > Q6: Actual comparisons with the ground-truth simulators used for each dataset
> > > > >
> > > > > - We thank the reviewer for the suggestions and we've added those descriptions in the caption of Table 10 (now Table 13).  Yes, the simulator figures were generated on CPU using the saved prediction results.
> > > > >
> > > > > > Q12: (1) Could you add that it is averaged over time in the main text when specifying that you use mesh quality metrics?
> > > > >
> > > > > - Thanks for pointing that out and we have added that in the caption of Table 3.
> > > > >
> > > > > > Q14: Address the typos and miscellaneous stuff listed above
> > > > >
> > > > > - We've addressed all these points in the newly revised manuscript.
> > > > >
> > > > > > Flag Datasets
> > > > >
> > > > > - We have included the initial results (full ablation study results are still on the way) in the revised manuscript (Table 5) and also provide them below here for the reviewer's convenience:
> > > > >
> > > > > | Dataset      | Model        | GFₙ (×10⁻²)           | GFₑ (×10⁻⁵)           | MC (×10⁻²)           | AR (×10⁻²)           | RMSE‑all           | Train Mem. [MB] | Test Time [ms] |
> > > > > |--------------|--------------|-----------------------|-----------------------|----------------------|----------------------|--------------------|-----------------|----------------|
> > > > > | **Flag** | MGN          | 1.82 ± 0.08           | 5.01 ± 0.34           | 6.02 ± 0.59          | 4.16 ± 0.05          | 0.25 ± 0.01        | 1060            | 36.5           |
> > > > > |              | EAGLE        | 1.73 ± 0.08           | 5.22 ± 0.56           | 6.71 ± 0.55          | 5.49 ± 1.05          | 1.01 ± 1.14        | 1336            | 41.9           |
> > > > > |              | **M4GN (Ours)** | **0.98 ± 0.05**       | **2.23 ± 0.19**       | **4.06 ± 0.40**      | **3.11 ± 0.62**      | **0.15 ± 0.01**    | **549**         | **30.6**       |
> > > > >
> > > > > In addition to the changes mentioned above, we have made further revisions throughout the paper to clarify our contributions and ensure that our claims are well-supported. We have also added new experimental results in the appendix to strengthen our findings, along with a new summary table (Table 7) that outlines the key hyperparameters used in our method, their tested ranges, observed performance impacts, and practical tuning recommendations based on the sensitivity analysis presented in Appendix D.
> > > > >
> > > > > Once again, we sincerely thank the reviewer for their time and thoughtful feedback throughout the review process. We are more than willing to address any further questions or suggestions.

---

> > > > > > ### Comment · Reviewer_odhF · 2025-07-26
> > > > > > **Review follow-up comment**
> > > > > >
> > > > > > Thank you for addressing these last points. For Q2, I indeed believe some clarifying plots or summary tables would help, even if I'm still not entirely convinced by the radar plots in the main text like this to convey the message but I understand the authors do not want the paper to be too cluttered. Thank you for providing results on the Flags dataset, as well as the other changes in the revision.
> > > > > >
> > > > > > I suggest accepting the paper with these revised changes, and I strongly recommend the authors to provide supplementary material with videos that give better details of the comparisons with baselines and GT.

---

### Review · Reviewer_ufzj · 2025-06-26

**Summary Of Contributions:**

# Updates after revision
Author adopted most of my suggestions. I am recommending acceptance now.

# Old Overall rating: **Major revision**

## Overall comments:
This paper combines MP-GNN and GraphTransformer for multi-scale information exchange in physics simulation on meshes. The methodology presents a solid improvement compared to EAGLE[1], specifically by: 1) incorporating a microscale level MP-GNN and 2) implementing a better mesh segmentation design. The experimental results are comprehensive, but the writing somewhat exaggerates the narrative in a "fancy" way that hinders clarity. While I recognize the method's technical contributions and improvements, I recommend a **major revision** focusing on writing clarity and reducing unnecessary embellishment of the "fancy" story-telling before publication.

Detailed comments are provided in later sections.

[1] EAGLE: Large-Scale Learning of Turbulent Fluid Dynamics with Mesh Transformers. https://arxiv.org/abs/2302.10803

**Audience:**

Yes

**Claims And Evidence:**

Yes

**Requested Changes:**

## Requested Changes:

The following changes should be **incorporated into the new manuscript for the benefit of readers:**

**Very important**:

- Clearly identify improved mesh segmentation as a key contribution and relocate the entire Section 3.3 to stand as a separate pre-processing section prior to model description. This will properly frame it as part of pre-processing rather than the forward training pipeline, improving logical flow.

- Remove the "meso-scale" terminology from the paper's narrative. Geometric segmentation is not a standard interpretation for meso-scale effects in physics and appears to be used primarily as a buzzword.

- Restructure the current Section 3.3's introduction to provide an immediate high-level overview of your segmentation method before diving into details. Currently, readers must reach Section 3.3.3 to understand your hybrid approach involves: 1) METIS as initial hot start, and 2) K-Means clustering using features that include modal decomposition. Introducing this framework earlier would significantly reduce confusion.

- Move the ablation study on segmentation types to the main results section, as this represents one of your primary contributions.

**Added results or technical details**:

- Address the time complexity of your hybrid segmentation approach for Lagrangian systems where segmentation may vary with time (changing world configuration). Since segmentation must be performed during inference, time complexity report is essential.

- Strengthen your results section with 1-2 additional demonstrations: either showcase scenarios where other methods (like EAGLE[1]) catastrophically fail (similar to the approach in your referenced BSMS-GNN[2] paper), or present a zero-shot demonstration for a scaled-up configuration not seen during training to highlight the GNN family's generalization capabilities.

- Consistently specify default values for all meaningful settings discussed in ablation studies (e.g., overlap layers). Remove unused notations (e.g., policy π which appears once but is never used again). Review the entire paper for similar instances of unnecessary or incomplete information.

[1] EAGLE: Large-Scale Learning of Turbulent Fluid Dynamics with Mesh Transformers. https://arxiv.org/abs/2302.10803

[2] Efficient Learning of Mesh-Based Physical Simulation with Bi-Stride Multi-Scale Graph Neural Network; https://proceedings.mlr.press/v202/cao23a/cao23a.pdf

**Strengths And Weaknesses:**

## Strengths:
- Comprehensive ablation studies of all involved components
- Clear writing overall; all notations are well-defined without excessive technical jargon, making all parts easy to understand
- Solid improvements to existing methodology that can be easily migrated and deployed in future studies or real applications

## Weaknesses:
- Several elements of the paper suggest the authors prefer "fancy storytelling" over honest and clear description of different components. This is evident in both text and figures:
- Text-wise: The Meso scale is essentially a made-up concept. The authors present it as part of the training pipeline when in reality, it is a pre-processing step. (I recognize their design and ablation study provides better pre-processing, but misrepresenting it as part of forward/training is problematic). This contributes to the convoluted logical flow in Section 3.3.
- Figure-wise: The figures use vivid colors and elements and are well-drawn but some lack clarity. For example, Figure 1, the main schematic plot for their method, has no caption at all. This figure also lacks any legend for micro/meso/macro scales, suggesting the multi-scale storytelling may have been retrofitted. Figures 2(a) and (c) are visually pleasing but significantly harder to interpret for baseline comparison. Basic bar/line plots with error bars would be more effective.
- (Minor) Some settings such as default values for overlap layers are not specified.

---

> ### Author Response · Authors · 2025-07-12
> **Reply to Reviewer ufzj**
>
> We'd like to sincerely thank the reviewer for providing valuable feedbacks. Below, we summarized the changes we've made to the manuscript and the answers to reviewer's questions:
>
> 1. We appreciate the suggestions from the reviewer regarding improving manuscript clarity and made the following revisions to the manuscript: (i) explicitly highlights our mesh-segmentation contribution, (ii) removes the ambiguous “meso” terminology, (iii) inserts a dedicated pre-processing section on the segmentation method before the framework description, and (iv) places the segmentation analysis within the main results.
>
> 2. We added the discussion regarding time complexity of our hybrid segmentation method in the Complexity Analysis section in Appendix F.2.
>
> 3. Regarding strengthen our results section to show case catastrophically fail for other methods, we've updated the scaled-up generalization visualizations in Figure 9. The results demonstrate that our method generalizes well to larger systems. In contrast, other hierarchical methods show large prediction errors and fail to capture the correct dynamics in these zero-shot scaled-up simulations. Besides, in Figure 12-15 we visualize simulation predictions across multiple baselines including EAGLE. These examples highlight cases in which strong baseline models like EAGLE exhibit substantial deviations from the ground truth, while our method consistently maintains high fidelity.
>
> 4. We have included a new table specifying the hyperparameters of models for each dataset in Appendix B.1. We have went through the paper and removed unnecessary notations.
>
> 5. To improve figure clarity, we have removed the original Figure 1 to avoid potential confusion with the updated manuscript structure and revised the original Figure 2 with enhanced caption descriptions. In the updated Figure 3(a), we retained the radar plot format as it more effectively illustrates the trade-off between accuracy and efficiency across methods compared to alternative plot types. For Figure 3(c), we explored bar and line plots; however, these resulted in visual clutter due to overlapping lines. To aid reader interpretation, we instead maintained the current format and added further explanatory details in the caption.
>
> Once again, we thank the reviewer for these thoughtful comments and would welcome any additional suggestions that could help further improve our manuscript.

---

> > ### Author Response · Authors · 2025-07-26
> > **Authors' Follow-up Reply to Reviewer ufzj**
> >
> > We appreciate the initial comments raised by the reviewer and have made major revisions to address these points. In addition to the changes mentioned above, we have made further revisions throughout the paper to clarify our contributions and ensure that our claims are well-supported. We have also added new experimental results in the appendix to strengthen our findings.
> >
> > Once again, we sincerely thank the reviewer for their time and thoughtful feedback. We are more than willing to address any further questions or suggestions.

---

> > > ### Comment · Reviewer_ufzj · 2025-08-02
> > > **Reply to authors**
> > >
> > > Dear authors,
> > >
> > > I find your revision impressive; it adopts most of my suggestion, most importantly, corrected the storyline of this work and removed tech jagorns, clearly showing your contribution.
> > >
> > > Given this, I am recommending acceptance.
> > >
> > > Best

---

### Author Response · Authors · 2025-07-12
**Updates on Manuscript**

Dear Reviewers:

Thank you for the time and thoughtful feedback you invested in evaluating our manuscript. We have just uploaded a revised version that incorporates many of your suggestions, and we are drafting a detailed, point‑by‑point response to each of your comments. We will post that reply shortly.

Best regards,
TMLR Paper4952 Authors

---

### Author Response · Authors · 2025-07-30
**Appreciation for the Review Process**

Dear Reviewers,

We sincerely appreciate your recognition of our revised manuscript and your thoughtful discussions and constructive suggestions. We are also grateful for the time and effort you devoted to reviewing our work and contributing to this process. Your feedback has been invaluable in helping us improve the quality and clarity of our paper.

Best regards,
Authors of TMLR Paper4952

---

### Decision · Action_Editor_ADCz · 2025-08-13

**Recommendation:** Accept as is

**Additional Comments:**

I want to thank both the reviewers and the authors for their very constructive discussion and scientific attitude. The paper has improved significantly during the process.

**Audience:**

Yes

**Audience Explanation:**

Building fast and accurate emulators of dynamical systems is a topic of direct interest for researchers working on AI for Science.

**Claims And Evidence:**

Yes

**Claims Explanation:**

The claims of the manuscript are supported by thorough empirical evidence. The paper improved significantly during the rebuttal period through successive revisions. It now presents clearly and honestly the results and conclusions.